# Deciphering the atomistic mechanism underlying highly tunable piezoelectric properties in perovskite ferroelectrics via transition metal doping

Peng Tan [1,8], Xiaolin Huang[1,8], Yu Wang[1,8], Bohan Xing[1], Jiajie Zhang[2], Chengpeng Hu[1], Xiangda Meng[1], Xiaodong Xu[3], Danyang Li[4], Xianjie Wang[1], Xin Zhou[5], Nan Zhang [2], Qisheng Wang [6], Fei Li [2]✉, Shujun Zhang [7]✉ & Hao Tian [1]✉

Piezoelectricity, a fundamental property of perovskite ferroelectrics, endows the materials at the heart of electromechanical systems spanning from macro to micro/nano scales. Defect engineering strategies, particularly involving heterovalent trace impurities and derived vacancies, hold great potential for adjusting piezoelectric performance. Despite the prevalent use of defect engineering for modification, a comprehensive understanding of the specific features that positively impact material properties is still lacking, this knowledge gap impedes the advancement of a universally applicable defect selection and design strategy. In this work, we select perovskite $KTa_{1-x}Nb_xO_3$ single crystals with orthorhombic phase as the matrix and introduce Fe and Mn elements, which are commonly used in "hard" ferroelectrics as dopants. We investigate how transition-metal doping modifies piezoelectric properties from the perspective of intrinsic polarization behaviors. Interestingly, despite both being doped into the B-site as an acceptor, Mn doping enhances the local structural heterogeneity, greatly bolstering the piezoelectric coefficient beyond 1000 pC/N, whereas Fe doping tends to stabilize the polarization, leading to a substantial improvement in the mechanical quality factor up to 700. This work deciphers the diverse impacts of transition metal impurities on regulating polarization structures and modifying piezoelectric properties, providing a good paradigm for strategically designing perovskite ferroelectrics.

Over the past 70 years, piezoelectricity has been an integral property of perovskite ferroelectrics[1]. Renowned for their large spontaneous polarizations ($P_s$), these materials excel in efficiently converting both mechanical and electrical energies, establishing themselves as excellent piezoelectric materials. With the exceptional piezoelectric coefficients and electromechanical coupling factors, perovskite ferroelectrics serve as the cornerstone for electromechanical systems across macro, micro, and nano scales[2]. As device and application standards become increasingly stringent, the continuous pursuit of comprehensive

improvements in piezoelectric properties remains an ongoing objective.

Great efforts have been devoted to introducing phase instabilities for enhancing piezoelectricity by facilitating polarization variation, including the establishment of morphotropic phase boundary (MPB) and local structure heterogeneity (LSH) or polar nanoregions (PNR)[3–5]. To precisely control the polarization structure, a solid-solution approach has been generally employed by introducing different transition metals into the A-site or B-site of the $ABO_3$ perovskite structure[6]. Transition metals are elements characterized by partially filled $d$ orbitals or the ability to form them. Leveraging their diverse $d$-electron configurations, these elements serve as valuable models for the modification of perovskite ferroelectrics, allowing for the significant adjustment in $P_s$ by various oxidation and substitution states of transition elements[7]. An exemplary instance is lead zirconate titanate (PZT), where $Zr^{4+}$ or $Ti^{4+}$ occupies the B-sites, a crucial compositional foundation for piezoelectric ceramics since its invention in the 1950s[8]. In these solid solutions, constructing an MPB by adjusting the proportion of transition metals with the same lattice occupancy is a vital method for enhancing piezoelectricity[9]. Additionally, the presence of the random mixtures of heterovalent cations in the B-site, as observed in the case of $Pb(Mg_{1/3}Nb_{2/3})_{1-x}Ti_xO_3$ (PMN-PT), leads to strong random electric fields, promoting the formation of LSH and amplifying the piezoelectric response[10].

Of particular interest is that transition-metal impurity doping stands out as the most frequently employed method for regulating piezoelectricity. Surprisingly, even minute amounts of transition-metal doping in perovskite ferroelectrics, typically below 1 mol% content, can yield a substantial improvement in piezoelectric properties[11]. This type of impurity doping often involves different valence states compared to the substituted ions, qualifying it as an acceptor or donor defect. Acceptor defects tend to stabilize the domain structure, reduce dielectric loss, and enhance the mechanical quality factor, whereas donor defects increase domain activity and piezoelectric response[12]. The mechanism of trace doping is more intricate compared to materials without impurities or those in solid-solution systems. While most explanations have focused on domain wall motion, the characteristics of transition metals have been overlooked[13]. Iron and manganese serve as the most representative trace acceptor dopants. In PZT ceramics, $Fe^{3+}$ and $Mn^{3+,2+}$ are common acceptor substitutes for $Ti^{4+}$ or $Zr^{4+}$. Theoretically, the lower valence states of these dopant ions can introduce oxygen vacancies, create defect dipoles, and impede domain wall motion, being expected to achieve similar property modifications. However, the effects of Mn and Fe on piezoelectric properties differ greatly. PZT-4, predominantly doped with Mn, exhibits a piezoelectric coefficient $d_{33}$ of ~370 pC/N and a mechanical quality factor $Q_m$ of <500[14]. In contrast, PZT-8, mainly doped with Fe, has an inferior $d_{33}$ of <280 pC/N but a high $Q_m$ of nearly 1000[15]. This challenges the sole explanation based on domain wall motion and emphasizes the importance of considering the impact of transition metals on intrinsic polarization.

Therefore, exploring the intricate effects of transition-metal impurities on piezoelectric properties is essential. Piezoelectric ceramics, exemplified by PZT, showcase the diverse impacts of Mn, Fe, and other transition-metal impurities. However, understanding the nuances is challenging due to the complexities inherent in these ceramics and the presence of grain boundaries. For a clearer insight into the underlying mechanisms, high quality single crystals provide a valuable perspective. $KTa_{1-x}Nb_xO_3$ (KTN) solid-solution single crystal, a key member of lead-free ferroelectric materials, possesses a simple lattice, ease of doping, controllable phase structure, and a high degree of tunability in $P_s$ due to its infinite miscibility of $KNbO_3$ and $KTaO_3$[16,17]. Similar effects to those of Fe and Mn doping in PZT ceramics have been observed in tetragonal KTN single crystals[18]. This enables KTN a promising medium for studying transition-metal doping. However, in tetragonal KTN, the alignment of spontaneous and defect polarizations ($P_s$ and $P_D$) results in a pinning effect that binds the intrinsic polarization due to the symmetry-conforming principle[19]. Consequently, defect-modulated domain wall motion often dominates, making it difficult to fully observe the contribution of intrinsic polarization behavior. In contrast, the different orientations of $P_s$ and $P_D$ in orthorhombic phase facilitate polarization rotation.

Hence, this work explored orthorhombic KTN single crystals as the matrix and incorporated Fe and Mn as dopants to investigate how transition-metal impurity influences intrinsic polarization behaviors and piezoelectric properties. Our exploration delved into the micro–meso structure and macroscopic performances, aiming to clarify the pathways for controlling the piezoelectric properties through transition-metal doping. The thorough comprehension of transition-metal effects on the local structure and piezoelectric response, as observed in KTN crystals, establishes a foundation for the design of perovskite ferroelectrics.

## Results and discussion

### Transition-metal impurity versus properties

We used a modified top-seeded solution growth method to grow pristine, Fe-doped, and Mn-doped KTN single crystals[20]. The electron probe X-ray microanalyzer determined the Fe and Mn contents at ~0.5 mol% (Table S1). Figure 1A–C, J illustrates the dielectric properties of these crystals, showcasing high maximum dielectric constant ($\varepsilon_r$~10k at the Curie temperature) and low dielectric loss (tan$\delta$ < 0.026 at room temperature), indicating the high quality of the crystals. Through careful design of the Ta/Nb ratio, all crystals exhibit identical orthorhombic–tetragonal phase transition temperatures, $T_{O-T}$, of 50 °C, wherein the crystals are in orthorhombic phase at room temperature.

Remarkably, despite the similar doping amounts and phase transition temperatures, distinct variations emerge in their piezoelectric responses. All crystals were poled along the $[001]_C$ crystallographic direction to assess the intrinsic piezoelectric activity. The strain–electric field (S–E) loops and quasi-static $d_{33}$ measurements are given in Fig. 1D–F. The results reveal a highly linear relationship between the induced strain and applied electric field up to 20 kV/cm with minimal hysteresis. The indexes of strain hysteresis $Hys$ for pristine, Fe-doped, and Mn-doped KTN are on the order of 7.1%, 1.3%, and 4.2%, respectively, determined by $Hys = \Delta S_{Emax/2}/S_{Emax}$[21]. Here $S_{Emax}$ represents the strain at the maximum electric field $E_{max}$ and $\Delta S_{Emax/2}$ is the strain difference at half of $E_{max}$. The low $Hys$ values demonstrate the stable polarization structures after poling treatment. This underscores a typical feature of intrinsic piezoelectric response, with the piezoelectric strain coefficient up to 1000 pm/V for Mn-doped KTN. Consistent piezoelectric charge coefficients were observed in quasi-static $d_{33}$ measurements, with $d_{33}$ for the pristine, Fe-doped, and Mn-doped KTN single crystals measuring at 460, 530, and 1020 pC/N, respectively. Mn doping doubles the piezoelectric response, whereas Fe doping maintains a similar $d_{33}$ level to that of the pristine crystal. Additionally, the Mn-doped samples with different $T_{O-T}$ yet remaining in orthorhombic phase consistently exhibit enhanced $d_{33}$, as depicted in Fig. S1.

The difference of Fe and Mn doping effects extends to polarization characteristics. The hysteresis (P–E) loops and current density (J–E) curves of KTN samples are illustrated in Fig. 1G–I. The internal bias fields ($E_{int}$) in the Fe- and Mn-doped KTN single crystals, extracted into Fig. 1K, exhibit the "hardening" feature associated with acceptor doping. The largely increased $E_{int}$ in the Fe-doped crystal, reaching 0.9 kV/cm, provides a strong restoring force for polarization orientation. In stark contrast, although $E_{int}$ slightly increases due to the acceptor doping in the Mn-doped crystal, it does not exert strong coupling effect on polarization. Furthermore, we calculated the $Q_m$ of the poled samples, as depicted in Fig. 1L, by analyzing the frequency

dependence of resonance–antiresonance impedance, as illustrated in Fig. S2. The $Q_m$ of the Mn- and Fe-doped crystals are found to increase compared to that of the pristine crystal, being about 20% and 500% enhancement, respectively.

It is evident that both the Fe and Mn dopants induce the "hardening" effect, but their respective roles in the system differ significantly. The Mn-doped KTN crystals show a greatly enhanced intrinsic piezoelectric activity, while the Fe-doped KTN crystals exhibit a robust $E_{int}$, acting as a driving force stabilizing polarization and

contributing to an enhanced $Q_m$. Understanding the underlying micro–meso mechanism driving the distinct effects of Fe and Mn, despite their shared role as acceptors, is essential.

## Polarization revolution at microscale and mesoscale

The piezoelectric properties of perovskite ferroelectrics are closely related to their polarization structure[22]. Considering the trace impurity dopant amount in KTN crystals, a mere 0.5 mol%, the effectiveness of these dopants in impacting the overall polarization framework

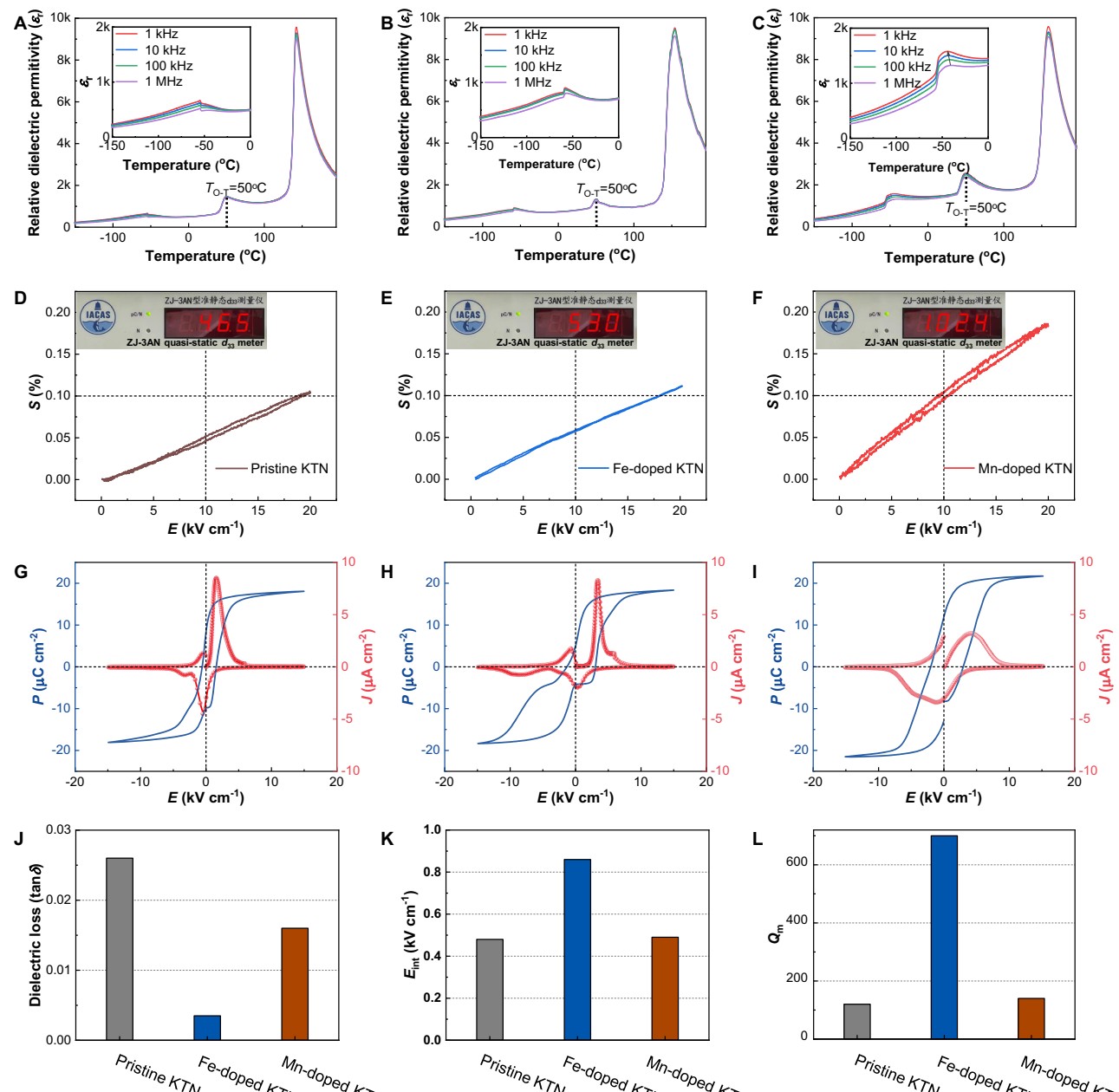

**Fig. 1 | Comparison of the various properties for pristine, Fe-doped, and Mn-doped KTN crystals.** **A**–**C** Dielectric constant $\varepsilon_r$ as a function of temperature and frequency, **D**–**F** strain–electric field (S–E) loops at 1 Hz, **G**–**I** polarization–electric field (P–E) loops, current–electric field (J–E) loops at 1 Hz, **J** dielectric losses tan $\delta$ at 100 kHz, **K** internal bias fields, $E_{int}$, extracted from P–E loops, and **L** lateral-mode mechanical quality factors, $Q_m$, for pristine, Fe-doped, and Mn-doped KTN crystals, respectively. The orthorhombic–tetragonal phase transition temperatures of 50 °C are marked in (**A**–**C**), and the insets show the corresponding low-temperature

dielectric properties in the range of −150 to 0 °C. The arrow in (**C**) inset indicates a frequency dependence of dielectric maxima at the rhombohedral-to-orthorhombic phase transition, showing a typical relaxor behavior. The insets in (**D**–**F**) are the photos giving the results of quasi-static $d_{33}$ tests. In the S–E, P–E, and $Q_m$ measurements, the samples poled along the [001]$_C$ crystallographic direction were used. The magnitude of $E_{int}$ is determined by $E_{int} = (E_+ + E_-)/2$, where $E_+$ and $E_-$ are the intersections of polarization loop with positive and negative electric field axis.

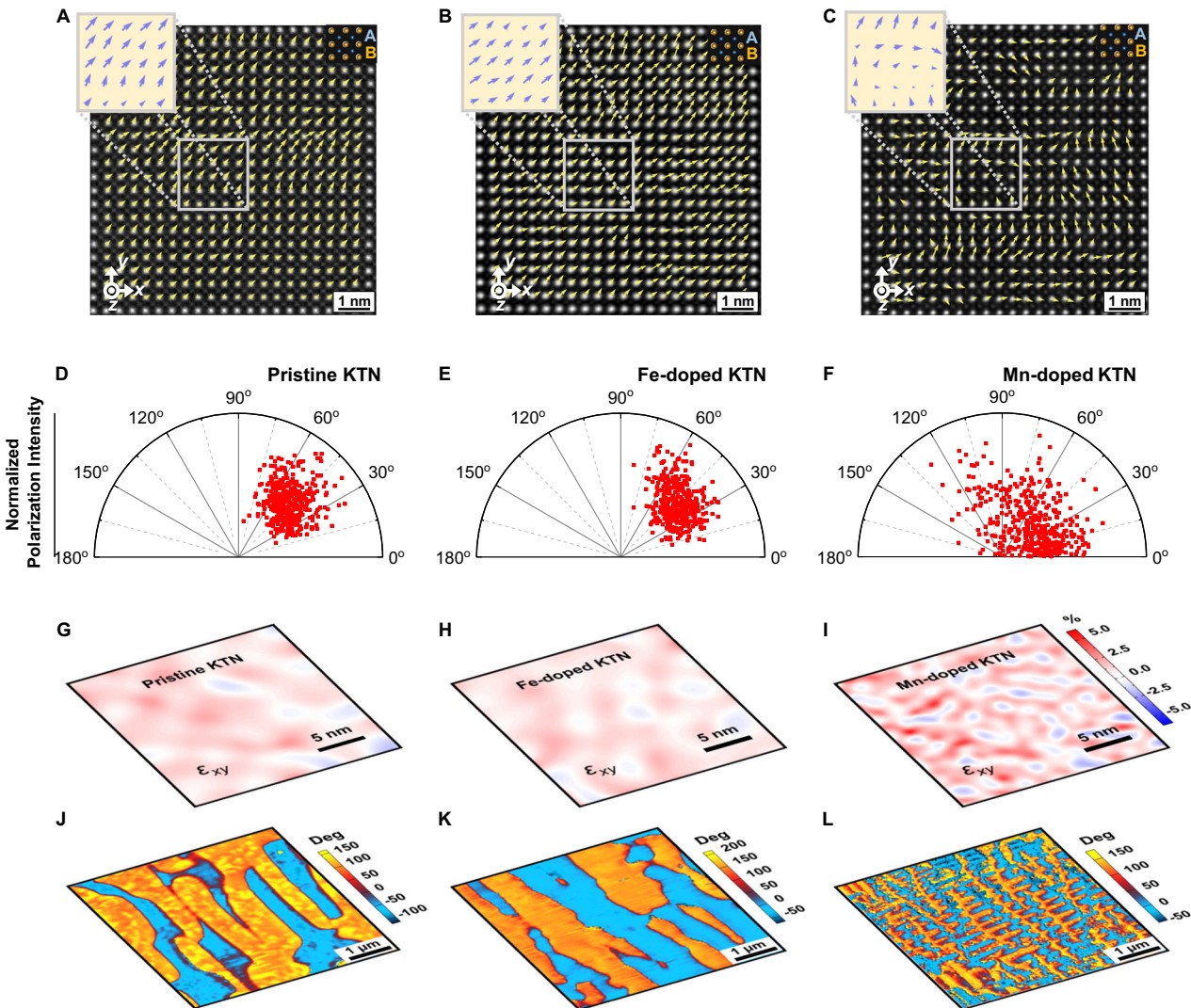

**Fig. 2 | Local structures of the crystals. A–C** The high-angle annular dark-field (HAADF) images of the pristine, Fe-doped, and Mn-doped KTN crystals, obtained along the $[001]_C$ direction at room temperature, respectively. The small and large bright points represent the A-site and B-site ions, respectively. Mappings of the polar vectors are marked by yellow arrows in (**A–C**). The atomic displacements are presented as vectors pointing from the center of a B-site cation to the center of its four nearest neighboring A-site cations. These atomic displacements represent the magnitudes and directions of the polar vectors for each unit-cell column. The *x, y,* and *z* directions correspond to the $[100]_C$, $[010]_C$, and $[001]_C$ crystallographic directions, respectively. The statistical scatterplots of the absolute value of the angle between the polar vector and the *x*-axis are shown in (**D–F**) for pristine, Fe-doped, and Mn-doped samples, respectively. **G–I** Distributions of shear strain $e_{xy}$ and **J–L** phase of piezoresponse force microscopy (PFM) images on the $(001)_C$ faces of the pristine, Fe-doped, and Mn-doped samples, respectively. The white color in the $e_{xy}$ distribution maps indicates the zero $e_{xy}$ strain regions. The $e_{xx}$, $e_{yy}$, and $R_{tot}$ distributions are shown in Fig. S4.

becomes pivotal in their capacity modifying the properties of the crystals. Figure 2A–C illustrates the microscale atomic configurations characterized by spherical aberration-corrected scanning transmission electron microscopy (STEM, imaged along the $[001]_C$ axis). The high-angle annular dark-field (HAADF) images distinguish between the A-site ions and B-site ions. The polarization vectors of the pristine, Fe-doped, and Mn-doped samples were deduced by analyzing the projected displacements of the B-site ions[23], encompassing both direction and magnitude, as depicted by the yellow arrows in Fig. 2A–C. In contrast to the uniform polarization in the pristine sample, the projected polarization vectors in the Mn-doped sample are highly disordered, whereas the Fe-doped sample retains a nearly consistent polarization orientation.

Figure 2D–F presents the statistical analysis of polarization distribution. In pristine KTN, the polar vectors predominantly align along the $[110]_C$ orientation, in agreement with the expected $P_s$ directions in orthorhombic phase. This alignment is similarly observed in the Fe-

doped sample. However, the Mn-doped sample deviates significantly, displaying a broad range of polarization projection orientations, indicative of substantial heterogeneity. Unlike the Fe dopant, the Mn dopant tends to disrupt the uniformity of polarization. Interface strain distribution maps in Fig. 2G–I, acquired via geometric phase analysis (GPA) across an extended area (Figs. S3 and S4), reveal that the Mn-doped sample experiences more notable lattice distortions compared to both pristine and Fe-doped samples, which exhibits a relatively similar extent of lattice distortion[24]. Such structural heterogeneity leads to the evolution of ferroelectric domains, as evidenced by piezoresponse force microscopy (PFM) and depicted in Fig. 2J–L. In Mn-doped KTN, a complex domain structure with a nested, branching morphology was observed, contrasting with the striped-like domains present in both pristine and Fe-doped KTN. The trace amount of Mn dopant results in a considerable reduction in the average size of the domains, with dimensions below 200 nm, compared to the 800 nm–1 µm range observed in pristine and Fe-doped crystals, as

given in Fig. S5. Similar results were also identified on a larger scale using polarized light microscopy (PLM), as shown in Fig. S6. For the $[001]_C$-poled samples, the Mn-doped crystal exhibits a higher density of O120 domain walls compared to its pristine counterpart when observed along the $[001]_C$ direction. Moreover, the Mn-doped sample exhibits curved domain walls, which tend to reduce the dipole–dipole interaction and electrostatic energies strengthened by disordered polar vectors[25]. The curvature of these domain walls is indicative of a larger interfacial energy which is a characteristic of structural heterogeneity. This delicate balance between interfacial and bulk energies is expected to flatten the local thermodynamic energy landscape, thus facilitating polarization rotations under an electric field[26].

The Mn-doped KTN crystal exhibits an intricate distribution in polar vector projection. However, since the investigated samples are all in orthorhombic phase, the projection vector can be oriented horizontally, vertically, or diagonally. Therefore, relying solely on polar vector mapping proves insufficient to fully unveil the crystal structural model. To enhance structural clarity, we conducted a three-dimensional synchrotron X-ray diffraction (XRD) study to characterize the differences in domain structure, as presented in Fig. 3. Due to diverse polar vector orientations in different domains, diffraction peaks stemming from the same crystallographic plane split in reciprocal space[27,28]. Figure 3C showcases the reciprocal space mapping of the $(400)_C$ Bragg scattering in the Mn-doped KTN crystal. The map indicates the presence of all six orthorhombic domains within the crystal, illuminated by an X-ray beam with a spot area of 628 μm². While both pristine and Fe-doped KTN crystals are also in orthorhombic phase, only partial orientations of the ferroelectric domains

were observable within the testing area, as illustrated in Fig. 3A, B. The modulus of scattering vector $|\mathbf{Q}|$ ($4\pi\sin\theta/\lambda$, where $\lambda$ is the wavelength of the incident X-ray) directly relates to the scattering angle $2\theta$, providing a clearer reflection of domain orientation distribution, as given in Fig. 3D. Hence, the Mn-doped crystal remains in orthorhombic phase, albeit with more disordered domain orientations compared to the pristine and Fe-doped crystals, with all orientations appearing in nearly equal proportions within the radiation area. This observed more disordered polarization agrees well with polar vector mapping attained by employing HAADF-STEM (Fig. 2C).

## Defect dipoles
Recognizing the substantial differences in the macroscopic performance and local structure of crystals demands our attention. The introduction of Fe and Mn impurities into KTN crystals leads to the formation of defects within the crystal lattice, due to the inequivalence between the dopants and the cations they replace. This defect structure plays a crucial role in shaping the properties of the crystals[19]. To delve deeper into the defect structure, we investigated the valence, occupancy, and derived defects of the Fe and Mn dopants using electron paramagnetic resonance (EPR) spectra, as shown in Fig. 4A. The pristine KTN crystal, lacking diamagnetism, does not exhibit any EPR signal[29]. On the contrary, the EPR resonances are clearly exhibited in the Fe-doped and Mn-doped samples. Sharp EPR signals around 345 mT suggest that Fe and Mn ions substitute for the Nb/Ta ions at the B-site, resulting in the formation of $Fe_B$ and $Mn_B$ defects, respectively[30,31]. The sextet hyperfine peaks in the Mn-doped sample arise from the interaction between the electron spin and nucleus spin of the Mn dopant

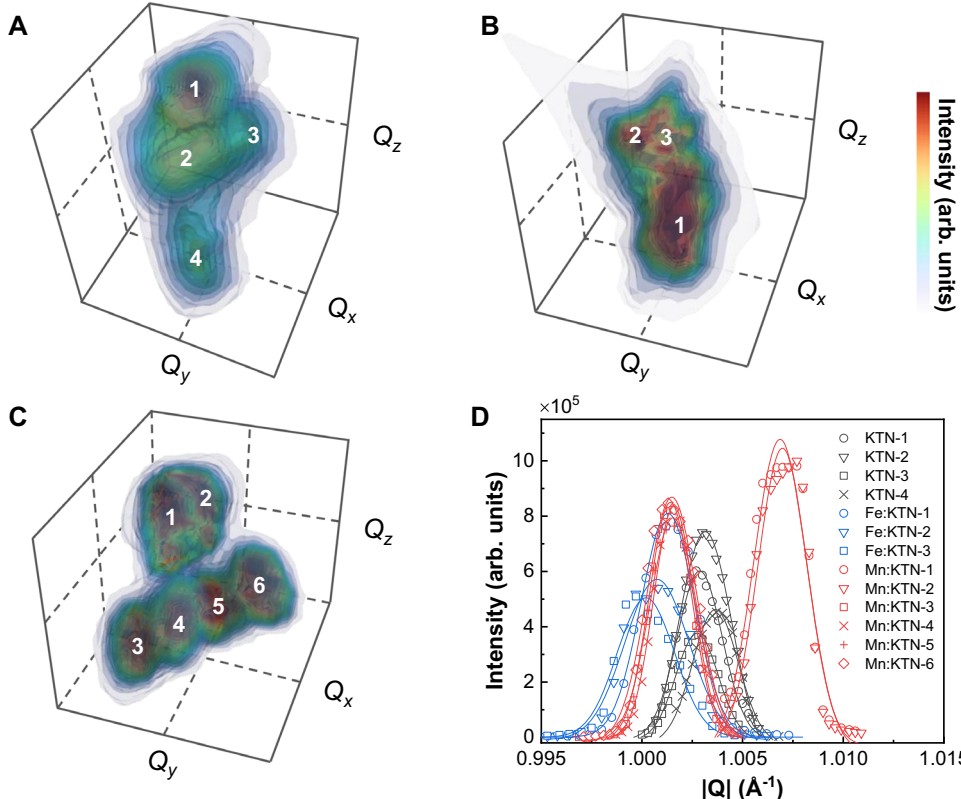

**Fig. 3 | Three-dimensional, synchrotron-based reciprocal space mapping around the $(400)_C$ Bragg diffraction. A** Pristine KTN crystal. **B** Fe-doped KTN crystal. **C** Mn-doped KTN crystal. The $Q_x$, $Q_y$, and $Q_z$ are reciprocal space coordinates of scattering vector $\mathbf{Q}$, where $|\mathbf{Q}| = (Q_x^2 + Q_y^2 + Q_z^2)^{1/2}$. $\mathbf{Q} = \mathbf{k_f} - \mathbf{k_i}$, where $\mathbf{k_i}$ is the wavevector of the incident X-ray beam and $\mathbf{k_f}$ is the wavevector of the scattered beam. The spot of the X-ray beam used was an ellipse with a long axis of 40 μm and

a short axis of 20 μm. Within the detection area, ferroelectric domains with different orientations are present. The color-mapped regions corresponding to domains with different polarization orientations are marked by numbers. The scattering intensity versus $|\mathbf{Q}|$ of all detected regions in pristine (black), Fe-doped (blue), and Mn-doped (red) samples are depicted in (**D**).

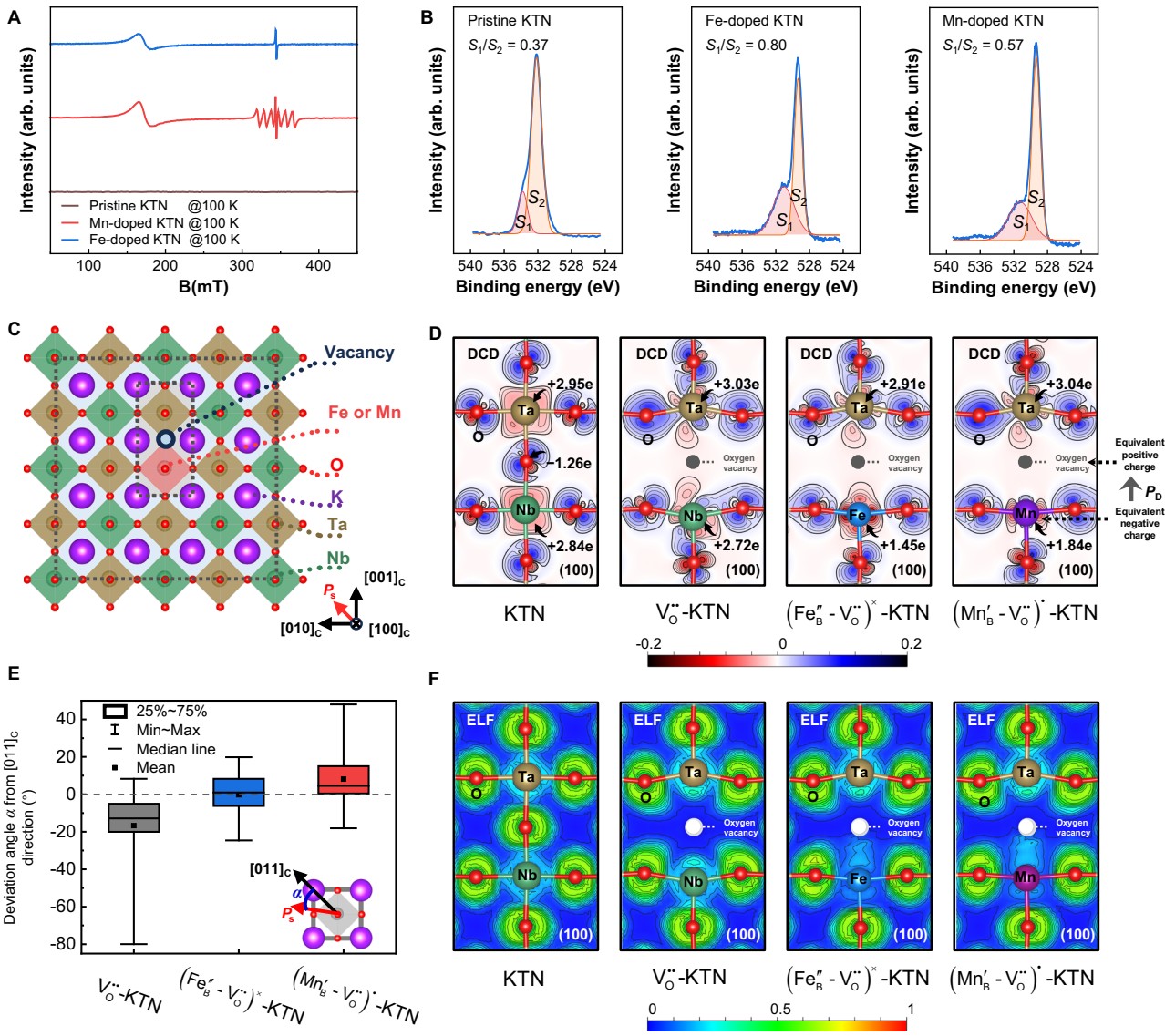

**Fig. 4 | Defect dipoles. A** Electron paramagnetic resonance (EPR) spectra of the pristine, Fe-doped, and Mn-doped KTN crystals at 100 K. **B** The O 1$s$ X-ray photo-electron spectra (XPS) of the pristine, Fe-doped, and Mn-doped samples. $S_1/S_2$ is the ratio of the integrated area of oxygen vacancy ($V_O^{··}$) and lattice oxygen peaks. **C** The schematic structure used in DFT simulation. The $[100]_C$, $[010]_C$, and $[001]_C$ crystallographic directions are given, and the spontaneous polarization $P_s$ in the pristine KTN is along the $[011]_C$ direction. In the doped lattices, a Fe or Mn ion substitutes a Nb ion, and a $V_O^{··}$ is created at one of the nearest neighboring oxygen ion sites of the dopant. **D** The $(100)_C$-section views of differential charge density

(DCD) of the pristine KTN lattice, the KTN lattice with $V_O^{··}$, the KTN lattice with $(Fe''_B - V_O^{··})^×$, and the KTN lattice with $(Mn'_B - V_O^{··})^·$, respectively, in the inside dotted box of (**C**). The Bader charges of the B-site cations and oxygen ion immediately adjacent to the vacancy site are marked. **E** Box charts of deviation angles of dipole moments from the $[011]_C$ direction in the octahedra within the $(100)_C$ sections containing the defect dipoles, statistically obtained from Fig. S8. **F** Two-dimensional contour plots of the electron localization function (ELF) on the $(100)_C$ sections.

($I = 5/2$ system)[32,33]. The Goldschmidt tolerance factors $t$ for the B-site doped Fe and Mn ion structures using the ionic radii (Tables S2 and S3) all fall within the range of the empirical structure rules ($0.81 < t < 1.11$), indicating the stability of the B-site substitutions[34].

As acceptor dopants, the lower valences of dopants compared to $Ta^{5+}$ and $Nb^{5+}$ result in the generation of oxygen vacancies ($V_O^{··}$) with positive charges to maintain charge balance. The ratios of $V_O^{··}$ and lattice oxygen were quantified using the O 1$s$ X-ray photoelectron spectroscopy (XPS) profiles[35], as illustrated in Fig. 4B. The increased integral area ratio of the $V_O^{··}$ and lattice oxygen peaks in the XPS profiles of the Fe-doped and Mn-doped samples indicates an increase in $V_O^{··}$. Notably, despite the same amounts of Fe and Mn doping, the increase in $V_O^{··}$ in the Fe-doped sample is almost twice that of the Mn-doped sample. From the viewpoint of charge equilibrium, the $Fe_B$ defect

should carry twice the negative charge of the $Mn_B$ defect. This matches $Fe^{3+}$ and $Mn^{4+}$ ions corresponding to the $Fe''_B$ and $Mn'_B$ defects, which implies that Fe and Mn ions in the doped KTN are mainly $Fe^{3+}$ and $Mn^{4+}$.

Moreover, $V_O^{··}$ exhibits a strong correlation with acceptors due to their propensity to generate around acceptors. The wide EPR peaks of the Fe-doped and Mn-doped samples at 166 and 174 mT, respectively, as shown in Fig. 4A, are a result of the strong spin–orbit coupling around the acceptor defects and $V_O^{··}$ [30,36]. The majority of $Fe''_B$ and $Mn'_B$ tend to trap $V_O^{··}$, forming the $(Fe''_B - V_O^{··})^×$ and $(Mn'_B - V_O^{··})^·$ defect dipoles, respectively. These defect dipoles affect the local structure and macroscopic properties greatly. The presence of defect dipoles increases $E_{int}$ and inhibits the motion of domain walls, thus enhancing $Q_m$[37]. Despite possessing similar defect structures, however, the two transition-metal dopants exhibit significantly different piezoelectric

responses. This divergence in performance probably arises from the nature of the electronic configuration of the transition metals.

## Differences in the defect dipoles from an electronic configuration perspective

Transition-metal dopants, characterized by diverse $d$-orbital configurations, offer a range of strategies for polarization control in perovskite ferroelectrics. To comprehend the differences in electronic configurations between $(Fe''_B - V_O^{··})^{×}$ and $(Mn'_B - V_O^{··})^{·}$ defect dipoles, we analyzed their features using density functional theory (DFT)[38,39]. The local structural characteristics were captured by using a $4 × 4 × 4$ supercell with 320 atoms[40]. Due to the preference for the lowest formation energy (Fig. S7) in combination with the strong binding effect of the acceptor on the $V_O^{··}$, we replaced the Nb ions with the dopants and placed the $V_O^{··}$ at the closest oxygen site of the dopant, as schematically shown in Fig. 4C.

The introduction of defects into the lattice inevitably leads to a variation in the local structure, thereby affecting the inherent polarization. The change in charge distribution is the primary source of local structural distortions. Figure 4D depicts the differential charge density (DCD) of the simulated structures, covering pristine KTN, $V_O^{··} - KTN$, $(Fe''_B - V_O^{··})^{×} - KTN$, and $(Mn'_B - V_O^{··})^{·} - KTN$. Meanwhile, the Bader charges $Q_B$ of the B-site cations and oxygen ion immediately adjacent to the vacancy site are labeled in Fig. 4D. In the pristine KTN lattice, electrons transfer from the B-site Nb or Ta ions to adjacent oxygen ions, forming a stable oxygen octahedral configuration. The heterovalent doping often creates $V_O^{··}$. When only concerning the effect of $V_O^{··}$, the $V_O^{··}$ notably reduces electrons at the defect site, forming an equivalent positive charge center. This yields the local structural distortion and disrupts the pattern of charge transfer, as the DCD of $V_O^{··} - KTN$ shown.

The doped Fe and Mn ions can compensate for the electron reduction in the defect structure due to their smaller $Q_B$ than the original B-site ions. In the Fe-doped lattice, the decrease in $Q_B$ at the doped site closely matches the increase in $Q_B$ at the vacancy site, which benefits the local charge equilibrium and yields a stronger compensation effect. This enables the charge transfer case to be more similar to that of the pristine lattice, thereby reducing lattice distortion. Figure 4E shows the statistics of the distribution of dipole moments within the lattices. In the Fe-doped lattice, the average orientation of the dipole moments on the $(100)_C$ section containing the defect dipole, remains aligned along the $[011]_C$ direction, i.e., the uniform orientation of $P_s$ in the pristine KTN model.

In terms of the $Q_B$ of Fe and Mn ions (+1.45 e and +1.84 e, respectively), the Mn ion loses nearly ¼ more electrons than the Fe ion, consistent with the nature of positive trivalent Fe ion and positive tetravalent Mn ion. In the Mn-doped lattice, in addition to the weaker electron compensation effect than in the Fe-doped case, the smaller radius of $Mn^{4+}$ ($r = 0.53$ Å) further expands the void caused by the vacancy. This is in contrast to the Fe-doped lattice, where the radius of $Fe^{3+}$ ($r = 0.645$ Å) is nearly identical to that of the original B-site ions ($r = 0.64$ Å)[41]. The increased spacing of the Mn ion from the neighboring oxygen weakens the bonding, especially on the Mn–O bond located on the side away from the vacancy, as shown by the DCD of Mn-doped lattice in Fig. 4D. These local differences enhance the lattice distortion within the Mn-doped lattice, undermining the order of local polarization.

Moreover, according to the nominal oxidation states of the ions, i.e., $Fe^{3+}$, $Mn^{4+}$, $Ta^{5+}$, $Nb^{5+}$, and $O^{2-}$, the $Fe''_B$ and $Mn'_B$ defects exhibit −2 valence and −1 valence, respectively, while the stabilized $V_O^{··}$ acquires a positive bivalence state after crystallization. Therefore, following the formation of defect dipoles, the charge balance is maintained in the Fe-doped system, whereas Mn doping results in a localized +1 valence net charge, leading to the presence of random fields. The random fields from Mn doping yield relaxor behavior, as witnessed by a pronounced frequency dispersion in dielectric behavior, along with the occurrence of dielectric relaxation observed in the low-temperature range of −150 to 0 °C, as depicted in Fig. 1C. Heterogeneous charge distributions largely impact local structures through random electric fields, thereby also exacerbating the disorder of local polarization[10].

Defect dipoles formed by the $V_O^{··}$ and B-site heterovalent ions largely influence the behaviors of ferroelectric polarization. Figure 4F depicts the electron localization function (ELF) representations for the pristine and defective lattices. The ELF of $V_O^{··} - KTN$ agrees with the case of doubly charged oxygen vacancies[42]. In terms of the Bader charge, the $V_O^{··}$ and doped ions form the equivalent positive charge center and equivalent negative center, respectively. This drives the enhancement of electrostatic interactions between $V_O^{··}$ and dopant ions, changing the spatial representation of the electron distribution, as illustrated in ELF. The ELF distribution can reveal polarization characteristics[43,44]. The localized charge distribution around the defect dipole in the Fe-doped crystal is clearly more polarized, indicating the stronger defect polarization[45]. The strong and electrical neutral defect dipoles tend to stabilize the polarization ordering. The enhanced stability endows the Fe-doped sample with a "hardening" characteristic, as reflected in the increased $E_{int}$ and $Q_m$ as shown in Fig. 1K, L. The Mn-doped crystal, on the contrary, has relatively weak polarized defect dipoles, strong local distortions, and a random field effect around the defect structure, all of which favor a disordered polarization orientation. This echoes the STEM observations, as illustrated in Fig. 2C, F. This facilitates the enhancement of local structural heterogeneity, which is recognized as a key factor in boosting piezoelectric activity.

## Polarization dynamics and its roles in piezoelectric response

Defect dipoles $(Mn'_B - V_O^{··})^{·}$ and $(Fe''_B - V_O^{··})^{×}$ induce distinct polarization structures ranging from lattices to ferroelectric domains. The dynamics of polarization, as a key origin determining the functionality of perovskite ferroelectrics, intricately shape the piezoelectric properties. The Landau free-energy landscapes serve as the foundation for evaluating the resistances during polarization rotation, an important indicator of piezoelectric activity[3,26]. The structural heterogeneity, arising from disordered polarization orientations and lattice distortions, tends to induce interfacial energies encompassing electrostatic, elastic, and gradient components[46]. The interplay between the bulk Landau energy and interfacial energy can lead to diverse free-energy landscapes in systems with varying degrees of disorder. In structures exhibiting a greater level of disorder, the free-energy profile tends to be more flattened, thereby facilitating the polarization rotation, as illustrated in Fig. 5A.

The evidence of the flattened free-energy profile was observed in Mn-doped samples, that is, a more accessible field-induced phase transition. The $P$–$E$ loops and $S$–$E$ curves of the poled samples, shown in Fig. 5B, C, were obtained under an applied electric field up to 30 kV/cm. Notably, abrupt changes in polarization and strain were detected at a threshold electric field of 23 kV/cm. The current density $J$, depicted in Fig. S9, also displays an obvious deviation at the same electric field. These variations in the $P$–$E$ and $S$–$E$ loops symbolize field-induced phase transition[47]. The ease of field-induced phase transition signifies that the disordered local structure in the Mn-doped sample reduces the thermodynamic barrier separating the orthorhombic and tetragonal phases, thereby manifesting a more flattened free-energy profile.

The dynamic evolution of phase structures under the action of an electric field is evident in Raman spectroscopy[48]. A theoretical connection between lattice vibrations and Raman eigenpeaks, employing density functional perturbation theory (DFPT)[49], was established. Because the polarization directly relates to the $BO_6$ octahedron, the nondegenerate $A_1$ and $B_1$ vibration modes of $BO_6$ octahedron were studied. As depicted in Fig. 5D, the Raman peaks at 509 and 536 cm$^{-1}$ indicate the approximately mirrored vibration modes $B_1(TO3)$ and $A_1(TO3)$ of KTN in orthorhombic phase, respectively. Upon transition

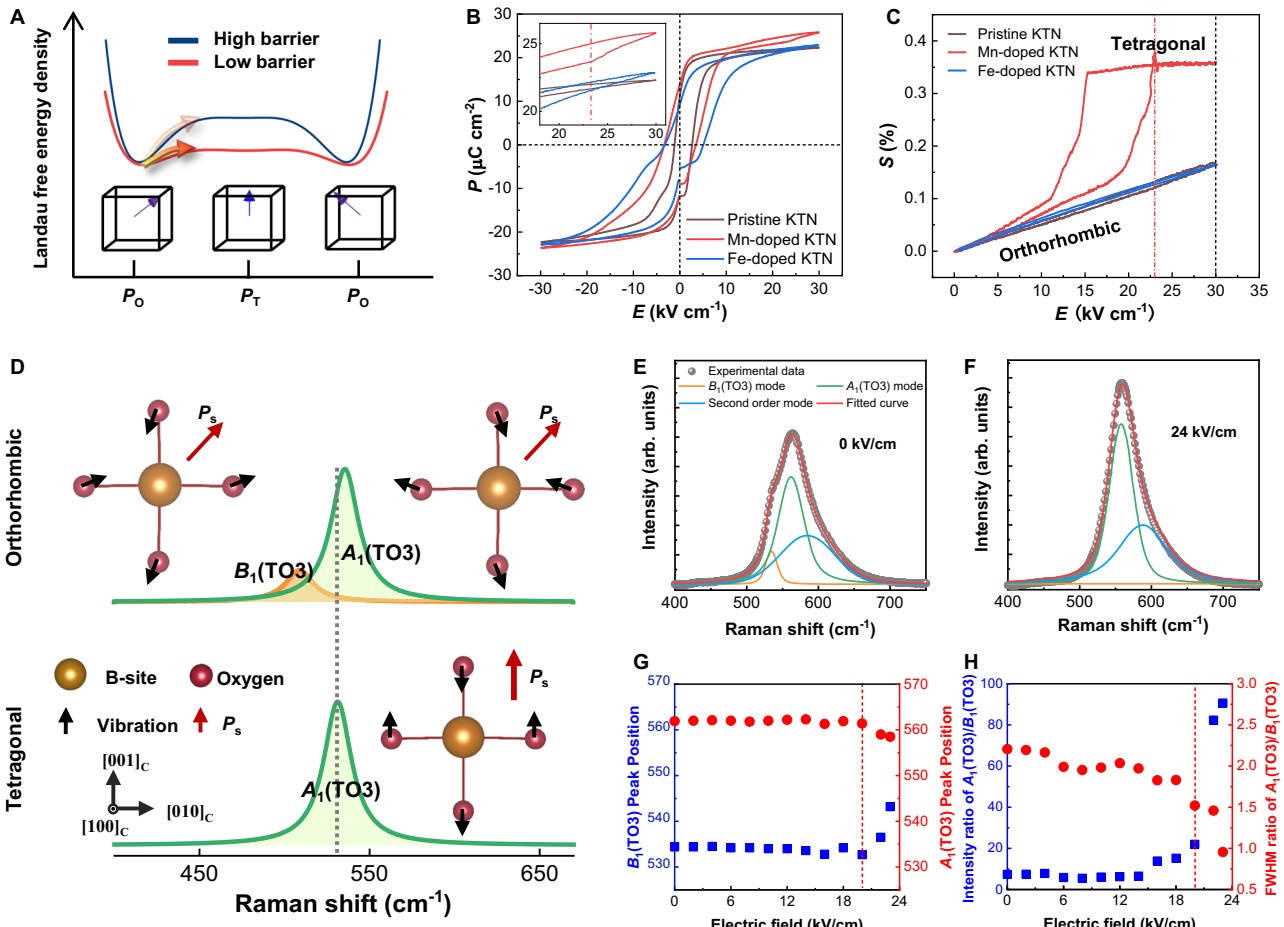

**Fig. 5 | Polarization dynamics and field-induced phase transition. A** Schematic of Landau free-energy landscapes. $P_O$ and $P_T$ represent the spontaneous polarizations in orthorhombic phase and tetragonal phase, respectively. Polarization rotation from $P_O$ to $P_T$ due to the electric field is more likely to occur in the low-barrier case. **B** Polarization and electric field ($P$–$E$) loops under a maximum electric field (1 Hz) of 30 kV/cm. The inset shows a detailed image above 20 kV/cm. **C** Unipolar strain–electric field ($S$–$E$) curves with a maximum electric field (1 Hz) of 30 kV/cm. **D** The $A_1$(TO3) and $B_1$(TO3) Raman vibrational modes of orthorhombic and tetragonal phase structures, simulated via the DFPT method. Accompanying the Raman shifts are the schematics depicting the vibrational modes, with the red arrows indicating the directions of spontaneous polarization and the black arrows indicating the directions of atomic vibrations. **E, F** Raman spectra at an $E$ of 0 and 24 kV/cm. The global field-induced phase transition is completed under the $E$ of 24 kV/cm. The $B_1$(TO3), $A_1$(TO3), and second-order modes were fitted using the Voigt line shape. **G** Shifts of $B_1$(TO3) and $A_1$(TO3) peaks under the action of $E$. **H** Ratio evolutions of intensity and full width of half maximum (FWHM) of $A_1$(TO3) and $B_1$(TO3) under the action of $E$.

to tetragonal phase, these vibrational modes evolve into a singular $A_1$(TO3) mode, as evidenced by the Raman peak at 531 cm⁻¹, due to the change in structural symmetry[50]. The Raman spectral evolution of Mn-doped crystals under different applied electric fields was characterized, as shown in Fig. S10. The Raman peaks in the 400–700 cm⁻¹ range contain the $B_1$(TO3), $A_1$(TO3), and second-order vibrational modes, fitted via the Voigt line shape[51]. The Raman spectra and fitting results across electric fields of 0 and 24 kV/cm are illustrated in Fig. 5E, F, respectively. The third fitted peak at ~590 cm⁻¹, associated with a second-order mode in KTN crystals, stems from the combination of the transverse $A_1$(TO3) mode and the acoustic TA mode at the critical $X$-point of the Brillouin zone[52,53]. The full Raman spectra are compared between theory and experiment, as given in Fig. S12.

In the $S$–$E$ curve, the onset of the field-induced phase transition is observed as the applied electric field approaches 20 kV/cm. This characteristic is evident in the Raman spectra, where changes in peak position, peak intensity, and full width of half maximum (FWHM) with varying electric fields represent the initiation of a phase transformation[54]. The electric-field-dependent Raman peak evolution shows an obvious shift around an electric field of 20 kV/cm, as illustrated in Fig. 5G, H. With the continued increase in electric field, the peak of the $B_1$(TO3) mode gradually weakens, while the peak of the $A_1$(TO3) mode gradually strengthens. Ultimately, the $B_1$(TO3) vibrational mode completely vanishes beyond the threshold electric field of the field-induced phase transition. This is consistent with the results obtained from DFPT calculations, demonstrating the occurrence of a global field-induced phase transition. The continuous alteration in modes observed experimentally reflects an ongoing transformation process of the phase structure. This evolving pattern is an expression of structural symmetry alternations and local phase transformations induced by the easily occurring polarization rotation, an important feature of the flattened free-energy profile. The facilitated polarization rotation elucidates the underlying mechanism behind the high piezoelectric response in Mn-doped KTN samples. With the application of an electric field, more pronounced polarization rotation leads to a greater deformation, that is, higher piezoelectricity.

In the case of Fe doping, on the contrary, although the $(Fe''_B-V_O^{··})^{×}$ defect dipole does not result in significant structural heterogeneity, the reinforcement of polarization ordering and stability induces noteworthy performance modifications. An increased $E_{int}$ of 0.9 kV/cm strengthens the stability of ferroelectric polarization, resulting in $Q_m$ about a fivefold increase over the pristine KTN crystals.

Reflecting on the B-site Fe and Mn doping in PZT ceramics, akin to our observations in this research, the diverse impacts of the introduced defect dipoles on the structure might be a critical factor. Despite occupying the same lattice sites, the differences in the electronic configurations are sufficient to create distinctions in polarization dynamics, ultimately leading to the differences observed in $d_{33}$ and $Q_m$ values. Moreover, akin to the A-site doping systems[55,56], in this work, the local structural heterogeneity enhanced by the induced defect structures through B-site doping with transition metals, can also remarkably increase piezoelectric activity, showcasing the widespread effectiveness of structural heterogeneity in improving piezoelectric properties. With a highly disordered structure, the Mn-doped KTN crystal achieves a $d_{33}$ value exceeding 1000 pC/N, placing them among the elite in lead-free piezoelectric materials, as compared to Fig. S13. The incorporation of transition-metal impurities, along with the resulting defect dipoles, presents diverse avenues for the modification of piezoelectric properties.

In summary, the introduction of transition-metal impurities into perovskite ferroelectrics alters the lattice features, spanning from spontaneous polarization to structural heterogeneity. The specific electronic configurations within the defect structures exert a profound influence on a range of factors, leading to distortions in the local lattice, alterations in polarization, and shifts in local charge distribution. These factors intricately shape the polarization frameworks, exhibiting a remarkable and diverse impact on piezoelectric properties. In this context, both Fe and Mn dopants occupy the B-site, yet they exhibit distinct effects on KTN. Mn doping disrupts the continuity of polarization, introducing notable local distortion, and the resulting local structural heterogeneity elevates the $d_{33}$ value to above 1000 pC/N– twice that of pristine KTN crystals. Fe doping, on the contrary, enhances the ordering of polarization, yielding a "hardening" effect of properties, with $Q_m$ value fivefold improvement over the pristine KTN. This stark contrast highlights the potential of transition-metal impurities for versatile property modifications in perovskite ferroelectrics. The strategic selection of transition-metal dopants and engineering of functional defect dipoles open a rich array of possibilities for the design of perovskite ferroelectrics. This approach holds great promise not only for piezoelectric applications, but also for a range of polarization-related fields, such as ferroelectric catalysis, energy storage, electrocaloric solid-cooling, and cutting-edge sensing.

## Methods

### Growth of KTN-based crystals and sample fabrication

KTN-based single crystals were grown using a modified top-seeded solution growth approach[20]. According to the phase diagram of KTN[57], potassium carbonate ($K_2CO_3$, 99.99%), tantalum pentoxide ($Ta_2O_5$, 99.99%), and niobium pentoxide ($Nb_2O_5$, 99.99%) powders were weighed with the mole ratio 1.04:0.28:0.72. The excess $K_2CO_3$ was added as a self-flux to decrease the growth temperature. In the raw materials of 0.5 mol% Mn-doped and 0.5 mol% Fe-doped crystals, manganese oxide ($MnO_2$, 99.99%) powder and iron oxide ($Fe_2O_3$, 99.99%) powder were added to the mixture, respectively. For each crystal growth, the total weight of mixed powders was 150 g. The raw materials were subjected to ball-milling in anhydrous ethanol for 24 h, after which the mixture was baked at 150 °C. The dried mixture was put into a platinum crucible and calcined at 900 °C to synthesize the KTN-based compound. Thereafter, the compound was melted at 1200 °C. A [001]$_C$-oriented seed was lifted at the crystallization temperature to grow the KTN-based single crystals. The heating system was a medium-frequency induction furnace at 2 kHz in an air atmosphere, with the temperature controlled by a Eurotherm 818 controller (Eurotherm) to within ±0.3 °C.

The KTN-based crystals were oriented using a Laue X-ray machine and cut into rectangular bulks with the dimensions of $2_{[100]c} \times 1_{[010]c} \times 0.5_{[001]c}$ mm³ along the crystallographic directions. Gold electrodes were sputtered on both (001)$_C$ faces for electrical measurements. All the samples were annealed at 100 °C and aged for 2 weeks before experiments. Before structural characterizations, the samples were polished using polycrystalline diamond suspensions with abrasive particles of 3 μm, 1 μm, and 20 nm (MetaDi Supreme, Buehler). In the poling treatments, the doped and pristine samples were all poled along the [001]$_C$ direction. The KTN samples were first poled under an electric field of 15 kV/cm at 200 °C and then cooled to room temperature at a rate of 0.5 °C min⁻¹ under the applied electric field.

### Structural characterization

The composition of crystals was investigated using an electron probe microanalyzer (JXA-8230, JEOL). The EPR (Bruker A300) were measured at 100 K and room temperature. XPS (Escalab 250Xi, Thermo Fisher, Britain) equipped with a standard monochromatic Al excitation source was used to characterize oxygen vacancies. In situ synchrotron XRD experiments were performed at the beamline BL10U2 of the Shanghai Synchrotron Radiation Facility (SSRF) using synchrotron radiation with a wavelength of 0.6888 Å. The atomic configurations were characterized by spherical aberration-corrected STEM (imaged along the [001]$_C$ axis). STEM samples were prepared by a focused ion beam (Tescan LYRA-3 XUM Model). The high-resolution transmission electron microscopy (HRTEM) images, the HAADF images, and selected area electron diffraction (SAED) patterns were acquired on an FEI Titan G2 60-300. The method of GPA (in the Gatan Digital Micrograph software) was used to estimate the interface strains in HRTEM micrographs[24]. The vertical piezoresponse force microscopy (V-PFM) patterns were captured at room temperature using a commercial microscope (Cypher ES, Asylum Research) with conductive Pt/Ir-coating probes (EFM, Nanoworld). The domain structures were also observed at room temperature on the (001)$_C$ facets using in situ PLM (Axioskop40 ZEISS). The Raman spectra were measured via a Renishaw inVia confocal micro-Raman spectroscopy system using a 576 × 400 CCD array with a high resolution of 1 cm⁻¹. A 532-nm laser without polarization was employed for the Raman test.

### Properties measurements

The temperature dependence of relative dielectric permittivity ($\varepsilon_r$) and dielectric losses (tan δ) were obtained using an inductance–capacitance–resistance LCR meter (E4980A, Agilent Technologies) with a probing voltage of 1 V. The data acquisition interval was 1 K. The polarization–electric field hysteresis loops (P–E) and the S–E were measured using a ferroelectric tester (TF Analyzer 3000, aixACCT Systems) at 1 Hz and a 632-nm laser distance measuring device. The piezoelectric coefficient $d_{33}$ was measured using a quasi-static piezoelectric constant testing meter (ZJ-3AN, Chinese Academy of Sciences). The mechanical quality factors were characterized through a resonance–antiresonance impedance method using an impedance analyzer (HP4294A, Agilent).

### Simulations

The electronic configurations were calculated using the Vienna Ab-initio Simulation Package (VASP)[38,39]. Lattice constants of experimental orthorhombic crystal structure ($a = 3.985$ Å, $b \approx c = 3.992$ Å), measured by XRD, were used to construct the supercells. The XRD diffraction pattern is shown in Fig. S14. The $a$, $b$, and $c$ correspond to the [100]$_C$, [010]$_C$, and [001]$_C$ crystallographic directions, respectively. Projector augmented wave (PAW) method was used with a plane wave cut-off energy of 520 eV[58]. Spin-polarization was fully considered in all calculations. During structural optimizations, the exchange-correlation energy was calculated in Perdew–Burke–Ernzerhof (PBE) functional[59]. The KTN crystals were simulated in a 4 × 4 × 4 supercell with k-point sampling on a 3 × 3 × 3 grid. The ionic relaxations were performed with the energy and force convergence set at $1.0 \times 10^{-6}$ eV and 0.01 eV Å⁻¹, respectively. To ensure the lowest-energy configuration, the formation energies $E_{form}$ of the defect systems were estimated using the equation of

$E_{form} = E_{bulk}' + E_O - E_{bulk} + (E_{B\text{-}site} - E_d)$, where $E_{bulk}$ and $E_{bulk}'$ are the total energies of the pristine and doped KTN supercells, respectively, $E_{B\text{-}site}$ and $E_d$ are the energies of the original B-site atom and doped atom in the bulk phase, respectively, and $E_O$ is half of the energy of an isolated $O_2$ molecule[60]. To address the limitation of semi-local exchange-correlation PBE method[61], the nonlocal hybrid functional of Heyd, Scuseria, and Enzerhof (HSE06) was employed to accurately describe the electronic structure of the defect system[62]. A single k-point was used, which is sufficiently accurate for treating such a large defect system[63]. The DCD $\Delta\rho$ was calculated using $\Delta\rho = \rho_{total} - \rho_K - \rho_{Ta} - \rho_{Nb} - \rho_O - \rho_d$ where the subscripts "total" and "d" represent the defect system and dopant, respectively[64]. To calculate the Raman spectra, DFPT was conducted and implemented in the Quantum ESPRESSO software package, using norm-conserving pseudopotentials with a 100 Ry plane wave cut-off energy, and the local-density approximation (LDA) exchange-correlation function[49,65–67]. In the double perovskites $A_2B'B''X_6$, B-site cations favor rock salt ordering[68]. To describe the B-site cation ordering, the $\sqrt{2} \times \sqrt{2} \times 2$ KTN supercells with orthorhombic and tetragonal symmetries for DFPT calculations were constructed using the lattice parameters ($a = 3.985$ Å, $b \approx c = 3.992$ Å, and $a = b = 3.991$ Å, $c = 4.007$ Å, respectively), containing 20 atoms (4 K, 2 Nb, 2 Ta, 12 O) with k-point sampling on a $6 \times 6 \times 4$ grid. The lattice parameters of tetragonal KTN are cited from ref. 69. The Raman modes $A_1$(TO3) and $B_1$(TO3) in the range of 400–700 cm$^{-1}$ are both intrinsic vibrational modes of BO$_6$ octahedron, not perturbed by the supercell size. The structural information, overall magnetic moments of the systems, and the magnetic moments of each atom are listed in the Supplementary Information. The crystal structures and maps of DCD were produced using the VESTA software[70].

## Reporting summary

Further information on research design is available in the Nature Portfolio Reporting Summary linked to this article.

## Data availability

All relevant data that support the findings of this study are available in the main text and/or the Supplementary Information.

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

## Acknowledgements

H.T., P.T., and Y.W. acknowledged the support of the National Natural Science Foundation of China (grants 12474423 and 12074092), the Natural Science Foundation of Heilongjiang Province of China (grants ZD2022E003, YQ2022A010, and YQ2023A006), the China National Postdoctoral Program for Innovative Talents (grant BX20200111), and the Fundamental Research Funds for the Central Universities (2023FRFK06004). C.H. and X.M. acknowledged the support of the National Natural Science Foundation of China (grants 12374082 and 12104115). N.Z. acknowledged the support of the National Natural Science Foundation of China (grant 12161141012). F.L. acknowledged the support of the National Natural Science Foundation of China (grant 52325205). The authors thank the teams at the BL10U2 beamline of the Shanghai Synchrotron Radiation Facility for the beamtime.

## Author contributions

The work was conceived and designed by H.T., S.Z. and F.L.; P.T., X.H. and Y.W. prepared the samples and performed experiments, with assistance from D.L.; H.T., S.Z. and F.L. supervised the experiments; C.H. and X.M. grew the crystals; H.T. supervised crystal growth; J.Z., N.Z. and Q.W. performed synchrotron radiation experiments and data analysis, with assistance from X.W.; B.X., P.T. and X.X. performed the DFT calculations, with assistance from X.Z.; P.T. and X.H. drafted the manuscript. S.Z., H.T. and F.L. revised the manuscript, and all authors discussed the results.

## Competing interests

The authors declare no competing interests.

## Additional information

[1]School of Physics, Harbin Institute of Technology, Harbin, China. [2]Electronic Materials Research Laboratory, Key Laboratory of the Ministry of Education & International Center for Dielectric Research, School of Electronic Science and Engineering, Xi'an Jiaotong University, Xi'an, China. [3]School of Materials Science and Engineering, Harbin Institute of Technology, Harbin, China. [4]Laboratory for Space Environment and Physical Sciences, Harbin Institute of Technology, Harbin, China. [5]School of Chemistry and Chemical Engineering, Harbin Institute of Technology, Harbin, China. [6]Shanghai Synchrotron Radiation Facility, Shanghai Advanced Research Institute, Chinese Academy of Sciences, Shanghai, China. [7]Institute for Superconducting and Electronic Materials, Faculty of Engineering and Information Sciences, University of Wollongong, Wollongong, NSW, Australia. [8]These authors contributed equally: Peng Tan, Xiaolin Huang, Yu Wang. ✉e-mail: ful5@xjtu.edu.cn; shujun@uow.edu.au; tianhao@hit.edu.cn

