## [Transparent Peer Review file · Nature Communications]

Deciphering the Mechanism Underlying Highly Adjustable Piezoelectric Properties in Perovskite Ferroelectrics via Transition Metal Doping

Corresponding Author: Professor Hao Tian

Version 0:

Reviewer comments:

Reviewer #1

(Remarks to the Author)

Transition metal doping is a crucial way to optimize the properties of perovskite ferroelectrics. In particular, piezoelectricity, a fundamental property of perovskite ferroelectrics, exhibits high tunability in the case of different transition metal doping. However, due to the diversity of transition metals and the variability of their modification on the properties, a universal explanation of how transition metal doping affects piezoelectric properties has long remained lacking. In this work, the authors unravel the mechanism underlying highly adjustable piezoelectric properties in perovskite ferroelectrics via transition metal doping from a new perspective. The diverse effects of the typical transition metals Mn and Fe doped into the B-site as acceptor dopants are investigated through comprehensive experimental and theoretical approaches. The work reveals the relationship between the defect structure, the polarization framework, and the piezoelectric properties, demonstrating the important role of the electronic configuration of the defect structure in modifying the intrinsic piezoelectric properties. Moreover, the work has exploited the local structural heterogeneity enhanced by Mn doping and the polarization stability improved by Fe doping to achieve a piezoelectric coefficient d_{33} of up to 1000 pC/N and a mechanical quality factor Q_m of up to 700, respectively, which are at the top level of lead-free piezoelectric materials. I believe the work will provide a useful guide for the strategic design of perovskite ferroelectrics through transition metal doping and defect engineering. I do find the paper interesting and innovative, thus suitable for publication in Nature Communications. Only some minor revisions are suggested to be made.

1. The Q_m and the internal bias field E_{int} are the important parameters to exhibit the “hardening” or “softening” features after doping. In the main text, the authors give the Q_m and E_{int} of KTN, Fe-doped KTN, and Mn-doped KTN. The “hardening” features show that both Fe and Mn ions act as acceptor dopants. The measurement method for Q_m has been described in SI, but the description of E_{int} tests is missing and is suggested to be added.
2. The insets in Fig. 1(D-F) are unclear. Please replace the clearer photos or change the presentation to a clearer one.
3. In Figure 4C, the schematic shows that the dopants replace the central Nb ion, and the oxygen vacancy is placed to the nearest neighbor position in the $[001]_c$ direction. What is the reason for constructing the lattice this way? And, is this 4×4 supercell applicable enough for defect structures? In addition, please describe the statistical range in Figure 4H.
4. The acceptor impurities at the B-site and the oxygen vacancies form the defect dipoles, which are usually considered to have a pinning effect on the domain walls and limit the piezoelectric activity. For the Mn-doped case, the authors suggest a variety of reasons for the enhanced piezoelectricity. What do the authors think to be the most important reason?
5. In Figure 5(D-H), the authors study the polarization dynamics and field-induced phase transition by using the Raman spectra. There is the third fitted peak called “Second order mode” in Figure 5(E, F). What is the origin of the peak? Because it is not shown in DFT simulations, does it influence the results of the analysis?

Reviewer #2

(Remarks to the Author)

The authors have studied the structural, dielectric, and piezoelectric properties of $KTa_{1-x}Nb_xO_3$ (KTN) perovskite oxide that they have doped with Fe or Mn atoms. The focus of the work is on investigation of the atomic-level mechanisms of doping effects on the piezoelectricity. The KTN material itself is one of the lead-free piezoelectric materials that has been

investigated actively during the past 10 years.

Some of the authors have recently (May 2024) published a related paper: Huang et al. Impact of defect concentration on piezoelectricity in Mn/Fe-doped KTN crystals, *Appl. Phys. Lett.* 124, 192906 (2024) (<https://doi.org/10.1063/5.0206593>). In the paper they show how Mn-doped KTN with 0.1 and 0.5 mol% Mn increases the piezoelectric strain coefficient d_{33} rather significantly ($d_{33} = \sim 600$ pC/N for 0.5 mol% Mn). However, in the present manuscript they do not seem to cite this previous work and they do not explain, how Mn relatively similar dopant content of 0.55 mol% leads here in d_{33} of 1020 pC/N. Some of the authors have also discussed the Mn-doping of KTN in "Manganese-doping enhanced local heterogeneity and piezoelectric properties in potassium tantalate niobate single crystals" Wang et al. *IUCrJ* Volume 8| Part 2| March 2021| Pages 319-326 (<https://doi.org/10.1107/S2052252521000890>). There, Mn-doped KTN was predicted to possess twice as good piezoelectric properties as KTN. To summarize, the benefits of Mn-doping of KTN as such are not a new finding.

The authors present HAADF images and synchrotron data that illustrate some of the atomic-level differences for Mn- and Fe-doped KTN compared to pristine KTN. The findings are explained with help of density functional theory (DFT) calculations. The DFT calculations have serious limitations. The authors have employed a simple defect model where the oxygen vacancy defect is always in the same octahedron as the Mn/Fe dopant. It remains to be shown that this is really the lowest-energy configuration, as the vacancy could be also further away in an Nb or Ta octahedron. The authors used DFT-PBE functional, which is not that reliable for d-metal oxides (especially when considering band structure/DOS, which the authors use in their arguments). HSE06 hybrid functional is expected to yield more reliable energetics and band structures. Furthermore, the authors do not in any way comment spin polarization in their calculations. With PBE, the even the magnetic ground states of the systems can be incorrect.

To summarize, I think the main novelty of the paper is the new experimental characterization by microscopy and perhaps synchrotron XRD. The computational results and discussion based on them are not as rigorous. Improved piezoelectricity of Mn-doped KTN has been reported before by the authors. With a more rigorous and systematic computational part, the work could be publishable somewhere else.

Some further technical comments:

- (1) What is the uncertainty (standard deviation) of the compositions reported in Table S1?
- (2) How does the doping affect the lattice parameters of KTN (in XRD)
- (3) What is the second-order Raman mode discussed in Figure 5? Can the authors show that this is really a second-order mode and not a normal Raman mode? The full Raman spectrum should be compared between theory and experiment.
- (4) Raman spectra were simulated with Quantum Espresso. What pseudopotentials were used in these calculations? Why were supercells used in Raman spectrum calculations, this complicates the interpretation of vibrational modes?

Reviewer #3

(Remarks to the Author)

Piezoelectricity endows the perovskite ferroelectrics at the heart of electromechanical systems spanning from macro to micro/nano scales. Defect engineering strategies based on transition metal doping offer great potential for adjusting piezoelectric performance.

It is crucial to explore the key roles of transition metal impurities and derivative defect dipoles on piezoelectric properties. This study investigates how the transition metal impurity shapes piezoelectric properties, taking the Fe-doped and Mn-doped KTN as paradigms. The work provides insights into the micro-meso structure and macroscopic performances to elucidate the routes to control the piezoelectric properties via transition metal doping. The comprehensive understanding of the impacts of transition metals on the piezoelectric properties, as observed in KTN crystals, lays an essential basis for the design of perovskite ferroelectrics. The manuscript is innovative and well-organized. Hence, the work deserves publication in *Nature Communications* after addressing the following points with minor revisions.

1. It is well known that the strain-hysteresis performance of piezoelectric materials is of great significance for realizing high-speed and accurate signal control. In the manuscript, KTN-based single crystals show excellent low-strain-hysteresis characteristics. Please add relevant data to evaluate the strain-hysteresis characteristics.
2. It seems that the "Figure 4I" in line 342 should be corrected to "Figure 4E". Please check throughout the manuscript and supplementary information for accurate descriptions.
3. In line 589, in order to construct the supercell, the manuscript provided the values of lattice constants, while the XRD diffraction data was not given. Please give the source of data.
4. In figures S6 (a3 and a4), why is the overall direction of the curved domain walls in the Mn-doped KTN different from that of the domain walls in the pristine KTN?

Version 1:

Reviewer comments:

Reviewer #1

(Remarks to the Author)

The revised manuscript is ready for publication.

Reviewer #2

(Remarks to the Author)

The authors have improved the manuscript by seriously considering comments from all three referees. With respect to my original comments, the most significant change was using hybrid HSE exchange-correlation functional for investigating the electronic structure. For structural optimizations, using PBE is sufficient in this case. But for electronic structure studies, PBE can be even qualitatively wrong in the case of strongly correlated oxides with open d-shells. The use of HSE has made work more rigorous. My remaining few comments are as follows:

1. I don't really see the point of using ELF to describe the electronic structure. ELF values of less than $\frac{1}{2}$ mean electron localization that is less than in homogenous electron gas. So, discussing ELF features with values less than $\frac{1}{2}$ does not make sense. Charge density differences are a more rigorous approach.
2. The authors now mention that they have used spin polarization, but they do not give more detailed description of the spin states that they end up using. This is needed for reproducibility of the work. For the systems illustrated in Figure 4, the authors need to describe the spin state (magnetic moment of different atoms, overall magnetic moment of the system). This can be done in the Supporting information.
3. Related to point (2), CIF or POSCAR files of the used DFT models need to be given as Supporting information. This is crucial for reproducibility of the computational work.

After considering the above points, which are minor, the work is publishable.

Reviewer #3

(Remarks to the Author)

The author has solved all the problems. This is meaningful work.

Version 2:

Reviewer comments:

Reviewer #2

(Remarks to the Author)

The authors have revised the manuscript according to reviewer suggestions and the manuscript can be accepted for publication.

(For the record, I still think that the ELF plots are not a valuable contribution to otherwise rigorous paper. The literature added by the authors does not include any rigorous explanation of ELF values smaller than 0.5. Instead, just similar plots as in the current paper. The authors state in the rebuttal that "For example, in BiFeO₃ and GaFeO₃ materials, the finite ELF values of about 0.3~0.4 between A-site atoms and oxygen atoms can indicate the hybridization interaction and some degree of covalent characteristics" (2,3). Here, "hybridization interaction" is an ill-defined term with no meaning and the claim about covalent characteristics has no foundation. I suggest that the authors go back to the fundamental papers regarding ELF, written before people started misusing and misinterpreting ELF values smaller than 0.5. In his 1998 review, Burdett writes: "Thus it seems that ELF values below ELF = 0.5 can be interpreted as representing regions where either there is **very little electron density**, such as between atomic shells, or in regions where the contribution of a great number of nodes to ELF far outweighs the contribution of the density." (Electron Localization in Molecules and Solids: The Meaning of ELF, J. Phys. Chem. A 1998, 102, 31, 6366–6372). This statement is based on rigorous analysis, which is typically lacking from ELF papers using ELF scale [0..1].)

Response to Reviewers' comments and the description of revisions in the revised manuscript and supplementary information

We sincerely thank the reviewers for their comments and suggestions that are essential for the improvement of our work. We have revised our manuscript accordingly and the point-by-point responses to the comments are enclosed for your reference. We hope all the concerns and questions raised by the reviewers have been fully addressed and the manuscript is now satisfactory for the consideration of publication in Nature Communications.

[Reviewers' comments are in black; Author responses are in blue; Revisions in the manuscript are highlighted.]

Response to Reviewer #1

Transition metal doping is a crucial way to optimize the properties of perovskite ferroelectrics. In particular, piezoelectricity, a fundamental property of perovskite ferroelectrics, exhibits high tunability in the case of different transition metal doping. However, due to the diversity of transition metals and the variability of their modification on the properties, a universal explanation of how transition metal doping affects piezoelectric properties has long remained lacking. In this work, the authors unravel the mechanism underlying highly adjustable piezoelectric properties in perovskite ferroelectrics via transition metal doping from a new perspective. The diverse effects of the typical transition metals Mn and Fe doped into the B-site as acceptor dopants are investigated through comprehensive experimental and theoretical approaches. The work reveals the relationship between the defect structure, the polarization framework, and the piezoelectric properties, demonstrating the important role of the electronic configuration of the defect structure in modifying the intrinsic piezoelectric properties. Moreover, the work has exploited the local structural heterogeneity enhanced by Mn doping and the polarization stability improved by Fe doping to achieve a piezoelectric coefficient d_{33} of up to 1000 pC/N and a mechanical quality factor Q_m of up to 700, respectively, which are at the top level of lead-free

piezoelectric materials. I believe the work will provide a useful guide for the strategic design of perovskite ferroelectrics through transition metal doping and defect engineering. I do find the paper interesting and innovative, thus suitable for publication in Nature Communications. Only some minor revisions are suggested to be made.

RESPONSE: We sincerely appreciate the reviewer's valuable comments, and thanks for the high recognition of our work. We have revised the manuscript according to the reviewer's suggestions. We also added the responses and revisions in the following.

1. The Q_m and the internal bias field E_{int} are the important parameters to exhibit the "hardening" or "Softening" features after doping. In the main text, the authors give the Q_m and E_{int} of KTN, Fe-doped KTN, and Mn-doped KTN. The "hardening" features show that both Fe and Mn ions act as acceptor dopants. The measurement method for Q_m has been described in SI, but the description of E_{int} tests is missing and is suggested to be added.

RESPONSE: Thanks for the valuable suggestion. We characterized the E_{int} by the following method. Because E_{int} is the source of the asymmetry of P - E loops, the magnitude of E_{int} can be determined by

$$E_{int} = \frac{E_+ + E_-}{2}$$

where E_+ and E_- are the intersections of polarization loop with positive and negative electric field axis, respectively. Hence, the E_{int} can be obtained by extracting E_+ and E_- from the P - E loops. This characterization method of E_{int} is widely accepted in perovskite ferroelectric study¹⁻⁴.

According to this suggestion, we have added the characterization method for E_{int} on Line 187 of the revised manuscript as follows: "The magnitude of E_{int} is determined by $E_{int} = (E_+ + E_-)/2$, where E_+ and E_- are the intersections of polarization loop with positive and negative electric field axis."

2. The insets in Fig. 1(D-F) are unclear. Please replace the clearer photos or change the presentation to a clearer one.

RESPONSE: Thanks for the valuable suggestion. The insets in Fig. 1(D-F) represent the piezoelectric charge coefficients measured by the Berlincourt type d_{33} meter. To achieve a better visualization, we have replaced the insets in Fig. 1(D-F) with high resolution photographs in the revised manuscript. In the revised version, the clarity, contrast, and consistency of the style of the photos are all improved, as shown in Figure R1.

Figure R1. A part of Figure 1 in the revised manuscript. The insets in Figures 1(D-F) have been replaced by high resolution photographs.

3. In Figure 4C, the schematic shows that the dopants replace the central Nb ion, and the oxygen vacancy is placed to the nearest neighbor position in the $[001]_c$ direction. What is the reason for constructing the lattice this way? And, is this $4*4*4$ supercell applicable enough for defect structures? In addition, please describe the statistical range in Figure 4H.

RESPONSE: Thanks very much for the valuable feedback. To reveal the impact of defect dipoles on the local polarization at a lattice scale, we have performed the DFT calculations. Regarding the comments on the computational model, we address them below, please kindly check.

(Q1) What is the reason for constructing the lattice this way?

The lattice was constructed with two key considerations in mind: the strong binding effect of the acceptor on the oxygen vacancy $V_{\text{O}}^{\bullet\bullet}$, and the preference for the lowest formation energy.

The wide electron paramagnetic resonance (EPR) peaks of the Fe-doped and Mn-doped samples at 166 and 174 mT, respectively, are attributed to the presence of the

$(\text{Fe}_B'' - \text{V}_O'')$ and $(\text{Mn}_B' - \text{V}_O'')$ defect dipoles^{5,6}. This demonstrates a strong coupling between the acceptor dopants and V_O'' . The V_O'' , tend to form around acceptors, especially at the nearest neighboring oxygen sites, which promotes defect dipole formation. Therefore, we position the V_O'' adjacent to the dopants in our model.

Regarding whether the acceptor dopants substitute Ta or Nb ions, and the specific adjacent oxygen site where the V_O'' occurs, we used the principle of the lowest formation energy for the selection. To identify the lowest-energy configuration, the formation energies E_{form} of the defect systems were estimated using the equation of

$$E_{\text{form}} = E_{\text{bulk}'} + E_{\text{O}} - E_{\text{bulk}} + (E_{\text{B-site}} - E_{\text{d}})$$

where E_{bulk} and $E_{\text{bulk}'}$ are the total energies of the pristine and doped KTN supercells, respectively, $E_{\text{B-site}}$ and E_{d} are the energies of the original B-site atom and doped atom in the bulk phase, respectively, and E_{O} is half of the energy of an isolated O₂ molecule⁷. As Figure S7 in the supplementary information depicted, we calculated the formation energies of the defect structures corresponding to all possible occurrences of dopants and V_O'' , where O₁–O₆ represent the oxygen sites adjacent to the dopant ions. Consistent with the principle of the lowest formation energy, the dopant ions tend to replace the Nb ions and the V_O'' tend to form at the O₆ site, the oxygen site closest to the central B-site ion. The result agrees well with the widely reported occupation of oxygen vacancies in perovskite ferroelectrics⁸⁻¹⁰.

Based on the results above, we constructed the defect structure models by substituting Nb ions with Fe or Mn dopants, coupled with the formation of a V_O'' at the O₆ site, for the simulations of structural parameters and electronic configurations.

(Q2) Is this 4*4*4 supercell applicable enough for defect structures? In addition, please describe the statistical range in Figure 4H.

In actual crystals, such dopant and defect concentrations are generally very low. In order to match the actual situation, a large supercell is conducive to obtaining accurate conclusions. However, considering the cost of DFT calculations, researchers

are often unable to construct huge supercell models according to the actual dopant and defect concentrations. Instead, smaller supercells are likely to be chosen to simulate the local structure around dopants or defects and to present the simulation results qualitatively or semi-quantitatively¹¹. In DFT simulations of perovskite ferroelectrics, the $2 \times 2 \times 2$ 40-atom supercells have been proven to be large enough to capture the local structural characteristics^{12,13}. Empirically, the $2 \times 2 \times 2$ supercells can effectively simulate defect structures and investigate the role of dopants and defects, as demonstrated in studies on PMN-PT reported in *Science* and hybrid perovskite materials reported in *Nature Photonics*^{13,14}. Hence, using a larger $4 \times 4 \times 4$ supercell for DFT calculations is feasible. Recently, Tian et al. investigated the charge density of $(\text{K}_{0.5}\text{Na}_{0.5})\text{NbO}_3$ with a defect dipole by a $4 \times 4 \times 4$ supercell, confirming that the model is suitable for the defect system [*Sci. Adv.* **10**, eadn2829 (2024)]¹⁵.

The choice of larger $4 \times 4 \times 4$ supercells in this work has two further important reasons: (1) Firstly, in order to avoid the interactions between adjacent defects to simulate the low concentration of defects in actual situations, we need to ensure that the defects have a sufficiently large spacing. Considering the application of periodic boundary conditions in the DFT calculations, this implies that the supercell containing only one defective site needs to be large enough. When using a $4 \times 4 \times 4$ supercell, the spacing between the periodic defective sites approaches 15 Å, which achieves an empirically enough spacing to prevent the interactions between periodic defect structures^{16,17}. The dielectric screening in supercells of this scale is also sufficiently large to eliminate the electrostatic interaction between a defect and its periodically repeated images¹⁸. Thus, the $4 \times 4 \times 4$ supercell allows for the unveiling of the localized effect of the defect structure.

(2) Secondly, in order to compare with the experimental results in Figure 2(D-F), we intended to depict the distribution of local polarization orientations from the lattice perspective, which is presented in Figure 4C. Therefore, a sufficient amount of oxygen octahedra in the supercell is required to ensure the validity of the statistics about the local polarization orientation. The statistics are obviously difficult to accomplish within

a $2 \times 2 \times 2$ supercell. Given the computational cost, we chose the $4 \times 4 \times 4$ supercell.

According to the suggestion, we have added a description of the statistical range of Figure 4C in the revised manuscript. We obtained the vector in oxygen octahedron pointing from the negatively charged center to the positively charged center using the simulated structural parameters. This vector indicates the recognized orientation of the local polarization in the oxygen octahedron. We then analyzed the polarization orientations in oxygen octahedra within the $(100)_C$ section containing defect dipoles to represent the distribution of these polarization orientations. The revised description has been given on Line 337, as follows: "Box charts of deviation angles of dipole moments from the $[011]_C$ direction in the octahedra within the $(100)_C$ sections containing the defect dipoles, statistically obtained from Figure S8". Additionally, Figure S8 has been added to the revised supplementary information.

4. The acceptor impurities at the B-site and the oxygen vacancies form the defect dipoles, which are usually considered to have a pinning effect on the domain walls and limit the piezoelectric activity. For the Mn-doped case, the authors suggest a variety of reasons for the enhanced piezoelectricity. What do the authors think to be the most important reason?

RESPONSE: Thanks for the valuable question.

This study demonstrates that Mn doping significantly enhances piezoelectricity, elevating the d_{33} value to above 1000 pC/N—more than twice that of pristine KTN crystals. Contrary to the typical pinning effect caused by acceptor heterovalent dopants at the B-site and the oxygen vacancies, which limits domain wall motions and reduces piezoelectric activity, the greatly enhanced piezoelectricity after Mn doping implies an alternative mechanism at play, specifically related to the intrinsic polarization behaviors studied here.

To explore this, we investigated the local polarization distributions and polarization dynamics in pristine KTN, Fe-doped, and Mn-doped KTN crystals. Our work demonstrates that Mn doping introduces disordered polarization orientations and

strong local distortions. This increased local structural heterogeneity tends to induce interfacial energies encompassing electrostatic, elastic, and gradient components¹⁹. The interplay between the bulk Landau energy and interfacial energy can lead to diverse free-energy landscapes with varying levels of disorder. In structures exhibiting a greater level of disorder, the free energy profile tends to be more flattened, thus promoting the polarization rotation. *In summary, the local structural heterogeneity strengthened by Mn dopants enhances the level of disorder in Mn-doped KTN, further promoting the polarization rotation, being responsible for the increased piezoelectric activity.*

5. In Figure 5(D-H), the authors study the polarization dynamics and field-induced phase transition by using the Raman spectra. There is the third fitted peak called “Second order mode” in Figure 5(E, F). What is the origin of the peak? Because it is not shown in DFT simulations, does it influence the results of the analysis?

RESPONSE: Thanks for the valuable comments.

The third fitted peak at $\sim 590\text{ cm}^{-1}$, associated with a second-order mode in KTN crystals, has been demonstrated to stem from the combination of the transverse $A_1(3TO)$ mode and the acoustic TA mode at the critical X -point of the Brillouin zone^{20,21}. The Raman spectra calculated by DFPT do not include the second-order Raman effect. Hence, the second-order Raman peaks do not appear in the calculation results. Given that the second-order Raman scattering is a two-phonon process, we obtained the Raman shift of the mode by calculating the phonon spectra. The phonon spectra of the optical $A_1(3TO)$ mode and the acoustic TA mode are shown in Figure R2. For the orthorhombic structure, the wavenumbers of the $A_1(3TO)$ and TA modes at the X point of the Brillouin zone are 507 and 90 cm^{-1} , respectively. For the tetragonal structure, the wavenumbers of the $A_1(3TO)$ and TA modes at the X point of the Brillouin zone are 507 and 97 cm^{-1} , respectively. The sum of the two phonons induces the generation of the second-order Raman mode with a Raman shift of $\sim 600\text{ cm}^{-1}$, which is in good agreement with the $\sim 590\text{ cm}^{-1}$ peak measured in the experiment. This alignment strongly supports the identification of the third fitted peak as a second-order Raman

mode.

As shown in Figure 5 in the main text, the second-order Raman peak, the neighboring $A_1(3TO)$ and $B_1(3TO)$ peaks can be clearly identified in the range of 400-700 cm^{-1} , respectively. This allows us to obtain the individual Raman peaks by fitting. Therefore, when using the $A_1(3TO)$ and $B_1(3TO)$ modes to investigate the field-induced phase transition process, the second-order Raman peaks do not influence the results of the analysis.

According to the reviewer's comment, we have added the description of the origin of the second-order Raman peak in the revised manuscript on Line 451, as follows:

“The third fitted peak at $\sim 590 \text{ cm}^{-1}$, associated with a second-order mode in KTN crystals, stems from the combination of the transverse $A_1(3TO)$ mode and the acoustic TA mode at the critical X -point of the Brillouin zone^{50,51}”, and the following Figure R2 and its relevant description have been added to the revised supplementary information, as follows:

Figure R2 (Figure S11 in the supplementary information). The phonon spectra of the optical $A_1(3TO)$ mode and the acoustic TA mode in (A) orthorhombic KTN and (B) tetragonal KTN, respectively. In the orthorhombic KTN, the wavenumbers of the $A_1(3TO)$ and TA modes at the X point of the Brillouin zone are 507 and 90 cm^{-1} , respectively. In the tetragonal KTN, the wavenumbers of the $A_1(3TO)$ and TA modes at the X point of the Brillouin zone are 507 and 97 cm^{-1} , respectively. The sum of the two phonons induces the generation of the second-order Raman mode with a Raman shift of $\sim 600 \text{ cm}^{-1}$, which is in good agreement with the $\sim 590 \text{ cm}^{-1}$ peak measured in the

experiment. This alignment strongly supports the identification of the third fitted peak as a second-order Raman mode.

New references [50,51] have been cited in the revised manuscript:

- 50 Kugel, G. E., Mesli, H., Fontana, M. D. & Rytz, D. Experimental and theoretical study of the Raman spectrum in $\text{KTa}_{1-x}\text{Nb}_x\text{O}_3$ solid solutions. *Phys. Rev. B* **37**, 5619-5628 (1988).
- 51 Manlief, S. K. & Fan, H. Y. Raman Spectrum of $\text{KTa}_{0.64}\text{Nb}_{0.36}\text{O}_3$. *Phys. Rev. B* **5**, 4046-4060 (1972).

Response to Reviewer #2

The authors have studied the structural, dielectric, and piezoelectric properties of $\text{KTa}_{1-x}\text{Nb}_x\text{O}_3$ (KTN) perovskite oxide that they have doped with Fe or Mn atoms. The focus of the work is on investigation of the atomic-level mechanisms of doping effects on the piezoelectricity. The KTN material itself is one of the lead-free piezoelectric materials that has been investigated actively during the past 10 years. Some of the authors have recently (May 2024) published a related paper: Huang et al. Impact of defect concentration on piezoelectricity in Mn/Fe-doped KTN crystals, *Appl. Phys. Lett.* 124, 192906 (2024) (<https://doi.org/10.1063/5.0206593>). In the paper they show how Mn-doped KTN with 0.1 and 0.5 mol% Mn increases the piezoelectric strain coefficient d_{33} rather significantly ($d_{33} = \sim 600$ pC/N for 0.5 mol% Mn). However, in the present manuscript they do not seem to cite this previous work and they do not explain, how Mn relatively similar dopant content of 0.55 mol% leads here in d_{33} of 1020 pC/N. Some of the authors have also discussed the Mn-doping of KTN in “Manganese-doping enhanced local heterogeneity and piezoelectric properties in potassium tantalate niobate single crystals” Wang et al. *IUCrJ* Volume 8, Part 2, March 2021, Pages 319-326 (<https://doi.org/10.1107/S2052252521000890>). There, Mn-doped KTN was predicted to possess twice as good piezoelectric properties as KTN. To summarize, the benefits of Mn-doping of KTN as such are not a new finding. The authors present HAADF images and synchrotron data that illustrate some of the atomic-level differences for Mn- and Fe-doped KTN compared to pristine KTN. The findings are explained with help of density functional theory (DFT) calculations. The DFT calculations have serious limitations. The authors have employed a simple defect model where the oxygen vacancy defect is always in the same octahedron as the Mn/Fe dopant. It remains to be shown that this is really the lowest-energy configuration, as the vacancy could be also further away in an Nb or Ta octahedron. The authors used DFT-PBE functional, which is not that reliable for d-metal oxides (especially when considering band structure/DOS, which the authors use in their arguments). HSE06 hybrid functional is expected to yield more reliable energetics and band structures.

Furthermore, the authors do not in any way comment spin polarization in their calculations. With PBE, the even the magnetic ground states of the systems can be incorrect. To summarize, I think the main novelty of the paper is the new experimental characterization by microscopy and perhaps synchrotron XRD. The computational results and discussion based on them are not as rigorous. Improved piezoelectricity of Mn-doped KTN has been reported before by the authors. With a more rigorous and systematic computational part, the work could be publishable somewhere else.

RESPONSE: We appreciate the reviewer for their knowledgeable feedback. The reviewer raised two main concerns: the difference between the present work and previous reports, and the reliability of the DFT calculations used in the work. We will address each of these points in turn.

(Q1) Novelty of this work and differences from previous reports

The modification of piezoelectricity in perovskite ferroelectrics by Fe and Mn doping has been studied extensively. Previous research has highlighted the different roles of Fe and Mn doping in PZT ceramics. For example, PZT-4, predominantly doped with Mn, exhibits a piezoelectric coefficient d_{33} of approximately 370 pC/N and a mechanical quality factor Q_m of less than 500²². In contrast, PZT-8, mainly doped with Fe, has an inferior d_{33} of less than 280 pC/N but a high Q_m of nearly 1,000²³. *Similar observations have been made in lead-free piezoelectric materials, such as tetragonal KTN crystals, as noted in the study published in Appl. Phys. Lett. that the reviewer refers to.*²⁴

However, the differences observed with Fe and Mn doping in those studies are limited, and more importantly, all of the studies attributed the difference to the defect-modulated domain wall motion behavior, ignoring the contribution of intrinsic polarization behavior. Our present work, however, focused on revealing the role of intrinsic polarization behavior in piezoelectric properties and how Fe and Mn doping - the most actively studied transition metal dopant elements - modify the local polarization, providing a general rule in designing high performance ferroelectric materials using transition metal dopants. Therefore,

this study is fundamentally different from previous research²⁴, though the studied KTN system is the same.

We discovered significant differences in the piezoelectric properties of orthorhombic KTN crystals doped with Fe and Mn. Compared to pristine KTN, which has a d_{33} of 420 pC/N and a Q_m of 120, Mn doping led to a substantial increase in d_{33} , raising it to above 1000 pC/N—double that of the undoped material. In contrast, Fe doping resulted in a substantial hardening effect, with the Q_m value increasing fivefold compared to pristine KTN. Additionally, the S - E curves for all samples, as shown in Figures 1(D-F) of the manuscript, display minimal hysteresis below 5%. Typically, low strain hysteresis suggests a minimal domain wall motion contribution to the piezoelectric properties, suggesting that these properties are primarily determined by intrinsic polarization behavior.

This observation highlights intrinsic polarization as a key factor behind the differing properties resulting from Fe and Mn doping. As noted by the reviewer, we have employed HAADF, synchrotron XRD, and other techniques to investigate the micro- and mesoscopic polarization structures, aiming to uncover the correlation between defect structure, polarization behavior, and macroscopic piezoelectric properties in KTN crystals doped with Fe and Mn. Unlike the tetragonal KTN crystals examined in previous studies published in Appl. Phys. Lett. and IUCrJ^{24,25}, this work focused on KTN crystals in the orthorhombic phase. As illustrated in Figure R3, in tetragonal KTN, the orientation of the defect dipole aligns with one of the polarization directions. According to the symmetry-conforming principle, the defect dipole's pinning effect tends to anchor the polarization, significantly limiting polarization rotation, thereby making it difficult to observe the contribution of intrinsic polarization behavior. In contrast, in orthorhombic KTN, the polarization and defect dipole orientations differ, allowing for easier polarization rotation under an applied electric field. This makes KTN crystals with orthorhombic phase particularly suitable for a more thorough investigation of intrinsic polarization behavior. The ability to control the phase boundaries of KTN crystals further motivates our study of the intrinsic

polarization behavior.

Figure R3. Orientations of spontaneous polarization and defect polarization in the tetragonal crystal and orthorhombic crystal, respectively. Here, a typical ABO_3 perovskite structure is considered, which contains an oxygen vacancy. For simplicity, only the (100) plane of the structure is shown and thus the two O^{2-} sites above/below the B-site ion are not shown; also, A-site cations are omitted.

In summary, this work demonstrates the ability to achieve highly tunable piezoelectric properties in orthorhombic KTN crystals through Fe and Mn doping, while uncovering how these dopants modify piezoelectric properties by influencing intrinsic polarization behavior. Unlike the conventional explanation that focuses on domain wall motion, i.e., an extrinsic contribution which has been generally accepted in the last several decades, this study offers new insights into the mechanisms behind the adjustable piezoelectric properties in perovskite ferroelectrics achieved through transition metal doping.

In the revised manuscript, to highlight the significance of this study, the abstract has been modified as follows: “... In this work, we selected perovskite $KTa_{1-x}Nb_xO_3$ (KTN) single crystals with orthorhombic phase as the matrix and introduced Fe and Mn elements, which are commonly used in “hard” ferroelectrics as dopants. We investigated how transition-metal doping modifies piezoelectric properties from the perspective of intrinsic polarization behaviors. ...”.

Additionally, we have expanded the introduction to comment on previous studies on KTN and explain the rationale for the choice of orthorhombic KTN crystals, as follows: “This challenges the sole explanation based on domain wall motion and

emphasizes the importance of considering the impact of transition metals on intrinsic polarization.”, “ $\text{KTa}_{1-x}\text{Nb}_x\text{O}_3$ (KTN) solid solution single crystal, a key member of lead-free ferroelectric materials, possesses a simple lattice, ease of doping, controllable phase structure, and a high degree of tunability in P_s due to its infinite miscibility of KNbO_3 and KTaO_3 ^{16,17}. Similar effects to those of Fe and Mn doping in PZT ceramics have been observed in tetragonal KTN single crystals¹⁸. This enables KTN a promising medium for studying transition metal doping. However, in tetragonal KTN, the alignment of spontaneous and defect polarizations (P_s and P_D) results in a pinning effect that binds the intrinsic polarization due to the symmetry-conforming principle¹⁹. Consequently, defect-modulated domain wall motion often dominates, making it difficult to fully observe the contribution of intrinsic polarization behavior. In contrast, the different orientations of P_s and P_D in orthorhombic phase facilitate polarization rotation. Hence, this work explored orthorhombic KTN single crystals as the matrix and incorporated Fe and Mn as dopants to investigate how transition metal impurity influences intrinsic polarization behaviors and piezoelectric properties.”

The references [18,67] have been cited in the revised manuscript:

18 Huang, X. et al. Impact of defect concentration on piezoelectricity in Mn/Fe-doped KTN crystals. *Applied Physics Letters* 124, 192906 (2024).

67 Wang, Y. et al. Manganese-doping enhanced local heterogeneity and piezoelectric properties in potassium tantalate niobate single crystals. *IUCrJ* **8**, 319-326, (2021).

(Q2) Clarification and improvement of the DFT calculation method

We appreciate the reviewer’s insightful comments regarding the DFT calculations. In our study, DFT calculations were performed to explore the differences in defect structures from the point of view of electronic configuration. Based on the valuable comments, we have optimized the DFT calculations. In the following, we first explain the rationale behind the original method, and then describe the modified computational scheme.

a. Clarification of the DFT calculation in the original version

The original manuscript employed the PBE functional in DFT calculations for two

main reasons:

1. The primary goal of the DFT calculations was to qualitatively analyze the electronic structures of defected materials. This approach, in conjunction with experimental results, helps clarify the features of local polarization in pristine KTN, Fe-doped KTN, and Mn-doped KTN crystals. Numerous studies have demonstrated that the PBE functional provides sufficiently accurate structural parameters²⁶⁻²⁸. Although it may be less precise than the HSE hybrid functional regarding formation energy, band structure, and polarization, the PBE functional effectively captures the relative relationships and trends of these parameters²⁹⁻³². Even for band gap evaluations, where errors might be significant, the PBE functional accurately reflects the relative magnitude of the band gap, especially when it exceeds 0.76 eV³⁰. Therefore, the PBE functional was used to qualitatively reveal doping-induced changes, aiding in our analysis.
2. To calculate the defect structure and manifest its influences on the local polarization, we constructed a $4 \times 4 \times 4$ supercell containing more than 300 atoms, including elements with large atomic numbers, such as Ta, Nb, Fe, Mn, and K. Evidently, it is a large system for DFT calculations. Unfortunately, HSE calculations are 2 orders of magnitude more computationally demanding than those with PBE functional, making their use on a large-scale impractical²⁹. In contrast, PBE functional shows high feasibility in the simulations of large system structures and has been repeatedly reported for structural optimization and electronic structure calculations involving transition metal³³⁻³⁵. Recently, Tian et al. used PBE functional to investigate the charge density of $(\text{K}_{0.5}\text{Na}_{0.5})\text{NbO}_3$ with a defect dipole by a $4 \times 4 \times 4$ supercell, confirming that the PBE is effective for qualitative physical observations [*Sci. Adv.* **10**, eadn2829 (2024)]¹⁵.

For the above reasons, the PBE functional was employed in the original manuscript to qualitatively characterize the defect structure.

b. The modified computational scheme

Given the limitations of the PBE functional for electronic structure calculations

involving strongly correlated materials with transition-metal, lanthanide, or actinide elements with partially filled *d*- or *f*-shells³⁶, according to the reviewer's valuable comment, we have adopted a more refined calculation scheme. Usually, to achieve higher computational accuracy, DFT calculations can be enhanced by PBE+U and HSE methods. However, the PBE+U method often requires system-specific U values, which can be challenging to determine accurately and may introduce errors if the values are not properly selected. Consequently, we did not use this method to avoid potential inaccuracies due to inappropriate U values. The HSE calculation suggested by the reviewer is indeed a very good way to improve the calculation accuracy. However, given the large system size and the need to simulate multiple occupancy cases in this work, the computational cost of HSE calculations is prohibitively high. Hence, we aim to employ a new computational strategy that balances higher accuracy with computational efficiency.

Recent research has provided valuable insights into our approach. Liu et al. employed high-throughput computational techniques to assess the accuracy of PBE and HSE methods across over 1,200 solid-state materials²⁹. They found a strong linear correlation between formation energies calculated using PBE and HSE, with both methods showing similar mean absolute errors (MAE) for transition-metal compounds. Additionally, numerous studies have demonstrated that PBE and HSE both predict structural parameters in good agreement with experimental values²⁶⁻²⁸. While PBE offers a cost-effective means of structural prediction, HSE's complex and time-consuming nature provides little additional improvement in structural accuracy over PBE²⁷. Based on these findings, we have adopted the following computational procedure:

1. Given the strong correlation between formation energies calculated by PBE and HSE, and the satisfactory accuracy of PBE for structure optimization, we first use the PBE functional for optimizing the structures of both pristine and defective KTN supercells. We then calculate the formation energies for defect structures where

dopant ions and oxygen vacancies occupy various sites to identify those with the lowest formation energies.

2. Next, we use the HSE functional to calculate the electronic structures of the identified structures with the minimum formation energies. This step also includes determining the differential charge density and electron localization function. Using these refined electronic configurations, we analyze the impact of defect structures on local polarization.

This strategy leverages the efficiency of PBE for structural optimization and the accuracy of HSE for electronic structure calculations, making it well-suited for large-scale DFT simulations. A similar approach of first using PBE for structural optimization followed by HSE for electronic structure calculations has been reported for other materials^{37,38}.

c. Updated results of the DFT calculations in the revised manuscript

Using the computational strategy outlined above, we computed four structures, covering pristine KTN, KTN with oxygen vacancy, Fe-doped KTN, and Mn-doped KTN. This approach provided more accurate results for the electronic configurations. As depicted in Figure R4, while the new results are similar to the qualitative trends observed with the original PBE-only functional calculations, they give a more accurate description of electron localization features. We have updated the manuscript to include the new scheme.

In addition, regarding the spin polarization, we had covered it in the original DFT calculations, we apologize for not including a description of it in the method section.

Figure R4. The differential charge density of the simulated structures, covering pristine KTN, V_O'' -KTN, $(Fe_B'' - V_O'')^x$ -KTN, and $(Mn_B' - V_O'')^\bullet$ -KTN, performed by the (A) PBE and (B) HSE methods, respectively.

According to the reviewer’s feedback, we have revised the calculation methods section on Line 597 in the revised manuscript, as follows: “...Spin-polarization was fully considered in all calculations. During structural optimizations, the exchange-correlation energy was calculated in Perdew-Burke-Ernzerhof (PBE) functional.... To address the limitation of semi-local exchange-correlation PBE method⁵⁹, the nonlocal hybrid functional of Heyd, Scuseria, and Enzerhof (HSE06) was employed to accurately describe the electronic structure of the defect system⁶⁰. A single k-point was used, which is sufficiently accurate for treating such a large defect system⁶¹.”

The relevant descriptions of results and discussions have been optimized. The differential charge density, Bader charge analysis, and electron localization function were investigated to reveal the charge distribution of the defect structures. In addition, since only a single k-point is used in the HSE calculations, we therefore do not discuss the density of states in the revised manuscript. The section of the main text on the DFT calculations has been modified as follows:

“The introduction of defects into the lattice inevitably leads to a variation in the local structure, thereby affecting the inherent polarization. The change in charge

distribution is the primary source of local structural distortions. Figure 4(D) depicts the differential charge density (DCD) of the simulated structures, covering pristine KTN, $V_O^{\bullet\bullet}$ -KTN, $(Fe_B^{\prime\prime} - V_O^{\bullet\bullet})^{\times}$ -KTN, and $(Mn_B^{\prime} - V_O^{\bullet\bullet})^{\cdot}$ -KTN. Meanwhile, the Bader charges Q_B of the B-site cations and oxygen ion immediately adjacent to the vacancy site are labelled in Figure 4(D). In the pristine KTN lattice, electrons transfer from the B-site Nb or Ta ions to adjacent oxygen ions, forming a stable oxygen octahedral configuration. The heterovalent doping often creates $V_O^{\bullet\bullet}$. When only concerning the effect of $V_O^{\bullet\bullet}$, the $V_O^{\bullet\bullet}$ notably reduces electrons at the defect site, forming an equivalent positive charge center. This yields the local structural distortion and disrupts the pattern of charge transfer, as the DCD of $V_O^{\bullet\bullet}$ -KTN shown.

The doped Fe and Mn ions can compensate for the electron reduction in the defect structure due to their smaller Q_B than the original B-site ions. In the Fe-doped lattice, the decrease in Q_B at the doped site closely matches the increase in Q_B at the vacancy site, which benefits the local charge equilibrium and yields a stronger compensation effect. This enables the charge transfer case to be more similar to that of the pristine lattice, thereby reducing lattice distortion. Figure 4(E) shows the statistics of the distribution of dipole moments within the lattices. In the Fe-doped lattice, the average orientation of the dipole moments remains aligned along the $[011]_C$ direction, i.e., the uniform orientation of P_s in the pristine KTN model.

In terms of the Q_B of Fe and Mn ions (+1.45 e and +1.84 e, respectively), the Mn ion loses nearly $\frac{1}{4}$ more electrons than the Fe ion, consistent with the nature of positive trivalent Fe ion and positive tetravalent Mn ion. In the Mn-doped lattice, in addition to the weaker electron compensation effect than in the Fe-doped case, the smaller radius of Mn^{4+} ($r = 0.53 \text{ \AA}$) further expands the void caused by the vacancy. This is in contrast to the Fe-doped lattice, where the radius of Fe^{3+} ($r = 0.645 \text{ \AA}$) is nearly identical to that of the original B-site ions ($r = 0.64 \text{ \AA}$)⁴¹. The increased spacing of the Mn ion from the neighboring oxygen weakens the bonding, especially on the Mn-O bond located on the side away from the vacancy, as shown by the DCD of Mn-doped lattice in Figure 4(D).

These local differences enhance the lattice distortion within the Mn-doped lattice, undermining the order of local polarization.

Moreover, according to the nominal oxidation states of the ions, i.e., Fe^{3+} , Mn^{4+} , Ta^{5+} , Nb^{5+} , and O^{2-} , the Fe_B'' and Mn_B' defects exhibit -2 valence and -1 valence, respectively, while the stabilized V_O'' acquires a positive bivalence state after crystallization. Therefore, following the formation of defect dipoles, the charge balance is maintained in the Fe-doped system, whereas Mn doping results in a localized $+1$ valence net charge, leading to the presence of random fields. The random fields from Mn doping yield relaxor behavior, as witnessed by a pronounced frequency dispersion in dielectric behavior, along with the occurrence of dielectric relaxation observed in the low-temperature range of -150 to 0 °C, as depicted in Figure 1(C). Heterogeneous charge distributions largely impact local structures through random electric fields, thereby also exacerbating the disorder of local polarization¹⁰.

Defect dipoles formed by the V_O'' and B-site heterovalent ions largely influence the behaviors of ferroelectric polarization. Figure 4(F) depicts the electron localization function (ELF) representations for the pristine and defective lattices. The ELF of V_O'' -KTN agrees with the case of doubly charged oxygen vacancies⁴². In terms of the Bader charge, the V_O'' and doped ions form the equivalent positive charge center and equivalent negative center, respectively. This drives the enhancement of electrostatic interactions between V_O'' and dopant ions, changing the spatial representation of the electron distribution, as illustrated in ELF. The localized charge distribution around the defect dipole in the Fe-doped crystal is clearly more polarized, indicating the stronger defect polarization⁴³. The strong and electrical neutral defect dipoles tend to stabilize the polarization ordering. The enhanced stability endows the Fe-doped sample with a "hardening" characteristic, as reflected in the increased E_int and Q_m as shown in Figures 1(K, L). The Mn-doped crystal, on the contrary, has relatively weak polarized defect dipoles, strong local distortions, and a random field effect around the defect structure, all of which favor a disordered polarization orientation. This echoes the STEM

observations, as illustrated in Figures 2(C) and 2(F). This facilitates the enhancement of local structural heterogeneity, which is recognized as a key factor in boosting piezoelectric activity.

Figure 4. Defect dipoles. (A) Electron paramagnetic resonance (EPR) spectra of the pristine, Fe-doped, and Mn-doped KTN crystals at 100 K. (B) The O-1s X-ray photoelectron spectra (XPS) of the pristine, Fe-doped, and Mn-doped samples. S_1/S_2 is the ratio of the integrated area of oxygen vacancy ($V_O^{\bullet\bullet}$) and lattice oxygen peaks. (C) The schematic structure used in DFT simulation. The $[100]_c$, $[010]_c$, and $[001]_c$ crystallographic directions are given, and the spontaneous polarization P_s in the pristine KTN is along the $[011]_c$ direction. In the doped lattices, a Fe or Mn ion substitutes a Nb ion, and a $V_O^{\bullet\bullet}$ is created at one of the nearest neighboring oxygen ion sites of the dopant. (D) The $(100)_c$ -section views of differential charge density (DCD) of the pristine KTN lattice, the KTN lattice with $V_O^{\bullet\bullet}$, the KTN lattice with $(Fe_B^{\prime\prime} - V_O^{\bullet\bullet})^{\times}$, and

the KTN lattice with $(Mn'_B - V_o'')$, respectively, in the yellow box of (C). The Bader charges of the B-site cations and oxygen ion immediately adjacent to the vacancy site are marked. (E) Box charts of deviation angles of dipole moments from the $[011]_c$ direction in the octahedra within the $(100)_c$ sections containing the defect dipoles, statistically obtained from Figure S8. (F) Two-dimensional contour plots of the electron localization function (ELF) on the $(100)_c$ sections.”

New references [40-43,59-61] have been cited in the revised manuscript:

- 40 Tian, S. et al. Defect dipole stretching enables ultrahigh electrostrain. *Sci. Adv.* **10**, (2024).
- 41 Shannon, R. D. Revised effective ionic radii and systematic studies of interatomic distances in halides and chalcogenides. *Acta Cryst.* **32**, 751-767 (1976).
- 42 Rana, D. et al. Light-driven permanent transition from insulator to conductor. *Phys. Rev. B*, **104**, 245208 (2021).
- 43 Chi, Y.-T., et al. Complex Oxides under Simulated Electric Field: Determinants of Defect Polarization in ABO_3 Perovskites. *Adv. Sci.* **9**, 2104476 (2022).
- 59 Perdew, J. P., Burke, K., & Ernzerhof, M. *Phys. Rev. Lett.*, **77**, 3865-3868, (1996).
- 60 Heyd, J., Scuseria, G. E., & Ernzerhof, M. Hybrid functionals based on a screened Coulomb potential. *J. Chem. Phys.* **118**, 8207–8215 (2003).
- 61 Xu, X., et al. To define nonradiative defects in semiconductors: An accurate DLTS simulation based on first-principle. *Comp. Mater. Sci.*, **215**, 111760 (2020).

Some further technical comments:

1. What is the uncertainty (standard deviation) of the compositions reported in Table S1?

RESPONSE: Thanks for the good question. In this work, the compositions were investigated using an electron probe microanalyzer (EPMA, JXA-8230, JEOL), which is a microanalytical technique widely used for precise micro-composition analysis.

In the original version, we only used one set of data to represent the compositions

of the samples, thus not indicating the standard deviation. Based on the valuable suggestion, we obtained the average molar percentages and standard deviations of Ta, Nb, and B-site dopants by measuring the compositions at five points on pristine, Fe-doped, and Mn-doped samples, as shown in Table R1. Because KTN is a typical solid solution crystal, the B sites in KTN are randomly occupied by Ta or Nb ions, which yield a non-uniform distribution of compositions at a micro-scale. Therefore, the detected compositions of Ta, Nb, and B-site dopants exhibit some fluctuations. But on average, the doping levels of Fe and Mn are close to the proportion in the raw materials, both of which are approximately 0.5 mol%.

We have added the standard deviations of the compositions into Table S1 in the revised supporting information. The compositions are described using the way of $\mu \pm \sigma$, where μ is the average value and σ is the sample standard deviation.

Table R1. Compositional analysis of Ta, Nb, and B-site dopants in the pristine, Fe-doped, and Mn-doped samples. Here, μ is the average value, while σ is the sample standard deviation.

Samples (mol%)	Pristine KTN		Fe-doped KTN			Mn-doped KTN		
	Ta	Nb	Ta	Nb	Fe	Ta	Nb	Mn
No.1	42.06	57.94	42.08	57.40	0.52	42.52	56.93	0.55
No.2	42.97	57.03	41.31	58.24	0.45	43.12	56.42	0.46
No.3	40.95	59.05	43.64	55.88	0.48	41.59	57.98	0.43
No.4	41.45	58.55	42.22	57.18	0.60	42.43	56.91	0.66
No.5	42.38	57.62	41.90	57.47	0.63	41.00	58.43	0.57
$\mu \pm \sigma$	41.96± 0.79	58.04± 0.79	42.23± 0.86	57.23± 0.86	0.54± 0.08	42.13± 0.84	57.33± 0.84	0.53± 0.09

2. How does the doping affect the lattice parameters of KTN (in XRD)

RESPONSE: Thanks for the valuable comments. In this work, we measured the lattice constants of the orthorhombic KTN crystals by XRD diffraction with CuK α radiation (X'Pert PRO MPD, PANalytical). The XRD diffraction patterns are collected

at the (001)_C facet of the single crystal samples at room temperature. The characteristic peaks, as exhibited in Figure R5(A), correspond to (001)_C, (002)_C, and (003)_C planes, respectively, indicating a pure perovskite structure. To avoid destroying the single crystal samples, the method is often used to determine the lattice constants of single crystals in tetragonal phase and orthorhombic phase³⁹. In order to obtain the lattice constants, we extract the profiles of the (003)_C peak, as depicted in Figure R5(B). The higher peaks on the left are due to the combined effect of b and c , while the peaks on the right are caused by a . For pristine KTN, $a = 3.985 \text{ \AA}$, $b \approx c = 3.993 \text{ \AA}$, in terms of Bragg law. The phenomenon of $b \approx c$ is common in the orthorhombic materials⁴⁰⁻⁴². In the Fe-doped and Mn-doped KTN, the increased diffraction angles indicate the lattice contractions. For Fe-doped KTN, $a = 3.983 \text{ \AA}$, $b \approx c = 3.992 \text{ \AA}$, while for Mn-doped KTN, $a = 3.981 \text{ \AA}$, $b \approx c = 3.991 \text{ \AA}$. There are two main reasons for inducing the lattice contractions. On the one hand, previous studies have shown that the formation of oxygen vacancies can lead to the contraction in perovskite materials, due to the smaller effective vacancy radius as compared to the oxide ion^{43,44}. Thus, the oxygen vacancies caused by Fe and Mn doping at B-sites reduce the lattice constants. On the other hand, in contrast to Ta⁵⁺ and Nb⁵⁺, Fe³⁺ has a similar radius ($r = 0.645 \text{ \AA}$), while the radius of Mn⁴⁺ is much smaller ($r = 0.53 \text{ \AA}$)⁴⁵, which induces the more obvious lattice contraction in Mn-doped KTN. Despite these changes, the lattice cell volume contraction is minimal, only about 2%. Therefore, in the calculation section, in order to reveal the impact of doping and oxygen vacancies on the pristine local structure, we constructed the supercell using the lattice constants of pristine KTN. Then, we introduced the defect structure and performed the ionic relaxation. It is beneficial for investigating the role of defect structure as a single factor in the crystal.

Figure R5. X-ray diffraction pattern collected at the (001)_C facet at room temperature. (A) XRD pattern of pristine KTN. The inset shows the splitting of the (003)_C peak. (B) The characteristic peaks corresponding to the (003)_C planes in pristine, Fe-doped, and Mn-doped KTN.

3. What is the second-order Raman mode discussed in Figure 5? Can the authors show that this is really a second-order mode and not a normal Raman mode? The full Raman spectrum should be compared between theory and experiment.

RESPONSE: Thanks for the valuable comments. Based on the suggestions, we have added relevant supplementary notes in the revised manuscript.

(Q1) What is the second-order Raman mode discussed in Figure 5? Can the authors show that this is really a second-order mode and not a normal Raman mode?

The second-order Raman scattering is a scattering process involving the two-phonon processes. In this work, the Raman peak of the sample located at $\sim 590 \text{ cm}^{-1}$ is a second-order Raman peak. Momentum conservation in second-order Raman scattering processes is not restricted to the Γ point⁴⁶. The wave vector selection rule for the second-order Raman effect allows phonons from the whole Brillouin zone to participate in the scattering process⁴⁷. Previous studies have shown that a strong second-order peak does exist at $\sim 590 \text{ cm}^{-1}$ in KTN crystals, stemming from the combination of the transverse $A_1(3\text{TO})$ mode and the acoustic TA mode at the critical X -point of the Brillouin zone^{20,21}. Hence, we assign the $\sim 590 \text{ cm}^{-1}$ mode as a second-order Raman

mode.

The Raman spectra calculated by DFPT do not include the second-order Raman effect. Hence, the second-order Raman peaks do not appear in the calculation results. Given that the second-order Raman scattering is a two-phonon process, we obtained the Raman shift of this mode by calculating the phonon spectra. The program, Phonon Unfolding, was used for unfolding phonon dispersions of supercells⁴⁸. The phonon spectra of the optical $A_1(3TO)$ mode and the acoustic TA mode are shown in Figure R6. For the orthorhombic structure, the wavenumbers of the $A_1(3TO)$ and TA modes at the X point of the Brillouin zone are 507 and 90 cm^{-1} , respectively. For the tetragonal structure, the wavenumbers of the $A_1(3TO)$ and TA modes at the X point of the Brillouin zone are 507 and 97 cm^{-1} , respectively. The sum of the two phonons induces the generation of the second-order Raman mode with a Raman shift of $\sim 600 \text{ cm}^{-1}$, which is in good agreement with the $\sim 590 \text{ cm}^{-1}$ peak measured in the experiment. This is good proof that this peak is a second-order Raman mode rather than a normal Raman mode.

Based on this suggestion, we have added the description of the origin of the second-order Raman peak in the revised manuscript on Line 451, as follows: “The third fitted peak at $\sim 590 \text{ cm}^{-1}$, associated with a second-order mode in KTN crystals, stems from the combination of the transverse $A_1(3TO)$ mode and the acoustic TA mode at the critical X -point of the Brillouin zone^{50,51},” and the following Figure R6 and its relevant description have been added to the revised supplementary information, as follows:

Figure R6 (Figure S11 in the supplementary information). The phonon spectra of the optical $A_1(3TO)$ mode and the acoustic TA mode in (A) orthorhombic KTN and (B)

tetragonal KTN, respectively. In the orthorhombic KTN, the wavenumbers of the $A_1(3TO)$ and TA modes at the X point of the Brillouin zone are 507 and 90 cm^{-1} , respectively. In the tetragonal KTN, the wavenumbers of the $A_1(3TO)$ and TA modes at the X point of the Brillouin zone are 507 and 97 cm^{-1} , respectively. The sum of the two phonons induces the generation of the second-order Raman mode with a Raman shift of $\sim 600 \text{ cm}^{-1}$, which is in good agreement with the $\sim 590 \text{ cm}^{-1}$ peak measured in the experiment. This alignment strongly supports the identification of the third fitted peak as a second-order Raman mode.

New references [50,51] have been cited in the revised manuscript:

- 50 Kugel, G. E., Mesli, H., Fontana, M. D. & Rytz, D. Experimental and theoretical study of the Raman spectrum in $\text{KTa}_{1-x}\text{Nb}_x\text{O}_3$ solid solutions. *Phys. Rev. B* **37**, 5619-5628 (1988).
- 51 Manlief, S. K. & Fan, H. Y. Raman Spectrum of $\text{KTa}_{0.64}\text{Nb}_{0.36}\text{O}_3$. *Phys. Rev. B* **5**, 4046-4060 (1972).

(Q2) The full Raman spectrum should be compared between theory and experiment.

Thanks for the good suggestion. The comparison between theory and experiment on the full Raman spectrum [that is, Figure R7] has been added in the revised supplementary information.

Figures R7(A) and (B) show the Raman spectra of the sample in orthorhombic phase under zero electric field and the sample after the orthorhombic-tetragonal phase transition under a 24 kV/cm electric field, respectively. We obtained the different vibrational modes by fitting the spectra. The experimental Raman spectra contain the contributions from the normal Raman modes (i.e., the first-order Raman modes), the second-order Raman modes, and the central peak (CP). Since only the normal Raman modes are involved in the DFPT calculation of Raman spectra, we here compare the experimental and theoretical normal Raman modes. As clearly shown in Figure R7, the experimental and theoretical results are in good agreement.

Figure R7. The comparison between experimental and theoretical Raman spectra. (A, B) are the full experimental Raman spectra of orthorhombic and tetragonal KTN crystals, respectively. The normal Raman modes (i.e., the first-order Raman modes), the second-order Raman modes, and the central peak are obtained by fitting. The fitted curves (red lines), containing the contributions from these three parts, are in consistent with the experimental results. (C, D) show the calculated Raman spectra of orthorhombic and tetragonal KTN crystals, respectively. The red sticks indicate the calculated Raman intensities of the normal modes (i.e., the first-order modes). The blue lines are computed by applying a Gaussian broadening of 80 cm⁻¹ to the discrete spectrum. The grey line is the sum of the blue lines.

From the experimental spectra, multiple modes are blended below the Raman shift of 400 cm⁻¹. In this range, the second-order mode from the overtone of TA mode is far stronger than the normal modes, while a strong Fano resonance is formed due to the coupling of the CP and TO phonon at about 210 cm⁻¹⁴⁹. Moreover, the CP is also intense in this range. These modes hinder the accurate rendering of the normal Raman modes

by fitting. In contrast, above 400 cm^{-1} , the Raman peaks are clearer, and the first-order Raman modes are strong, especially in the range of $400\text{-}700\text{ cm}^{-1}$. Importantly, the A_1 and B_1 vibration modes of BO_6 octahedron, closely related to the polarization characteristics, are located in the range of $400\text{-}700\text{ cm}^{-1}$. Hence, in this work, we chose the Raman spectra in the range of $400\text{-}700\text{ cm}^{-1}$ to study the field-induced phase transition process.

In addition to Figure R7 being added to the supplementary information, the relevant description has been added on Line 453 in the revised manuscript, as follows: “The full Raman spectra are compared between theory and experiment, as given in Figure S12”.

4. Raman spectra were simulated with Quantum Espresso. What pseudopotentials were used in these calculations? Why were supercells used in Raman spectrum calculations, this complicates the interpretation of vibrational modes?

RESPONSE: Thanks for the valuable comments. In this work, we simulated Raman spectra of KTN with Quantum Espresso. According to the reviewer’s questions, a more detailed description has been added in the revised manuscript, also given below.

(Q1) Raman spectra were simulated with Quantum Espresso. What pseudopotentials were used in these calculations?

The calculation methods of vibrations and Raman intensities mainly include the density functional perturbation theory (DFPT) method and finite displacement method (FDM), et al. As compared to other methods, DFPT requires only one simulation to get the dynamical matrix and compute all modes. It can calculate phonon frequencies at arbitrary wave vectors avoiding the use of large supercells and with a workload that is essentially independent of the phonon wavelength⁵⁰. Moreover, DFPT constructs the full set of Raman tensors during the phonon perturbations in the Raman intensity calculation⁵¹. Hence, we chose the DFPT method to calculate vibrational properties using the Quantum Espresso software package (QE). To meet the requirements of Raman spectrum calculations using QE, norm-conserving pseudopotentials with a 100 Ry plane wave cutoff energy, and the local-density approximation (LDA) exchange-

correlation function were used^{52,53}. These approximations can well describe the vibrational properties of perovskites, which has been demonstrated by previous studies⁵⁴. We apologize for the lack of a detailed description of the simulation in the manuscript. The relevant descriptions have been added on Line 613 in the revised manuscript, as follows: "To calculate the Raman spectra, density functional perturbation theory (DFPT) was conducted and implemented in the Quantum ESPRESSO software package, using norm-conserving pseudopotentials with a 100 Ry plane wave cutoff energy, and the local-density approximation (LDA) exchange-correlation function^{47,63-65}."

(Q2) Why were supercells used in Raman spectrum calculations, this complicates the interpretation of vibrational modes?

The supercells are suitable for representing the intercalation of different components in solid solution crystals, though they introduce more vibrations. In the double perovskites $A_2B'B''X_6$, B-site cations favor rock salt ordering⁵⁵. To describe the arrangement of Ta and Nb ions at the B-site, the $\sqrt{2} \times \sqrt{2} \times 2$ supercells used in the DFPT calculations are already as simple as possible in terms of structure. The size of the supercell can present the cation ordering and octahedral distortion patterns. Therefore, the $\sqrt{2} \times \sqrt{2} \times 2$ supercell is often used to characterize vibrational properties in perovskite solid solutions⁵⁶, which is beneficial for more accurate results. Moreover, the Raman modes $A_1(3TO)$ and $B_1(3TO)$ used in the manuscript in the range of 400-700 cm^{-1} are all intrinsic vibrational modes of BO_6 octahedron, not perturbed by the supercell size⁵⁷. In summary, in this work, we employed the $\sqrt{2} \times \sqrt{2} \times 2$ supercells in Raman spectrum calculations. In the revised manuscript, we further explain the reason for using the $\sqrt{2} \times \sqrt{2} \times 2$ supercells in the DFPT calculations on Line 616, as follows: "... In the double perovskites $A_2B'B''X_6$, B-site cations favor rock salt ordering⁶⁶. To describe the B-site cation ordering, the $\sqrt{2} \times \sqrt{2} \times 2$ KTN supercells with orthorhombic and tetragonal symmetries for DFPT calculations were

constructed ... The Raman modes $A_1(3TO)$ and $B_1(3TO)$ in the range of 400-700 cm^{-1} are both intrinsic vibrational modes of BO_6 octahedron, not perturbed by the supercell size⁶⁸.”.

New references [66,68] have been cited in the revised manuscript:

66 King, G. & Woodward, P. M. Cation ordering in perovskites. *J. Mater. Chem.* **20**, 5785–5796 (2010).

68 Raman, C. V. The vibration spectrum of a crystal lattice. *Proc. Ind. Acad. Sci.* **18**, 237-250, (1943).

Response to Reviewer #3

Piezoelectricity endows the perovskite ferroelectrics at the heart of electromechanical systems spanning from macro to micro/nano scales. Defect engineering strategies based on transition metal doping offer great potential for adjusting piezoelectric performance. It is crucial to explore the key roles of transition metal impurities and derivative defect dipoles on piezoelectric properties. This study investigates how the transition metal impurity shapes piezoelectric properties, taking the Fe-doped and Mn-doped KTN as paradigms. The work provides insights into the micro-meso structure and macroscopic performances to elucidate the routes to control the piezoelectric properties via transition metal doping. The comprehensive understanding of the impacts of transition metals on the piezoelectric properties, as observed in KTN crystals, lays an essential basis for the design of perovskite ferroelectrics. The manuscript is innovative and well-organized. Hence, the work deserves publication in Nature Communications after addressing the following points with minor revisions.

RESPONSE: We sincerely appreciate the reviewer for the valuable comments and the high recognition of our work. We have revised the manuscript according to the reviewer's suggestions.

1. It is well known that the strain-hysteresis performance of piezoelectric materials is of great significance for realizing high-speed and accurate signal control. In the manuscript, KTN-based single crystals show excellent low-strain-hysteresis characteristics. Please add relevant data to evaluate the strain-hysteresis characteristics.

RESPONSE: Thanks very much for the valuable suggestion. The evaluation and description of hysteresis characteristics have been added in the revised manuscript.

In the revised manuscript, the hysteresis characteristics were evaluated using the index of strain hysteresis H_{ys} , determined by

$$H_{ys} = \Delta S_{E_{\max}/2} / S_{E_{\max}}$$

where $S_{E_{\max}}$ is the strain at the maximum electric field E_{\max} , and $\Delta S_{E_{\max}/2}$ is the strain

difference at the half value of the maximum electric field⁵⁸. Using the S - E loops in Figure 1(D-F), the H_{ys} values were calculated for pristine, Fe-doped, and Mn-doped KTN to be 7.1%, 1.3%, and 4.2%, respectively. The H_{ys} values demonstrate the ultra-low hysteresis of the samples^{59,60}. We have added the relevant descriptions in the revised manuscript on Line 141 as follows: “The results reveal a highly linear relationship between the induced strain and applied electric field up to 20 kV/cm with minimal hysteresis. The indexes of strain hysteresis H_{ys} for pristine, Fe-doped, and Mn-doped KTN are on the order of 7.1%, 1.3%, and 4.2%, respectively, determined by $H_{ys} = \Delta S_{E_{max}/2} / S_{E_{max}}$ ²¹. Here $S_{E_{max}}$ represents the strain at the maximum electric field E_{max} and $\Delta S_{E_{max}/2}$ is the strain difference at half of E_{max} . The low H_{ys} values demonstrate the stable polarization structures after poling treatment. This underscores a typical feature of intrinsic piezoelectric response, with the piezoelectric strain coefficient up to 1000 pm/V for Mn-doped KTN.”

The new reference [21] has been cited in the revised manuscript:

21 Yin, J. et al. Superior and anti-fatigue electro-strain in $\text{Bi}_{0.5}\text{Na}_{0.5}\text{TiO}_3$ -based polycrystalline relaxor ferroelectrics. *J. Mater. Chem. A* 7, 5391-5401 (2019).

2. It seems that the “Figure 4I” in line 342 should be corrected to “Figure 4E”. Please check throughout the manuscript and supplementary information for accurate descriptions.

RESPONSE: Really appreciate for the careful reading and pointing out the mistakes.

We have revised the “Figure 4I” in line 351 to “Figure 4C” in the revised manuscript.

We have also checked throughout the text to avoid similar errors.

3. In line 589, in order to construct the supercell, the manuscript provided the values of lattice constants, while the XRD diffraction data was not given. Please give the source of data.

RESPONSE: Thanks for the valuable comments. In this work, we measured the lattice constants of the orthorhombic KTN crystals by XRD diffraction with $\text{CuK}\alpha$ radiation

(X'Pert PRO MPD, PANalytical). The X-ray diffraction pattern collected at the (001)_C facet at room temperature is shown in Figure R8. The characteristic peaks correspond to (001)_C, (002)_C, and (003)_C planes, respectively, indicating a pure perovskite structure. The differences among lattice constants a , b , and c induce the splitting of the peaks. The inset of Figure R8 shows the profile of (003)_C peak. The higher peak on the left is due to the combined effect of b and c , while the peak on the right is caused by a . Hence, $a = 3.985 \text{ \AA}$, $b \approx c = 3.992 \text{ \AA}$, in terms of Bragg law. The XRD pattern of orthorhombic KTN has been added in the revised supporting information. The lattice constants of tetragonal KTN crystal, i.e., $a = b = 3.991 \text{ \AA}$, $c = 4.007 \text{ \AA}$, are cited from the reference [25].

According to the suggestion, we have clearly stated the sources of the data in the revised manuscript on Line 592, as follows: “Lattice constants of experimental orthorhombic crystal structure ($a = 3.985 \text{ \AA}$, $b \approx c = 3.992 \text{ \AA}$), measured by X-ray diffraction, were used to construct the supercells. The XRD diffraction pattern is shown in Figure S14.” and “The lattice parameters of tetragonal KTN are cited from the reference⁶⁷”.

Figure R8 (Supplementary Figure S14). X-ray ($\lambda = 1.5406 \text{ \AA}$) diffraction pattern collected at the (001)_C facet at room temperature. The inset shows the splitting of the (003)_C peak. Lattice constants of $a = 3.985 \text{ \AA}$ and $b \approx c = 3.992 \text{ \AA}$ are obtained.

The new reference [67] has been cited in the revised manuscript:

67 Wang, Y. et al. Manganese-doping enhanced local heterogeneity and piezoelectric properties in potassium tantalate niobate single crystals. *IUCrJ* **8**,

4. In figures S6 (a3 and a4), why is the overall direction of the curved domain walls in the Mn-doped KTN different from that of the domain walls in the pristine KTN?

RESPONSE: Thanks for the good comments. This difference mainly arises from the stronger polarization disorder within the Mn-doped crystal compared to the pristine KTN crystal.

There are 12 independent directions of the spontaneous polarizations in the orthorhombic ferroelectric crystals, allowing the formation of O60, O90, O120, and O180 domain walls⁶¹⁻⁶³. The set of mechanically compatible and electrically neutral domain walls in the orthorhombic phase is illustrated in Figure R9(A). The non-180° domain walls can be observed using a polarized light microscope. When observing along the $[001]_c$ direction, the complex domain configurations can yield different domain walls along eight directions, as shown in Figure R9(B). After poling the samples along the $[001]_c$ direction, the domain walls are dominated by the O60 and O120⁶⁴. Therefore, the domain walls along the $\langle 110 \rangle_c$ and $\langle 010 \rangle_c$ directions can be observed along the $[001]_c$ direction in the $[001]_c$ -poled samples, as shown in Figure R9(C).

Figure R9. Domain configurations in the orthorhombic crystal. (A) Set of mechanically compatible and electrically neutral domain walls in the orthorhombic phase. In the case

of 180° domain walls, where the orientation is not determined by symmetry, walls with the most important crystallographic directions are displayed. The blue arrows exhibit the directions of dipolar moments while the grey planes exhibit the domain walls. (B) Possible domain wall directions observed along the $[001]_C$ direction. The red lines are all possible observed domain walls, with two adjacent lines at an angle of 22.5° , where the solid lines are domain walls along the $\langle 110 \rangle_C$ and $\langle 010 \rangle_C$ directions. (C) The typical configurations of domain walls after poling the samples along the $[001]_C$ direction. The red solid lines represent the domain walls.

In both pristine and Mn-doped KTN samples, domain walls along the $\langle 110 \rangle_C$ direction are clearly visible, as shown in Figure R10. The overall orientation of the curved domain walls in Mn-doped KTN extends in the $[010]_C$ direction. The $\langle 010 \rangle_C$ -direction domain walls can also be found in pristine KTN, but with a relatively low density. Hence, the two type domain walls depicted in Figure R9(C) are present in both pristine and Mn-doped KTN.

Figure R10. Observations of the domain structures of (A) pristine and (B) Mn-doped KTN samples using polarized light microscopy. (C) Locally amplified domain structure in (B), showing the curved domain walls.

Unlike ferroelectric 180° domains, $O60$ and $O120$ domain walls exhibit both ferroelectric and ferroelastic features. The morphologies of the non- 180° domain walls are affected by the total free energy including strain energy, dipole-dipole interaction, and electrostatic energy⁶⁵. The latter two parts are independent of the domain wall area. The disordered polarization in the Mn-doped crystal tends to introduce greater inhomogeneity of the dipole-dipole and electrostatic effects, leading to curved domain walls that help reduce the total free energy. As clearly shown in Figure R10(C), the

curved domain walls are mainly O120 domain walls, suggesting their role in compensating for the additional interfacial energy arising from the discontinuous polarization. To minimize the interfacial energy in the Mn-doped crystal, the density of O120 domain walls is increased. The curvature of these domain walls is indicative of a larger interfacial energy, which is a characteristic of structural heterogeneity in the Mn-doped crystal.

According to this suggestion, we have added the relevant description in the revised manuscript on Line 223, as follows: “Similar results were also identified on a larger scale using polarized light microscopy (PLM), as shown in Figure S6. For the $[001]_c$ -poled samples, the Mn-doped crystal exhibits a higher density of O120 domain walls compared to its pristine counterpart when observed along the $[001]_c$ direction. Moreover, the Mn-doped sample exhibits curved domain walls, which tend to reduce the dipole-dipole interaction and electrostatic energies strengthened by disordered polar vectors²⁵. The curvature of these domain walls is indicative of a larger interfacial energy which is a characteristic of structural heterogeneity. This delicate balance between interfacial and bulk energies is expected to flatten the local thermodynamic energy landscape, thus facilitating polarization rotations under an electric field²⁶.”, and Figure S6 in the revised supplementary information has been modified in accordance with Figure R11.

Figure R11. Domain structures observed along the $[001]_c$ direction using polarizing

light microscopy. (A, B) are the domain structures in the pristine and Mn-doped KTN samples after poling along the $[001]_C$ direction, respectively. (C) The typical configurations of domain walls after poling along the $[001]_C$ direction. The red solid lines represent the domain walls. In the $[001]_C$ -poled samples, the direction of macroscopic polarization remains the same as the $[001]_C$ direction, so that the domain walls are dominated by the O60 and O120. Therefore, the domain walls along the $\langle 110 \rangle_C$ and $\langle 010 \rangle_C$ directions can be observed along the $[001]_C$ direction. The morphologies of the O60 and O120 domain walls are affected by the total free energy including strain energy, dipole-dipole interaction, and electrostatic energy⁶⁵. The latter two parts are independent of the area of the domain wall. The disordered polarization in the Mn-doped crystal tends to introduce larger dipole-dipole interaction and electrostatic energy, which can be reduced by the curving of the domain walls. Hence, to minimize the total free energy in the Mn-doped crystal, the curved O120 domain walls are increased.

References

- 1 Du, G. *et al.* Internal bias field relaxation in poled Mn-doped $\text{Pb}(\text{Mn}_{1/3}\text{Sb}_{2/3})\text{O}_3$ – $\text{Pb}(\text{Zr,Ti})\text{O}_3$ ceramics. *Ceramics International* **39**, 7703-7708, (2013).
- 2 Li, B., Ehmke, M. C., Blendell, J. E. & Bowman, K. J. Optimizing electrical poling for tetragonal, lead-free BZT–BCT piezoceramic alloys. *Journal of the European Ceramic Society* **33**, 3037-3044, (2013).
- 3 Zhao, Z., Dai, Y., Li, X., Zhao, Z. & Zhang, X. The evolution mechanism of defect dipoles and high strain in MnO_2 -doped KNN lead-free ceramics. *Applied Physics Letters* **108**, 172906 (2016).
- 4 Yang, Y. *et al.* Sm and Mn co-doped PMN-PT piezoelectric ceramics: Defect engineering strategy to achieve large d_{33} and high Qm. *Journal of Materials Science & Technology* **137**, 143-151, (2023).
- 5 Erdem, E. *et al.* Defect structure in aliovalently-doped and isovalently-substituted PbTiO_3 nano-powders. *Journal of Physics: Condensed Matter* **22**, 399804-399804, (2010).
- 6 Eichel, R.-A. Structural and dynamic properties of oxygen vacancies in perovskite oxides—analysis of defect chemistry by modern multi-frequency and pulsed EPR techniques. *Phys. Chem. Chem. Phys.* **13**, 368-384, (2011).
- 7 Erhart, P., Eichel, R.-A., Träskelin, P. & Albe, K. Association of oxygen vacancies with impurity metal ions in lead titanate. *Physical Review B* **76**, 174116 (2007).
- 8 Eichel, R.-A. *et al.* Defect-dipole formation in copper-doped PbTiO_3 ferroelectrics. *Physical Review Letters* **100**, 095504 (2008).
- 9 Meštrić, H. *et al.* Iron-oxygen vacancy defect centers in PbTiO_3 : Newman superposition model analysis and density functional calculations. *Physical Review B* **71**, 134109 (2005).
- 10 Nossa, J. F., Naumov, I. I. & Cohen, R. E. Effects of manganese addition on the electronic structure of BaTiO_3 . *Physical Review B* **91**, 214105 (2015).
- 11 Emery, A. A. & Wolverton, C. High-throughput DFT calculations of formation energy, stability and oxygen vacancy formation energy of ABO_3 perovskites.

- Scientific Data* **4**, 170153 (2017).
- 12 Tan, H. *et al.* First-principles studies of the local structure and relaxor behavior of $\text{Pb}(\text{Mg}_{1/3}\text{Nb}_{2/3})\text{O}_3\text{-PbTiO}_3$ -derived ferroelectric perovskite solid solutions. *Physical Review B* **97**, 174101 (2018).
 - 13 Li, F. *et al.* Giant piezoelectricity of Sm-doped $\text{Pb}(\text{Mg}_{1/3}\text{Nb}_{2/3})\text{O}_3\text{-PbTiO}_3$ single crystals. *Science* **364**, 264-268 (2019).
 - 14 Kim, Y.-H. *et al.* Comprehensive defect suppression in perovskite nanocrystals for high-efficiency light-emitting diodes. *Nature Photonics* **15**, 148-155 (2021).
 - 15 Tian, S. *et al.* Defect dipole stretching enables ultrahigh electrostrain. *Science Advances* **10**, eadn2829 (2024).
 - 16 Eya, H. I. & Dzade, N. Y. Density functional theory insights into the structural, electronic, optical, surface, and band alignment properties of BaZrS_3 chalcogenide perovskite for photovoltaics. *ACS Applied Energy Materials* **6**, 5729-5738 (2023).
 - 17 Jain, D., Chaube, S., Khullar, P., Goverapet Srinivasan, S. & Rai, B. Bulk and surface DFT investigations of inorganic halide perovskites screened using machine learning and materials property databases. *Physical Chemistry Chemical Physics* **21**, 19423-19436 (2019).
 - 18 Xue, H., Brocks, G. & Tao, S. First-principles calculations of defects in metal halide perovskites: A performance comparison of density functionals. *Physical Review Materials* **5**, 125408 (2021).
 - 19 Zhang, J., Xu, R., Damodaran, A. R., Chen, Z. H. & Martin, L. W. Understanding order in compositionally graded ferroelectrics: Flexoelectricity, gradient, and depolarization field effects. *Physical Review B* **89**, 224101 (2014).
 - 20 Kugel, G. E., Mesli, H., Fontana, M. D. & Rytz, D. Experimental and theoretical study of the Raman spectrum in $\text{KTa}_{1-x}\text{Nb}_x\text{O}_3$ solid solutions. *Physical Review B* **37**, 5619-5628 (1988).
 - 21 Manlief, S. K. & Fan, H. Y. Raman Spectrum of $\text{KTa}_{0.64}\text{Nb}_{0.36}\text{O}_3$. *Physical Review B* **5**, 4046-4060 (1972).
 - 22 Zhengran, C. *et al.* Poling above the Curie temperature driven large

- enhancement in piezoelectric performance of Mn doped PZT-based piezoceramics. *Nano Energy* **113**, 108546 (2023).
- 23 Sangawar, S. R., Praveenkumar, B., Divya, P. & Kumar, H. H. Fe doped hard PZT ceramics for high power SONAR transducers. *Materials Today: Proceedings* **2**, 2789-2794 (2015).
- 24 Huang, X. *et al.* Impact of defect concentration on piezoelectricity in Mn/Fe-doped KTN crystals. *Applied Physics Letters* **124**, 192906 (2024).
- 25 Wang, Y. *et al.* Manganese-doping enhanced local heterogeneity and piezoelectric properties in potassium tantalate niobate single crystals. *IUCrJ* **8**, 319-326 (2021).
- 26 Stoliaroff, A. & Latouche, C. Accurate ab initio calculations on various PV-based materials: Which functional to be used? *The Journal of Physical Chemistry C* **124**, 8467-8478 (2020).
- 27 Meng, Y. *et al.* When density functional approximations meet iron oxides. *Journal of Chemical Theory and Computation* **12**, 5132-5144 (2016).
- 28 Franchini, C., Podloucky, R., Paier, J., Marsman, M. & Kresse, G. Ground-state properties of multivalent manganese oxides: Density functional and hybrid density functional calculations. *Physical Review B* **75**, 195128 (2007).
- 29 Liu, M., Gopakumar, A., Hegde, V. I., He, J. & Wolverton, C. High-throughput hybrid-functional DFT calculations of bandgaps and formation energies and multifidelity learning with uncertainty quantification. *Physical Review Materials* **8**, 043803 (2024).
- 30 Wang, T., Tan, X., Wei, Y. & Jin, H. Accurate bandgap predictions of solids assisted by machine learning. *Materials Today Communications* **29**, 102932 (2021).
- 31 Rosen, A. S., Notestein, J. M. & Snurr, R. Q. Comparing GGA, GGA+U, and meta-GGA functionals for redox-dependent binding at open metal sites in metal-organic frameworks. *The Journal of Chemical Physics* **152**, 224101 (2020).
- 32 Zhu, H. *et al.* Rhombohedral BiFeO₃ thick films integrated on Si with a giant

- electric polarization and prominent piezoelectricity. *Acta Materialia* **200**, 305-314 (2020).
- 33 Zhou, K. *et al.* Interfacial C-O covalent bonds improving the piezo-assisted photocatalytic performance of Bi₄Ti₃O₁₂@Carbon Schottky heterojunction. *Chemical Engineering Journal* **480**, 148012 (2024).
- 34 Zhao, X.-G., Malyi, O. I., Billinge, S. J. L. & Zunger, A. Intrinsic local symmetry breaking in nominally cubic paraelectric BaTiO₃. *Physical Review B* **105**, 224108 (2022).
- 35 Bai, H., Ip, W. F., Feng, W. & Pan, H. Effect of ferroelectric polarization on the oxygen evolution reaction: a theoretical study of M₁Sn₂S₆ (M = Bi, Mn, and Sb). *Journal of Materials Chemistry A* **12**, 7724-7731 (2024).
- 36 Jiang, H. First-principles approaches for strongly correlated materials: A theoretical chemistry perspective. *International Journal of Quantum Chemistry* **115**, 722-730 (2015).
- 37 Dongho Nguimdo, G. M. & Joubert, D. P. A density functional (PBE, PBEsol, HSE06) study of the structural, electronic and optical properties of the ternary compounds AgAlX₂ (X = S, Se, Te). *The European Physical Journal B* **88**, 113 (2015).
- 38 Ngalyo, R. *et al.* Strong in-plane magnetization and spin polarization in (Co_{0.15}Fe_{0.85})₅GeTe₂/graphene van der Waals heterostructure spin-valve at room temperature. *ACS Nano*, 18, 5240-5248 (2024).
- 39 Hu, Y. *et al.* Effect of Mn-doping on optical properties of lead-free (K_{0.4}Na_{0.6})NbO₃ ferroelectric single crystals. *Journal of the European Ceramic Society* **40**, 2917-2921 (2020).
- 40 Zhao, Y., Liu, J., Zhang, X. & Zhou, H. Domain switching mechanism of orthorhombic-tetragonal coexistence (Li,K,Na)NbO₃ ceramics. *Journal of Alloys and Compounds* **763**, 695-700 (2018).
- 41 Wongsanmai, S., Ananta, S. & Yimnirun, R. Effect of Li addition on phase formation behavior and electrical properties of (K_{0.5}Na_{0.5})NbO₃ lead free ceramics. *Ceramics International* **38**, 147-152 (2012).

- 42 Du, J. *et al.* Effects of $\text{BiFe}_{0.5}\text{Ta}_{0.5}\text{O}_3$ addition on electrical properties of $\text{K}_{0.5}\text{Na}_{0.5}\text{NbO}_3$ lead-free piezoelectric ceramics. *Ceramics International* **42**, 1943-1949 (2016).
- 43 Marrocchelli, D., Perry, N. H. & Bishop, S. R. Understanding chemical expansion in perovskite-structured oxides. *Physical Chemistry Chemical Physics* **17**, 10028-10039 (2015).
- 44 Chatzichristodoulou, C., Norby, P., Hendriksen, P. V. & Mogensen, M. B. Size of oxide vacancies in fluorite and perovskite structured oxides. *Journal of Electroceramics* **34**, 100-107 (2014).
- 45 Wang, L. *et al.* Improved electrical properties for Mn-doped lead-free piezoelectric potassium sodium niobate ceramics. *AIP Advances* **5**, 097120 (2015).
- 46 Nemanich, R. J. & Solin, S. A. First- and second-order Raman scattering from finite-size crystals of graphite. *Physical Review B* **20**, 392-401 (1979).
- 47 García-Cristóbal, A., Cantarero, A., Trallero-Giner, C. & Cardona, M. Excitonic model for second-order resonant Raman scattering. *Physical Review B* **49**, 13430-13445 (1994).
- 48 Zheng, F. & Zhang, P. Phonon Unfolding : A program for unfolding phonon dispersions of materials. *Computer Physics Communications* **210**, 139-144 (2017).
- 49 Rahaman, M. M., Imai, T., Sakamoto, T., Tsukada, S. & Kojima, S. Fano resonance of Li-doped $\text{KTa}_{1-x}\text{Nb}_x\text{O}_3$ single crystals studied by Raman scattering. *Scientific Reports* **6**, srep23898 (2016).
- 50 Baroni, S., de Gironcoli, S., Dal Corso, A. & Giannozzi, P. Phonons and related crystal properties from density-functional perturbation theory. *Reviews of Modern Physics* **73**, 515-562 (2001).
- 51 Miwa, K. Prediction of Raman spectra with ultrasoft pseudopotentials. *Physical Review B* **84**, 094304 (2011).
- 52 Giannozzi, P. *et al.* QUANTUM ESPRESSO: a modular and open-source software project for quantum simulations of materials. *Journal of Physics:*

- Condensed Matter* **21**, 395502 (2009).
- 53 Hamann, D. R. Optimized norm-conserving Vanderbilt pseudopotentials. *Physical Review B* **88**, 085117 (2013).
- 54 Pérez-Osorio, M. A. *et al.* Raman spectrum of the organic–inorganic halide perovskite $\text{CH}_3\text{NH}_3\text{PbI}_3$ from first principles and high-resolution low-temperature Raman measurements. *The Journal of Physical Chemistry C* **122**, 21703-21717 (2018).
- 55 King, G. & Woodward, P. M. Cation ordering in perovskites. *Journal of Materials Chemistry* **20**, 5785-5796 (2010).
- 56 Dastidar, S. *et al.* High chloride doping levels stabilize the perovskite phase of cesium lead iodide. *Nano Letters* **16**, 3563-3570 (2016).
- 57 Raman, C. V. The vibration spectrum of a crystal lattice. *Proceedings of the Indian Academy of Sciences - Section A* **18**, 237-250 (1943).
- 58 Yin, J. *et al.* Superior and anti-fatigue electro-strain in $\text{Bi}_{0.5}\text{Na}_{0.5}\text{TiO}_3$ -based polycrystalline relaxor ferroelectrics. *Journal of Materials Chemistry A* **7**, 5391-5401 (2019).
- 59 Liu, Y. *et al.* Ultra-low strain hysteresis in BaTiO_3 -based piezoelectric multilayer actuators via microstructural texture engineering. *Journal of Materiomics*, doi:10.1016/j.jmat.2024.05.001 (2024).
- 60 Li, T. *et al.* Giant strain with low hysteresis in A-site-deficient $(\text{Bi}_{0.5}\text{Na}_{0.5})\text{TiO}_3$ -based lead-free piezoceramics. *Acta Materialia* **128**, 337-344 (2017).
- 61 Renault, A.-E., Dammak, H., Calvarin, G., Gaucher, P. & Pham Thi, M. Electric-field-induced orthorhombic phase in $\text{Pb}[(\text{Zn}_{1/3}\text{Nb}_{2/3})_{0.955}\text{Ti}_{0.045}]\text{O}_3$ single crystals. *Journal of Applied Physics* **97**, 044105 (2005).
- 62 Marton, P., Rychetsky, I. & Hlinka, J. Domain walls of ferroelectric BaTiO_3 within the Ginzburg-Landau-Devonshire phenomenological model. *Physical Review B* **81**, 144125 (2010).
- 63 Grünebohm, A. *et al.* Interplay of domain structure and phase transitions: theory, experiment and functionality. *Journal of Physics: Condensed Matter* **34**, 073002 (2022).

- 64 Sun, X. *et al.* Microscopic insight into domain configuration in orthorhombic $\text{K}_{0.52}\text{Na}_{0.48}\text{NbO}_3$ single crystals driven by electric-field. *Journal of the European Ceramic Society* **44**, 1581-1587 (2024).
- 65 Su, D. *et al.* Morphology and mobility of 90° domains in La-substituted bismuth titanate. *J Phys-Condens Mat* **16**, 4549-4556 (2004).

Response to Reviewers' comments and the description of revisions in the revised manuscript and supplementary information

We sincerely thank the reviewers for their recommendations and suggestions to further improve our work. We have revised our manuscript accordingly and the point-by-point responses are enclosed below. We hope our responses, along with the revisions we have made, will now make this manuscript suitable for publication in *Nature Communications*.

[Reviewers' comments are in black; Author responses are in blue; Revisions in the manuscript are highlighted.]

Response to Reviewer #1

The revised manuscript is ready for publication.

RESPONSE: We sincerely appreciate the reviewer's positive comment and the suggestion for the acceptance of our manuscript.

Response to Reviewer #2

The authors have improved the manuscript by seriously considering comments from all three referees. With respect to my original comments, the most significant change was using hybrid HSE exchange-correlation functional for investigating the electronic structure. For structural optimizations, using PBE is sufficient in this case. But for electronic structure studies, PBE can be even qualitatively wrong in the case of strongly correlated oxides with open d-shells. The use of HSE has made work more rigorous. My remaining few comments are as follows. After considering the points, which are minor, the work is publishable.

RESPONSE: We sincerely appreciate the reviewer's positive comments, and we also thank the reviewer for the additional valuable suggestions to improve the manuscript.

1. I don't really see the point of using ELF to describe the electronic structure. ELF

values of less than 1/2 mean electron localization that is less than in homogenous electron gas. So, discussing ELF features with values less than 1/2 does not make sense. Charge density differences are a more rigorous approach.

RESPONSE: Thanks for the good comments. We totally agree with the reviewer that charge density differences, which have been used in our work, are a more rigorous approach. In ferroelectric system, however, the electron localization function (ELF) has also been actively employed to understand the ferroelectric and chemical properties, which indicates the degree of localization of the electronic density in a material ¹. As the reviewer mentioned, ELF = 0.5 corresponds to the electron gas, while ELF = 1 corresponds to perfect localization. Although the ELF less than 0.5 means low electron density, it can also effectively reveal the characteristics of electron distribution, which has been demonstrated and employed in the investigations of bonding and polarization properties.

In terms of bonding properties, ELF values less than 0.5 are used to analyze the hybridization and covalent degree of bonding. For example, in BiFeO₃ and GaFeO₃ materials, the finite ELF values of about 0.3~0.4 between A-site atoms and oxygen atoms can indicate the hybridization interaction and some degree of covalent characteristics ^{2,3}. In addition, the minimum ELF value in the interaction region can be used to determine the strength of bonding, where the ELF value is usually much smaller than 0.5 ^{4,5}.

In terms of polarization characteristics, materials such as BiFeO₃ often use ELF isosurfaces visualized for a value of ~0.3 to display the asymmetric distribution of s-lone pair electrons of Bi³⁺ and other ions ^{6,7}. Such asymmetric distribution of charges leads to a structural distortion that gives rise to spontaneous polarization ⁸. Similarly, ELF can also reflect the distribution characteristics of d-orbital electrons. In CdPbO₃, the ELF around Cd ions, with values of about 0.2, exhibits a strong asymmetry along the [111]_c direction, indicating the contribution of Cd ions to polarization in the ferroelectric phase ⁹.

Hence, ELF values below 0.5 are helpful in revealing the bonding properties and polarization characteristics. In our work, the oxygen vacancies in the defect structures

are not involved in actual bonding, resulting in low ELF values near the vacant sites. In contrast to the Vo-KTN structure, Mn and Fe dopants induce obvious changes in ELF distribution, which is similar to the ELF distribution used to describe polarization in the references [6-9]. Therefore, we employed the ELF to investigate the defect dipole and discuss the effect of defect structure on polarization. As illustrated in ELF, the localized charge distribution around the defect dipole in the Fe-doped crystal is clearly more polarized, indicating a stronger defect polarization.

In order to demonstrate the feasibility of the low ELF (less than 0.5) in describing the electronic structures, we have cited relevant references and added descriptions in line 402 in the revised manuscript, as follows: “The ELF distribution can reveal polarization characteristics^{43,44}.”

New references [43,44] have been cited in the revised manuscript:

43 Xi, G. *et al.* Anion-induced robust ferroelectricity in sulfurized pseudo-rhombohedral epitaxial BiFeO₃ thin films via polarization rotation. *Materials Horizons* **10**, 4389-4397, (2023).

44 Xu, Y., Hao, X., Franchini, C. & Gao F. Structural, electronic, and ferroelectric properties of compressed CdPbO₃ polymorphs. *Inorganic Chemistry* **52**, 1032-1039, (2013).

2. The authors now mention that they have used spin polarization, but they do not give more detailed description of the spin states that they end up using. This is needed for reproducibility of the work. For the systems illustrated in Figure 4, the authors need to describe the spin state (magnetic moment of different atoms, overall magnetic moment of the system). This can be done in the Supporting information.

RESPONSE: We sincerely appreciate the valuable suggestion. For the sake of the reproducibility of the work, we have added the overall magnetic moments of the systems and the magnetic moments of different atoms to the supporting information, as suggested by the reviewer. The pristine KTN crystal is non-magnetic, with each atom having a magnetic moment of zero. The overall magnetic moments of KTN with only the oxygen vacancy, Fe-doped KTN, and Mn-doped KTN are not zero, as listed in Table

S4. The magnetic moments of each atom in these structures are listed in Tables S(5-7), respectively. In addition to Tables S(4-7) in the supplementary information, the relevant description has been added to the method section, as follows: “The structural information, overall magnetic moments of the systems, and the magnetic moments of each atom are listed in the supplementary information.”

Table S4. Total magnetic moments $\mu_{\text{tot}} (\mu_B)$ of the simulated structures.

System	$\mu_{\text{tot}} (\mu_B)$
Pristine KTN	0
KTN with the oxygen vacancy	1.634
Fe-doped KTN	2.873
Mn-doped KTN	3.798

Table S5. Magnetic moment $\mu (\mu_B)$ of each atom of pristine KTN with the oxygen vacancy. Here, the atoms are listed in the same order as the atoms in the POSCAR file shown in Appendix S2. In contrast, the pristine KTN crystal is non-magnetic, with each atom having a magnetic moment of zero.

atom	$\mu(\mu_B)$	atom	$\mu(\mu_B)$	atom	$\mu(\mu_B)$	atom	$\mu(\mu_B)$	atom	$\mu(\mu_B)$	atom	$\mu(\mu_B)$	atom	$\mu(\mu_B)$	atom	$\mu(\mu_B)$
K	0.001	K	0.001	Nb	0.054	O	-0.004	O	-0.002	O	-0.005	O	-0.01	O	-
K	0	K	0.001	Nb	0.055	O	-0.003	O	-0.001	O	-0.005	O	-0.008	O	0.001
K	0.001	K	0	Nb	0.052	O	-0.008	O	-0.001	O	-0.002	O	-0.001	O	0.002
K	0.001	K	0	Nb	0.016	O	-0.002	O	-0.004	O	-0.002	O	-0.002	O	0.002
K	0.001	K	0.001	Nb	0.053	O	-0.008	O	-0.001	O	-0.002	O	-0.005	O	-0.01
K	0.001	K	0	Nb	0.017	O	-0.008	O	-0.004	O	-0.001	O	-0.005	O	0.002
K	0.001	K	0.001	Nb	0.019	O	-0.008	O	-0.004	O	-0.002	O	-0.002	O	0.002
K	0	K	0.001	Nb	0.022	O	-0.001	O	-0.003	O	-0.007	O	-0.005	Ta	0.01
K	0	K	0	Nb	0.051	O	-0.002	O	-0.002	O	-0.002	O	-0.002	Ta	0.009
K	0.001	K	0.001	Nb	0.111	O	-0.002	O	-0.001	O	-0.011	O	-0.002	Ta	0.011
K	0.001	K	0.001	Nb	0.021	O	-0.002	O	-0.002	O	-0.002	O	-0.002	Ta	0.019
K	0.001	K	0.001	Nb	0.094	O	-0.001	O	-0.005	O	-0.002	O	-0.002	Ta	0.065
K	0	K	0.001	Nb	0.017	O	-0.002	O	-0.001	O	-0.002	O	-0.005	Ta	0.045
K	0.001	K	0.001	Nb	0.102	O	-0.001	O	-0.001	O	-0.002	O	-0.009	Ta	0.027
K	0	K	0.001	Nb	0.016	O	-0.002	O	-0.001	O	-0.006	O	-0.002	Ta	0.057
K	0.001	K	0	Nb	0.02	O	-0.002	O	-0.001	O	-0.011	O	-0.008	Ta	0.009
K	0	K	0.001	O	-0.002	O	-0.003	O	-0.001	O	-0.002	O	-0.002	Ta	0.008
K	0	K	0.001	O	-0.001	O	-0.002	O	-0.001	O	-0.005	O	-0.01	Ta	0.028
K	0	K	0.001	O	-0.002	O	-0.008	O	-0.001	O	-0.005	O	-0.001	Ta	0.01
K	0.001	K	0.001	O	-0.001	O	-0.004	O	-0.004	O	-0.002	O	-0.01	Ta	0.008

K	0	K	0.001	O	-0.002	O	-0.001	O	-0.001	O	-0.011	O	-0.001	Ta	0.01
K	0.001	K	0	O	-0.001	O	-0.001	O	-0.002	O	-0.007	O	-0.002	Ta	0.028
K	0.001	K	0.001	O	-0.004	O	-0.005	O	-0.001	O	-0.002	O	-0.002	Ta	0.042
K	0	K	0.001	O	-0.005	O	-0.006	O	-0.003	O	-0.002	O	-0.001	Ta	0.01
K	0	Nb	0.021	O	-0.004	O	-0.002	O	-0.004	O	-0.002	O	-0.002	Ta	0.027
K	0.001	Nb	0.019	O	-0.003	O	-0.009	O	-0.002	O	-0.002	O	-0.005	Ta	0.008
K	0	Nb	0.069	O	-0.004	O	-0.001	O	-0.001	O	-0.005	O	-0.005	Ta	0.009
K	0.001	Nb	0.112	O	-0.001	O	-0.001	O	-0.001	O	-0.002	O	-0.002	Ta	0.028
K	0	Nb	0.021	O	-0.001	O	-0.001	O	-0.002	O	-0.005	O	-0.002	Ta	0.027
K	0	Nb	0.111	O	-0.001	O	-0.009	O	-0.004	O	-0.002	O	-0.001	Ta	0.026
K	0.001	Nb	0.019	O	-0.001	O	-0.002	O	-0.001	O	-0.005	O	-0.001	Ta	0.009
K	0.001	Nb	0.018	O	-0.001	O	-0.001	O	-0.002	O	-0.011	O	-0.002	Ta	0.011
K	0.001	Nb	0.056	O	-0.001	O	-0.002	O	-0.007	O	-0.002	O	-0.009	Ta	0.053
K	0	Nb	0.017	O	-0.001	O	-0.002	O	-0.01	O	-0.002	O	-0.001	Ta	0.011
K	0.001	Nb	0.051	O	-0.002	O	-0.004	O	-0.002	O	-0.005	O	-0.002	Ta	0.029
K	0	Nb	0.015	O	-0.001	O	-0.004	O	-0.005	O	-0.002	O	-0.002	Ta	0.042
K	0.001	Nb	0.017	O	-0.001	O	-0.004	O	-0.002	O	-0.005	O	-0.001	Ta	0.011
K	0	Nb	0.069	O	-0.002	O	-0.002	O	-0.005	O	-0.005	O	-0.002	Ta	0.009
K	0.001	Nb	0.048	O	-0.001	O	-0.001	O	-0.002	O	-0.001	O	-0.01	Ta	0.052
K	0	Nb	0.055	O	-0.001	O	-0.001	O	-0.001	O	-0.002	O	-0.001	Total	1.634

Table S6. Magnetic moment μ (μ_B) of each atom of Fe-doped KTN. Here, the atoms are listed in the same order as the atoms in the POSCAR file shown in Appendix S3.

atom	$\mu(\mu_B)$	atom	$\mu(\mu_B)$	atom	$\mu(\mu_B)$	atom	$\mu(\mu_B)$	atom	$\mu(\mu_B)$	atom	$\mu(\mu_B)$	atom	$\mu(\mu_B)$	atom	$\mu(\mu_B)$
Fe	2.711	K	0	Nb	0.003	O	0	O	0	O	0	O	0	O	0.001
K	0	K	0	Nb	0.001	O	0	O	0	O	0	O	0.001	O	0
K	0	K	0	Nb	0	O	0	O	0	O	0	O	0	O	0
K	0	K	0	Nb	0	O	0	O	0	O	0	O	0	O	0
K	0	K	0	Nb	0	O	0	O	0	O	0	O	0	O	0.002
K	0.001	K	0	Nb	0.001	O	0.001	O	0	O	0	O	0	O	0
K	0	K	0	Nb	0	O	0	O	0	O	0	O	0	O	0
K	0	K	0	Nb	0	O	0	O	0	O	0	O	0	Ta	0
K	0	K	0	Nb	0	O	0	O	0	O	0	O	0	Ta	0
K	0	K	0	Nb	0	O	0	O	0	O	0	O	0	Ta	0
K	0	K	0	Nb	0.001	O	0	O	0	O	0	O	0	Ta	0
K	0	K	0	Nb	-0.002	O	0	O	0.003	O	0	O	0	Ta	0.016
K	0	K	0	Nb	0.001	O	0	O	0.002	O	0	O	0	Ta	0
K	0	K	0	Nb	0	O	0	O	0	O	0.001	O	0	Ta	0
K	0	K	0	Nb	0.001	O	0	O	0	O	0	O	0	Ta	0
K	0	K	0	Nb	0	O	0	O	0	O	0	O	0.002	Ta	0
K	0	K	0	O	0	O	0	O	0	O	0	O	0	Ta	0
K	0	K	0	O	-0.001	O	0	O	0	O	0	O	0	Ta	0
K	0	K	0	O	0	O	0.001	O	0	O	0	O	0	Ta	0

K	0	K	-0.001	O	0	O	0	O	0	O	0	O	0	Ta	0
K	0	K	0	O	0	O	0	O	-0.001	O	0	O	0	Ta	0
K	0	K	0	O	0	O	0	O	0.095	O	0	O	0	Ta	0
K	0	K	0	O	0	O	0	O	0	O	0	O	0	Ta	-0.006
K	0	K	0	O	0	O	0.074	O	0	O	0	O	0	Ta	0
K	0	K	0	O	0	O	0	O	0	O	0	O	0	Ta	0
K	0	Nb	0	O	0	O	0	O	0	O	0	O	-0.021	Ta	0
K	0	Nb	0	O	0	O	0	O	0	O	0	O	-0.018	Ta	0.002
K	0	Nb	-0.001	O	0	O	0.001	O	0	O	0	O	0	Ta	0
K	0	Nb	0	O	0	O	0	O	0	O	0	O	0	Ta	0
K	0	Nb	-0.001	O	0	O	0	O	0	O	0	O	0	Ta	0
K	0	Nb	0	O	0	O	0	O	0	O	0	O	0	Ta	0
K	0	Nb	0.002	O	0.008	O	0	O	0	O	0	O	0	Ta	0
K	0	Nb	0	O	0	O	0	O	0.001	O	0	O	0	Ta	0
K	0	Nb	0	O	0.001	O	0	O	0	O	0	O	0	Ta	0
K	0	Nb	0	O	0	O	0	O	0	O	0	O	0	Ta	0.006
K	0	Nb	-0.002	O	0	O	0	O	0	O	0	O	0	Ta	-0.006
K	0	Nb	0	O	0	O	0	O	0	O	0	O	0.001	Ta	0
K	0	Nb	0	O	0	O	0.001	O	0.001	O	0	O	0	Ta	0
K	0	Nb	-0.002	O	0	O	0	O	0.002	O	0.001	O	0	Ta	0
K	-0.001	Nb	0	O	0	O	0	O	0	O	0.001	O	0	Total	2.873

Table S7. Magnetic moment μ (μ_B) of each atom of Mn-doped KTN. Here, the atoms are listed in the same order as the atoms in the POSCAR file shown in Appendix S4.

atom	$\mu(\mu_B)$	atom	$\mu(\mu_B)$	atom	$\mu(\mu_B)$	atom	$\mu(\mu_B)$	atom	$\mu(\mu_B)$	atom	$\mu(\mu_B)$	atom	$\mu(\mu_B)$	atom	$\mu(\mu_B)$
Mn	3.667	K	0	Nb	0	O	0	O	0	O	0	O	0	O	0
K	0	K	0	Nb	0.002	O	0	O	0	O	0.001	O	0	O	0
K	0	K	0	Nb	0	O	0	O	0	O	0	O	0	O	0
K	0	K	0	Nb	0	O	0	O	0	O	0	O	0	O	0
K	0	K	0	Nb	0	O	0	O	0	O	0	O	0	O	0.001
K	0	K	0	Nb	0.001	O	0	O	0	O	0	O	0	O	0
K	0	K	0	Nb	0	O	0	O	0	O	0.001	O	0	O	0
K	0	K	0	Nb	0	O	0	O	0	O	0	O	0	Ta	0
K	0	K	0	Nb	0.002	O	0	O	0	O	0	O	0	Ta	0
K	0	K	0	Nb	0	O	0	O	0	O	0	O	0	Ta	0
K	0	K	0	Nb	0.002	O	0	O	0	O	0	O	0	Ta	0
K	0	K	-0.001	Nb	0	O	0	O	0	O	0	O	0	Ta	0
K	0	K	0	Nb	-0.001	O	0	O	0	O	0	O	0	Ta	0
K	0	K	0	Nb	0.001	O	0	O	0.003	O	0	O	0	Ta	0
K	0	K	0	Nb	-0.001	O	0	O	0.003	O	0	O	0	Ta	0
K	0	K	0	Nb	0	O	0	O	0	O	0	O	0	Ta	0.026
K	0	K	0	O	0	O	0	O	0	O	0	O	0	Ta	0
K	0	K	0	O	-0.007	O	0	O	0	O	0	O	0	Ta	0

K	0	K	0	O	0.002	O	0	O	0	O	0	O	0.002	Ta	0
K	0	K	0	O	0	O	0	O	0	O	0	O	0	Ta	0
K	0	K	0.001	O	0	O	0	O	0	O	0	O	0	Ta	0
K	0	K	0	O	0	O	0	O	0	O	0	O	0	Ta	0
K	0	K	0	O	0	O	0	O	0	O	0	O	0	Ta	0
K	0	K	0	O	0	O	0	O	0.001	O	0	O	0	Ta	0
K	0	K	0	O	0	O	0.067	O	0.002	O	0	O	0.004	Ta	0.005
K	0	Nb	0.002	O	0	O	0	O	0	O	0	O	0.004	Ta	0.005
K	0	Nb	0	O	0	O	0	O	0	O	0	O	0	Ta	0
K	0	Nb	0.002	O	0	O	0	O	0	O	0	O	0	Ta	0
K	0	Nb	0.002	O	0	O	0	O	0	O	0	O	0.002	Ta	0.005
K	0	Nb	0	O	0	O	0	O	0	O	0	O	0	Ta	0
K	0	Nb	0	O	0	O	0.001	O	0	O	0	O	0	Ta	0
K	0	Nb	0	O	0	O	0	O	0	O	0	O	0	Ta	0
K	0	Nb	0	O	0	O	0	O	0.003	O	0	O	0.001	Ta	0
K	0	Nb	0	O	0	O	0	O	0	O	0	O	0	Ta	0
K	0	Nb	0	O	0	O	0	O	0.003	O	0	O	0	Ta	0
K	0	Nb	0.002	O	0	O	0	O	0	O	0	O	0	Ta	0.004
K	0	Nb	0	O	0	O	0	O	0	O	0	O	0	Ta	0
K	0	Nb	0	O	0	O	0	O	0	O	0	O	0	Ta	0
K	0	Nb	0	O	0	O	0	O	0	O	0.001	O	0.002	Ta	0
K	0	Nb	0	O	0	O	0	O	0	O	0.001	O	0	Total	3.798

3. Related to point (2), CIF or POSCAR files of the used DFT models need to be given as Supporting information. This is crucial for reproducibility of the computational work.

RESPONSE: Thanks for the valuable suggestion. According to this suggestion, we have added the POSCAR files of the used DFT models in supporting information for the reproducibility of the computational work. Appendices S(1-6) show the structural information of pristine KTN, KTN with only the oxygen vacancy, Fe-doped KTN, and Mn-doped KTN, respectively.

In addition to Appendices S(1-6) in the supplementary information, the relevant description has been added to the method section, as follows: “The structural information, overall magnetic moments of the systems, and the magnetic moments of each atom are listed in the supplementary information.”

Response to Reviewer #3

The author has solved all the problems. This is meaningful work.

RESPONSE: We sincerely thank the reviewer for the positive and encouraging comments.

References

- 1 Xi, G. *et al.* Anion-induced robust ferroelectricity in sulfurized pseudo-rhombohedral epitaxial BiFeO₃ thin films via polarization rotation. *Materials Horizons* **10**, 4389-4397, (2023).
- 2 Ravindran, P., Vidya, R., Kjekshus, A. & Fjellvåg, H. Theoretical investigation of magnetoelectric behavior in BiFeO₃. *Physical Review B* **74**, 224412, (2006).
- 3 Roy, A. *et al.* Electronic structure, Born effective charges and spontaneous polarization in magnetoelectric gallium ferrite. *Journal Of Physics: Condensed Matter* **23**, 325902, (2011).
- 4 Liu, C. *et al.* Boosting upconversion photoluminescence and multielectrical properties via Er-doping-modulated vacancy control in Ba_{0.85}Ca_{0.15}Ti_{0.9}Zr_{0.1}O₃. *ACS Omega* **4**, 11004-11013, (2019).
- 5 Koumpouras, K. & Larsson, J. A. Distinguishing between chemical bonding and physical binding using electron localization function (ELF). *Journal Of Physics: Condensed Matter* **32**, 315502, (2020).
- 6 Saha, R. A. *et al.* The critical role of stereochemically active lone pair in introducing high temperature ferroelectricity. *Inorganic Chemistry* **60**, 4068-4075, (2021).
- 7 Diéguez, O, González-Vázquez, O. E., Wojdeł J. C. & Íñiguez J. First-principles predictions of low-energy phases of multiferroic BiFeO₃. *Physical Review B* **83**, 094105, (2011).
- 8 Martínez-Aguilar, E., Hmök, H., Ariño, J. R. & Beltrones J. M. S. First-principles study of the coexisting ferroelectric and ferromagnetic properties of the La_{0.75}Bi_{0.25}CrO₃ compound. *Computational Materials Science* **171**, 109262, (2020).
- 9 Xu, Y., Hao, X., Franchini, C. & Gao F. Structural, electronic, and ferroelectric properties of compressed CdPbO₃ polymorphs. *Inorganic Chemistry* **52**, 1032-1039, (2013).

Appendix S1. POSCAR file of the 4×4×4 supercell of pristine KTN

```
=====  
=====  
KTN  
1.0  
15.9399995804      0.0000000000      0.0000000000  
0.0000000000      15.9695816040      0.0000000000  
0.0000000000      0.0000000000      15.9695816040  
K   Nb   O   Ta  
64  32  192  32  
Direct  
0.374976000      0.373805000      0.126195000  
0.374976000      0.623805000      0.376195000  
0.374976000      0.623805000      0.876195000  
0.874976000      0.123805000      0.376195000  
0.874976000      0.123805000      0.876195000  
0.874976000      0.623805000      0.376195000  
0.125024000      0.123805000      0.376195000  
0.125024000      0.123805000      0.876195000  
0.125024000      0.623805000      0.376195000  
0.125024000      0.623805000      0.876195000  
0.625024000      0.123805000      0.376195000  
0.625024000      0.123805000      0.876195000  
0.625024000      0.623805000      0.376195000  
0.625024000      0.623805000      0.876195000  
0.124976000      0.123805000      0.126195000  
0.124976000      0.123805000      0.626195000  
0.124976000      0.623805000      0.126195000  
0.124976000      0.623805000      0.626195000  
0.624976000      0.123805000      0.126195000  
0.624976000      0.123805000      0.626195000  
0.624976000      0.623805000      0.126195000  
0.624976000      0.623805000      0.626195000  
0.624976000      0.623805000      0.626195000  
0.375024000      0.123805000      0.126195000  
0.375024000      0.123805000      0.626195000  
0.375024000      0.623805000      0.126195000  
0.375024000      0.623805000      0.626195000  
0.875024000      0.123805000      0.126195000  
0.875024000      0.123805000      0.626195000  
0.875024000      0.623805000      0.126195000  
0.875024000      0.623805000      0.626195000  
0.374976000      0.123805000      0.876195000  
0.374976000      0.123805000      0.376195000  
0.874976000      0.623805000      0.876195000  
0.875024000      0.873805000      0.376195000  
0.374976000      0.373805000      0.626195000  
0.374976000      0.873805000      0.126195000  
0.374976000      0.873805000      0.626195000  
0.874976000      0.373805000      0.126195000  
0.874976000      0.373805000      0.626195000  
0.874976000      0.873805000      0.126195000  
0.874976000      0.873805000      0.626195000  
0.874976000      0.873805000      0.626195000  
0.125024000      0.373805000      0.126195000  
0.875024000      0.873805000      0.876195000  
0.125024000      0.873805000      0.126195000  
0.125024000      0.873805000      0.626195000
```

0.625024000	0.373805000	0.126195000
0.625024000	0.373805000	0.626195000
0.625024000	0.873805000	0.126195000
0.625024000	0.873805000	0.626195000
0.125024000	0.373805000	0.626195000
0.124976000	0.373805000	0.876195000
0.124976000	0.373805000	0.376195000
0.875024000	0.373805000	0.876195000
0.875024000	0.373805000	0.376195000
0.375024000	0.873805000	0.876195000
0.375024000	0.873805000	0.376195000
0.375024000	0.373805000	0.376195000
0.375024000	0.373805000	0.876195000
0.624976000	0.873805000	0.376195000
0.624976000	0.373805000	0.876195000
0.624976000	0.373805000	0.376195000
0.124976000	0.873805000	0.876195000
0.124976000	0.873805000	0.376195000
0.624976000	0.873805000	0.876195000
0.000000000	0.997551000	0.252449000
0.500000000	0.997551000	0.252449000
0.000000000	0.997551000	0.752449000
0.500000000	0.497551000	0.252449000
0.500000000	0.497551000	0.752449000
0.500000000	0.997551000	0.752449000
0.750000000	0.997551000	0.002449000
0.250000000	0.497551000	0.502449000
0.250000000	0.997551000	0.002449000
0.250000000	0.997551000	0.502449000
0.750000000	0.497551000	0.002449000
0.750000000	0.497551000	0.502449000
0.000000000	0.497551000	0.752449000
0.250000000	0.497551000	0.002449000
0.000000000	0.497551000	0.252449000
0.750000000	0.997551000	0.502449000
0.750000000	0.747551000	0.252449000
0.750000000	0.747551000	0.752449000
0.000000000	0.247551000	0.002449000
0.000000000	0.247551000	0.502449000
0.000000000	0.747551000	0.502449000
0.500000000	0.247551000	0.002449000
0.500000000	0.247551000	0.502449000
0.500000000	0.747551000	0.002449000
0.000000000	0.747551000	0.002449000
0.250000000	0.247551000	0.252449000
0.250000000	0.247551000	0.752449000
0.250000000	0.747551000	0.252449000
0.250000000	0.747551000	0.752449000
0.750000000	0.247551000	0.252449000
0.500000000	0.747551000	0.502449000
0.750000000	0.247551000	0.752449000
0.375552000	0.750889000	0.249111000
0.375552000	0.250889000	0.749111000
0.375552000	0.250889000	0.249111000
0.874448000	0.250889000	0.999111000
0.874448000	0.750889000	0.499111000
0.874448000	0.250889000	0.499111000

0.374448000	0.750889000	0.999111000
0.375552000	0.750889000	0.749111000
0.874448000	0.750889000	0.999111000
0.875552000	0.250889000	0.249111000
0.124448000	0.750889000	0.749111000
0.875552000	0.750889000	0.249111000
0.875552000	0.750889000	0.749111000
0.124448000	0.250889000	0.249111000
0.124448000	0.250889000	0.749111000
0.124448000	0.750889000	0.249111000
0.374448000	0.750889000	0.499111000
0.624448000	0.250889000	0.249111000
0.624448000	0.250889000	0.749111000
0.624448000	0.750889000	0.249111000
0.875552000	0.250889000	0.749111000
0.374448000	0.250889000	0.999111000
0.125552000	0.750889000	0.499111000
0.625552000	0.750889000	0.999111000
0.000000000	0.876174000	0.499291000
0.000000000	0.876174000	0.999291000
0.500000000	0.376174000	0.499291000
0.500000000	0.376174000	0.999291000
0.500000000	0.876174000	0.499291000
0.500000000	0.876174000	0.999291000
0.125552000	0.000889000	0.249111000
0.125552000	0.000889000	0.749111000
0.125552000	0.500889000	0.249111000
0.125552000	0.500889000	0.749111000
0.625552000	0.000889000	0.249111000
0.625552000	0.000889000	0.749111000
0.625552000	0.500889000	0.249111000
0.625552000	0.500889000	0.749111000
0.125552000	0.250889000	0.499111000
0.125552000	0.250889000	0.999111000
0.624448000	0.750889000	0.749111000
0.125552000	0.750889000	0.999111000
0.625552000	0.250889000	0.499111000
0.625552000	0.250889000	0.999111000
0.625552000	0.750889000	0.499111000
0.374448000	0.250889000	0.499111000
0.250000000	0.250749000	0.374957000
0.374448000	0.000889000	0.249111000
0.250000000	0.750749000	0.374957000
0.624448000	0.000889000	0.999111000
0.624448000	0.500889000	0.499111000
0.624448000	0.500889000	0.999111000
0.000000000	0.000709000	0.123826000
0.000000000	0.000709000	0.623826000
0.000000000	0.500709000	0.123826000
0.000000000	0.500709000	0.623826000
0.500000000	0.000709000	0.123826000
0.500000000	0.000709000	0.623826000
0.624448000	0.000889000	0.499111000
0.500000000	0.500709000	0.123826000
0.250000000	0.375043000	0.499251000
0.250000000	0.375043000	0.999251000
0.250000000	0.875043000	0.499251000

0.250000000	0.875043000	0.999251000
0.750000000	0.375043000	0.499251000
0.750000000	0.375043000	0.999251000
0.750000000	0.875043000	0.499251000
0.750000000	0.875043000	0.999251000
0.000000000	0.376174000	0.999291000
0.500000000	0.500709000	0.623826000
0.124448000	0.500889000	0.999111000
0.124448000	0.500889000	0.499111000
0.124448000	0.000889000	0.999111000
0.250000000	0.750749000	0.874957000
0.750000000	0.250749000	0.374957000
0.750000000	0.250749000	0.874957000
0.750000000	0.750749000	0.374957000
0.750000000	0.750749000	0.874957000
0.374448000	0.000889000	0.749111000
0.374448000	0.500889000	0.249111000
0.374448000	0.500889000	0.749111000
0.874448000	0.000889000	0.249111000
0.874448000	0.000889000	0.749111000
0.874448000	0.500889000	0.249111000
0.874448000	0.500889000	0.749111000
0.375552000	0.000889000	0.499111000
0.375552000	0.000889000	0.999111000
0.375552000	0.500889000	0.499111000
0.375552000	0.500889000	0.999111000
0.875552000	0.000889000	0.499111000
0.875552000	0.000889000	0.999111000
0.875552000	0.500889000	0.499111000
0.875552000	0.500889000	0.999111000
0.124448000	0.000889000	0.499111000
0.250000000	0.250749000	0.874957000
0.000000000	0.376174000	0.499291000
0.750000000	0.750709000	0.623826000
0.750000000	0.500749000	0.124957000
0.000000000	0.375043000	0.749251000
0.000000000	0.875043000	0.249251000
0.000000000	0.875043000	0.749251000
0.500000000	0.375043000	0.249251000
0.500000000	0.375043000	0.749251000
0.500000000	0.875043000	0.249251000
0.750000000	0.500749000	0.624957000
0.000000000	0.125043000	0.499251000
0.000000000	0.125043000	0.999251000
0.000000000	0.375043000	0.249251000
0.000000000	0.625043000	0.499251000
0.500000000	0.125043000	0.499251000
0.500000000	0.125043000	0.999251000
0.500000000	0.625043000	0.499251000
0.500000000	0.625043000	0.999251000
0.000000000	0.000749000	0.374957000
0.000000000	0.000749000	0.874957000
0.000000000	0.500749000	0.374957000
0.000000000	0.500749000	0.874957000
0.500000000	0.000749000	0.374957000
0.000000000	0.625043000	0.999251000
0.500000000	0.750749000	0.624957000

0.50000000	0.75074900	0.12495700
0.50000000	0.25074900	0.62495700
0.25000000	0.00070900	0.37382600
0.25000000	0.00070900	0.87382600
0.25000000	0.50070900	0.37382600
0.25000000	0.50070900	0.87382600
0.75000000	0.00070900	0.37382600
0.75000000	0.00070900	0.87382600
0.75000000	0.50070900	0.37382600
0.75000000	0.50070900	0.87382600
0.25000000	0.12504300	0.24925100
0.25000000	0.12504300	0.74925100
0.25000000	0.62504300	0.24925100
0.25000000	0.62504300	0.74925100
0.75000000	0.12504300	0.24925100
0.75000000	0.12504300	0.74925100
0.75000000	0.62504300	0.24925100
0.75000000	0.62504300	0.74925100
0.00000000	0.25074900	0.12495700
0.00000000	0.25074900	0.62495700
0.00000000	0.75074900	0.12495700
0.00000000	0.75074900	0.62495700
0.50000000	0.25074900	0.12495700
0.50000000	0.00074900	0.87495700
0.50000000	0.50074900	0.37495700
0.50000000	0.87504300	0.74925100
0.25000000	0.12617400	0.49929100
0.00000000	0.75070900	0.37382600
0.00000000	0.75070900	0.87382600
0.50000000	0.25070900	0.37382600
0.50000000	0.25070900	0.87382600
0.50000000	0.75070900	0.37382600
0.50000000	0.75070900	0.87382600
0.25000000	0.37617400	0.24929100
0.25000000	0.37617400	0.74929100
0.25000000	0.87617400	0.24929100
0.00000000	0.25070900	0.87382600
0.25000000	0.87617400	0.74929100
0.75000000	0.37617400	0.74929100
0.50000000	0.50074900	0.87495700
0.75000000	0.87617400	0.74929100
0.25000000	0.00074900	0.12495700
0.25000000	0.00074900	0.62495700
0.25000000	0.50074900	0.12495700
0.25000000	0.50074900	0.62495700
0.75000000	0.00074900	0.12495700
0.75000000	0.00074900	0.62495700
0.75000000	0.37617400	0.24929100
0.00000000	0.25070900	0.37382600
0.75000000	0.87617400	0.24929100
0.50000000	0.62617400	0.24929100
0.50000000	0.62617400	0.74929100
0.25000000	0.12617400	0.99929100
0.25000000	0.62617400	0.49929100
0.25000000	0.62617400	0.99929100
0.75000000	0.12617400	0.99929100
0.75000000	0.62617400	0.49929100

0.750000000	0.626174000	0.999291000
0.250000000	0.250709000	0.123826000
0.250000000	0.250709000	0.623826000
0.250000000	0.750709000	0.123826000
0.750000000	0.126174000	0.499291000
0.750000000	0.250709000	0.123826000
0.750000000	0.250709000	0.623826000
0.750000000	0.750709000	0.123826000
0.000000000	0.126174000	0.249291000
0.000000000	0.126174000	0.749291000
0.000000000	0.626174000	0.249291000
0.000000000	0.626174000	0.749291000
0.500000000	0.126174000	0.249291000
0.250000000	0.750709000	0.623826000
0.500000000	0.126174000	0.749291000
0.250000000	0.497636000	0.752364000
0.250000000	0.997636000	0.252364000
0.250000000	0.997636000	0.752364000
0.750000000	0.497636000	0.252364000
0.750000000	0.497636000	0.752364000
0.750000000	0.997636000	0.252364000
0.000000000	0.997636000	0.002364000
0.000000000	0.497636000	0.002364000
0.000000000	0.497636000	0.502364000
0.000000000	0.997636000	0.502364000
0.500000000	0.497636000	0.002364000
0.250000000	0.497636000	0.252364000
0.500000000	0.497636000	0.502364000
0.750000000	0.997636000	0.752364000
0.500000000	0.747636000	0.752364000
0.750000000	0.247636000	0.002364000
0.500000000	0.247636000	0.752364000
0.500000000	0.247636000	0.252364000
0.000000000	0.747636000	0.752364000
0.000000000	0.747636000	0.252364000
0.000000000	0.247636000	0.752364000
0.000000000	0.247636000	0.252364000
0.750000000	0.747636000	0.502364000
0.750000000	0.747636000	0.002364000
0.750000000	0.247636000	0.502364000
0.250000000	0.747636000	0.502364000
0.250000000	0.747636000	0.002364000
0.250000000	0.247636000	0.502364000
0.250000000	0.247636000	0.002364000
0.500000000	0.997636000	0.002364000
0.500000000	0.747636000	0.252364000
0.500000000	0.997636000	0.502364000

Appendix S2. POSCAR file of the 4×4×4 supercell of KTN with only the oxygen vacancy

=====

Vo-KTN

1.0

15.9399995804	0.0000000000	0.0000000000
0.0000000000	15.9695816040	0.0000000000
0.0000000000	0.0000000000	15.9695816040

K	Nb	O	Ta
64	32	191	32

Direct

0.375406000	0.373620000	0.126644000
0.377045000	0.374345000	0.628412000
0.375158000	0.873644000	0.127131000
0.376534000	0.874245000	0.628271000
0.874907000	0.373597000	0.126780000
0.874815000	0.374066000	0.626897000
0.124594000	0.373620000	0.126644000
0.874891000	0.873708000	0.876935000
0.874911000	0.623698000	0.876844000
0.123466000	0.874245000	0.628271000
0.625093000	0.373597000	0.126780000
0.625185000	0.374066000	0.626897000
0.625141000	0.873678000	0.126924000
0.625100000	0.874064000	0.626843000
0.122955000	0.374345000	0.628412000
0.124842000	0.873644000	0.127131000
0.121768000	0.374041000	0.373414000
0.124922000	0.373687000	0.876721000
0.625109000	0.873708000	0.876935000
0.123418000	0.874031000	0.375587000
0.124919000	0.873755000	0.876993000
0.625324000	0.373964000	0.376469000
0.625104000	0.874029000	0.377057000
0.625020000	0.373620000	0.876658000
0.378232000	0.374040000	0.373415000
0.376582000	0.874031000	0.375587000
0.375081000	0.873755000	0.876993000
0.874676000	0.373964000	0.376469000
0.874980000	0.373620000	0.876658000
0.375078000	0.373687000	0.876721000
0.874896000	0.874029000	0.377057000
0.122999000	0.123143000	0.374804000
0.874900000	0.874064000	0.626843000
0.374978000	0.123846000	0.876654000
0.377311000	0.625235000	0.374818000
0.375048000	0.623638000	0.876762000
0.875065000	0.123967000	0.376882000
0.874895000	0.123660000	0.876799000
0.875055000	0.624008000	0.376921000
0.125022000	0.123846000	0.876654000
0.122689000	0.625235000	0.374818000
0.377001000	0.123143000	0.374804000
0.874859000	0.873678000	0.126924000

0.625105000	0.123660000	0.876799000
0.624945000	0.624008000	0.376921000
0.625089000	0.623698000	0.876844000
0.124899000	0.123689000	0.127071000
0.123275000	0.123288000	0.628212000
0.124952000	0.623638000	0.876762000
0.624935000	0.123967000	0.376882000
0.124889000	0.623514000	0.127226000
0.123098000	0.625287000	0.628431000
0.376725000	0.123288000	0.628212000
0.625153000	0.123603000	0.126839000
0.624958000	0.124033000	0.626782000
0.625134000	0.623738000	0.126888000
0.375101000	0.123689000	0.127071000
0.624942000	0.624043000	0.626838000
0.375111000	0.623514000	0.127226000
0.874847000	0.123603000	0.126839000
0.875042000	0.124033000	0.626782000
0.874866000	0.623738000	0.126888000
0.875058000	0.624044000	0.626838000
0.376902000	0.625288000	0.628431000
0.000000000	0.246599000	0.003892000
0.500070000	0.746669000	0.004196000
0.500617000	0.248824000	0.503704000
0.000000000	0.748279000	0.504085000
0.500087000	0.246599000	0.003892000
0.500218000	0.748279000	0.504085000
0.000000000	0.746669000	0.004196000
0.250000000	0.248368000	0.753470000
0.250000000	0.747480000	0.254440000
0.250000000	0.747772000	0.754580000
0.750000000	0.247264000	0.253950000
0.750000000	0.747264000	0.754232000
0.750000000	0.247194000	0.753992000
0.000000000	0.248824000	0.503704000
0.750000000	0.747400000	0.254254000
0.250000000	0.248178000	0.253093000
0.000000000	0.497619000	0.253710000
0.250000000	0.513048000	0.502120000
0.000000000	0.997422000	0.253821000
0.500286000	0.997377000	0.754325000
0.500466000	0.497619000	0.253710000
0.500169000	0.497459000	0.754133000
0.750000000	0.996659000	0.004138000
0.250000000	0.496887000	0.004118000
0.500378000	0.997422000	0.253821000
0.250000000	0.994910000	0.504015000
0.750000000	0.496716000	0.004119000
0.750000000	0.498308000	0.504028000
0.000000000	0.497459000	0.754133000
0.750000000	0.998260000	0.504105000
0.000000000	0.997377000	0.754325000
0.250000000	0.996864000	0.004464000
0.001025000	0.250848000	0.624243000
0.250000000	0.500310000	0.873075000
0.750000000	0.000914000	0.373687000
0.750000000	0.001070000	0.873604000

0.75000000	0.50092100	0.37375600
0.75000000	0.50108600	0.87353100
0.25000000	0.12588600	0.25260500
0.25000000	0.62342600	0.25263200
0.75000000	0.12492500	0.24870700
0.00026400	0.75065200	0.62452700
0.75000000	0.62486800	0.24865400
0.75000000	0.62498200	0.74852000
0.00048300	0.25094300	0.12464700
0.25000000	0.00152500	0.87362900
0.00020200	0.75100400	0.12467900
0.25000000	0.49112100	0.37709000
0.49951700	0.25094300	0.12464700
0.25000000	0.62420200	0.74557900
0.25000000	0.00103900	0.37453500
0.75000000	0.12506600	0.74854700
0.25000000	0.12538900	0.74560000
0.49985500	0.00091500	0.37460700
0.49979800	0.75100400	0.12467900
0.49984100	0.37506600	0.74864100
0.49971300	0.87480800	0.24864700
0.75000000	0.50097900	0.62436800
0.00326600	0.12566200	0.49844800
0.00026600	0.12516400	0.99860100
0.75000000	0.37471000	0.49841000
0.00308500	0.62375900	0.49845400
0.49673400	0.12566200	0.49844800
0.00012600	0.87496000	0.74857300
0.49973400	0.12516400	0.99860100
0.49973600	0.62513000	0.99862200
0.00014500	0.00091500	0.37460700
0.49981300	0.00101200	0.87475800
0.00073000	0.50095100	0.37464500
0.00055300	0.50115300	0.87461800
0.49927000	0.50095100	0.37464500
0.00026400	0.62513000	0.99862200
0.49973600	0.75065200	0.62452700
0.49897500	0.25084800	0.62424300
0.49691500	0.62375900	0.49845400
0.00028700	0.87480800	0.24864700
0.00018700	0.00101200	0.87475800
0.49981800	0.75110400	0.87376700
0.25000000	0.87576100	0.25009800
0.25000000	0.62764100	0.49942100
0.75000000	0.12633800	0.99855900
0.75000000	0.62568500	0.49846100
0.25000000	0.25055700	0.12454300
0.25000000	0.25826200	0.61975300
0.25000000	0.74945500	0.12449600
0.75000000	0.12625500	0.49846100
0.25000000	0.12630300	0.99892100
0.75000000	0.25109400	0.12356000
0.75000000	0.62623600	0.99852700
0.25000000	0.62590400	0.99887100
0.00009700	0.12606200	0.24848800
0.00007400	0.62607500	0.24848400
0.49990300	0.12606200	0.24848800

0.25000000	0.75068800	0.62297800
0.49998700	0.12618900	0.74858900
0.00015900	0.37506600	0.74864100
0.75000000	0.25094400	0.62336800
0.75000000	0.75103000	0.12359500
0.49999700	0.62622000	0.74862400
0.49999500	0.37495000	0.24851200
0.00001300	0.12618900	0.74858900
0.75000000	0.87607800	0.24867300
0.49992600	0.62607500	0.24848400
0.25000000	0.00000000	0.62391500
0.49940400	0.25070200	0.37353300
0.49951400	0.25101700	0.87356900
0.49981600	0.75067700	0.37360200
0.25000000	0.37578600	0.24769700
0.25000000	0.37591300	0.74965200
0.00048600	0.25101700	0.87356900
0.25000000	0.87595000	0.74817400
0.75000000	0.37627100	0.74848800
0.00018200	0.75110400	0.87376700
0.49944700	0.50115300	0.87461800
0.25000000	0.00178200	0.12547300
0.25000000	0.12336700	0.49983800
0.25000000	0.50000800	0.12574900
0.25000000	0.49238600	0.62074700
0.75000000	0.00097700	0.12461400
0.75000000	0.00095400	0.62454000
0.75000000	0.37619500	0.24863300
0.00059600	0.25070200	0.37353300
0.49987400	0.87496000	0.74857300
0.75000000	0.87618900	0.74851100
0.00018400	0.75067700	0.37360200
0.00000500	0.37495000	0.24851200
0.75000000	0.50098100	0.12462400
0.00021200	0.37634800	0.99855500
0.00000000	0.37589500	0.49818600
0.49924900	0.87592000	0.49864100
0.49977400	0.87625400	0.99863000
0.12586800	0.00089800	0.24844600
0.75000000	0.75086200	0.62353200
0.12600800	0.50068800	0.24937100
0.12590000	0.50101800	0.74749300
0.00000300	0.62622000	0.74862400
0.62556400	0.00079800	0.24821700
0.62553600	0.50079500	0.24842200
0.49978800	0.37634800	0.99855500
0.62544500	0.50088200	0.74782900
0.12591400	0.25142300	0.99845100
0.62432600	0.75095100	0.74823600
0.37081500	0.25748600	0.49762200
0.62483100	0.25030200	0.49864800
0.62546600	0.25141900	0.99828700
0.62518500	0.74942500	0.49832400
0.62553700	0.00102500	0.74818700
0.62550000	0.75121100	0.99835500
0.37408600	0.25142300	0.99845100
0.12918500	0.25748600	0.49762200

0.500478000	0.375895000	0.498186000
0.126507000	0.750373000	0.498227000
0.125866000	0.751207000	0.998525000
0.875679000	0.250898000	0.248339000
0.375301000	0.250742000	0.249014000
0.874534000	0.251419000	0.998287000
0.874815000	0.749425000	0.498324000
0.875169000	0.250302000	0.498648000
0.374134000	0.751207000	0.998525000
0.375450000	0.750710000	0.748328000
0.874500000	0.751211000	0.998355000
0.124550000	0.750710000	0.748328000
0.624321000	0.250898000	0.248339000
0.875674000	0.750951000	0.748236000
0.124699000	0.250741000	0.249013000
0.124604000	0.250844000	0.747572000
0.124636000	0.750346000	0.248478000
0.373493000	0.750373000	0.498227000
0.375396000	0.250844000	0.747572000
0.624227000	0.251146000	0.747879000
0.624334000	0.750757000	0.248264000
0.875773000	0.251146000	0.747879000
0.875666000	0.750757000	0.248264000
0.375364000	0.750346000	0.248478000
0.250000000	0.259019000	0.378050000
0.374100000	0.501018000	0.747493000
0.000751000	0.875920000	0.498641000
0.623568000	0.500318000	0.498812000
0.750000000	0.751048000	0.874663000
0.374240000	0.001025000	0.748302000
0.373992000	0.500688000	0.249371000
0.874436000	0.000798000	0.248217000
0.874463000	0.001025000	0.748187000
0.874464000	0.500795000	0.248422000
0.750000000	0.250868000	0.374778000
0.874555000	0.500882000	0.747829000
0.375356000	0.001297000	0.998484000
0.750000000	0.750820000	0.374720000
0.374132000	0.000898000	0.248446000
0.876167000	0.001314000	0.498359000
0.250000000	0.749734000	0.375590000
0.876432000	0.500318000	0.498812000
0.875775000	0.501262000	0.998329000
0.125273000	0.000030000	0.498041000
0.250000000	0.250279000	0.874092000
0.374727000	0.000030000	0.498041000
0.250000000	0.749652000	0.874690000
0.125760000	0.001025000	0.748302000
0.375246000	0.501319000	0.998454000
0.750000000	0.251046000	0.874586000
0.875706000	0.001334000	0.998313000
0.127762000	0.493827000	0.497857000
0.372238000	0.493827000	0.497857000
0.624225000	0.501262000	0.998329000
0.000382000	0.001082000	0.623494000
0.000625000	0.501078000	0.123652000
0.499833000	0.001025000	0.123627000

0.499618000	0.001082000	0.623494000
0.623833000	0.001314000	0.498359000
0.000167000	0.001025000	0.123627000
0.624294000	0.001334000	0.998313000
0.250000000	0.375064000	0.998861000
0.001260000	0.500881000	0.623212000
0.750000000	0.375233000	0.998518000
0.750000000	0.874662000	0.498621000
0.750000000	0.875112000	0.998597000
0.498740000	0.500881000	0.623212000
0.499375000	0.501078000	0.123652000
0.124754000	0.501319000	0.998454000
0.000226000	0.876254000	0.998630000
0.250000000	0.874303000	0.498505000
0.250000000	0.874931000	0.000000000
0.124644000	0.001297000	0.998484000
0.500055000	0.996727000	0.003998000
0.500151000	0.247419000	0.753942000
0.250000000	0.246855000	0.003638000
0.250000000	0.236874000	0.501807000
0.250000000	0.754213000	0.504111000
0.750000000	0.248222000	0.503798000
0.000000000	0.747681000	0.253772000
0.750000000	0.748161000	0.504086000
0.000000000	0.247419000	0.753942000
0.000000000	0.747568000	0.754343000
0.500410000	0.247436000	0.253512000
0.750000000	0.746807000	0.004115000
0.500301000	0.747567000	0.754343000
0.250000000	0.746758000	0.004558000
0.000000000	0.247435000	0.253512000
0.000000000	0.498002000	0.503657000
0.000000000	0.996727000	0.003998000
0.500390000	0.747681000	0.253772000
0.750000000	0.997298000	0.754036000
0.250000000	0.497590000	0.753805000
0.250000000	0.997368000	0.254187000
0.750000000	0.497509000	0.253972000
0.750000000	0.997386000	0.254054000
0.250000000	0.997903000	0.754245000
0.000000000	0.496793000	0.003881000
0.000000000	0.998217000	0.503892000
0.500069000	0.496793000	0.003881000
0.250000000	0.497079000	0.253362000
0.500626000	0.498002000	0.503657000
0.750000000	0.246773000	0.003873000
0.750000000	0.497366000	0.754016000
0.500084000	0.998217000	0.503892000

=====

Appendix S3. POSCAR file of the 4×4×4 supercell of Fe-doped KTN

=====

Fe-KTN

1.0

15.9399995804 0.0000000000 0.0000000000
0.0000000000 15.9695816040 0.0000000000
0.0000000000 0.0000000000 15.9695816040

Fe K Nb O Ta
1 64 31 191 32

Direct

0.248133000 0.506900000 0.503556000
0.374985000 0.872764000 0.627148000
0.874023000 0.373725000 0.126531000
0.874847000 0.374728000 0.625543000
0.874251000 0.873455000 0.626221000
0.125201000 0.375024000 0.625507000
0.375863000 0.122322000 0.375174000
0.873912000 0.873265000 0.126464000
0.623887000 0.373770000 0.126580000
0.623922000 0.374868000 0.625572000
0.624061000 0.873237000 0.126479000
0.624378000 0.873477000 0.626262000
0.124043000 0.873237000 0.126430000
0.124625000 0.374267000 0.876310000
0.123530000 0.872920000 0.627002000
0.874202000 0.373734000 0.876309000
0.123141000 0.374388000 0.376768000
0.375250000 0.374467000 0.376644000
0.123206000 0.872795000 0.376136000
0.124124000 0.873257000 0.876437000
0.623996000 0.374709000 0.377636000
0.623840000 0.373716000 0.876387000
0.373745000 0.374957000 0.625865000
0.624034000 0.873205000 0.876503000
0.375059000 0.872714000 0.376164000
0.373470000 0.374194000 0.876229000
0.874148000 0.873468000 0.376998000
0.373846000 0.873234000 0.876449000
0.874564000 0.374627000 0.377646000
0.624264000 0.873435000 0.377014000
0.373896000 0.873154000 0.126522000
0.873999000 0.873277000 0.876448000
0.124291000 0.374274000 0.127423000
0.373822000 0.123707000 0.876348000
0.373386000 0.622737000 0.875937000
0.874280000 0.123776000 0.376894000
0.873998000 0.123556000 0.876479000
0.874764000 0.622700000 0.377896000
0.124151000 0.123776000 0.876282000
0.124888000 0.622674000 0.378923000
0.874138000 0.623122000 0.876353000
0.623962000 0.123711000 0.376888000
0.122651000 0.122356000 0.375121000
0.623922000 0.622712000 0.377897000
0.623923000 0.623202000 0.876359000

0.123966000	0.123612000	0.126592000
0.123119000	0.122525000	0.627352000
0.124585000	0.622736000	0.876034000
0.125989000	0.621549000	0.624431000
0.623997000	0.123570000	0.876500000
0.875011000	0.622590000	0.625413000
0.124381000	0.622841000	0.127630000
0.373489000	0.622896000	0.127706000
0.623958000	0.123525000	0.126556000
0.624061000	0.123772000	0.626360000
0.623912000	0.623291000	0.126526000
0.623872000	0.622606000	0.625445000
0.373716000	0.374173000	0.127503000
0.375700000	0.122527000	0.627522000
0.373092000	0.622794000	0.378996000
0.372417000	0.621918000	0.624521000
0.873917000	0.123563000	0.126475000
0.874450000	0.123888000	0.626296000
0.874047000	0.623282000	0.126440000
0.373941000	0.123617000	0.126694000
0.747289000	0.746186000	0.253799000
0.997280000	0.246575000	0.003554000
0.998426000	0.248314000	0.503593000
0.497184000	0.246520000	0.003707000
0.499033000	0.248416000	0.503685000
0.497216000	0.746165000	0.003649000
0.247269000	0.746332000	0.752775000
0.247380000	0.247382000	0.253025000
0.247449000	0.247424000	0.752752000
0.747179000	0.246601000	0.253837000
0.498278000	0.746837000	0.503877000
0.747311000	0.746115000	0.753471000
0.997347000	0.746198000	0.003577000
0.000000000	0.746890000	0.503871000
0.747248000	0.996402000	0.003584000
0.247344000	0.746121000	0.254138000
0.497612000	0.496600000	0.253847000
0.747329000	0.246571000	0.753600000
0.748159000	0.997374000	0.503700000
0.997144000	0.996302000	0.253333000
0.497634000	0.496373000	0.752785000
0.497631000	0.996168000	0.253403000
0.497617000	0.996238000	0.753766000
0.997283000	0.996310000	0.753646000
0.247327000	0.996609000	0.003482000
0.248279000	0.990413000	0.503407000
0.748765000	0.497123000	0.503713000
0.997955000	0.496280000	0.752739000
0.247581000	0.496697000	0.006694000
0.997628000	0.496552000	0.253594000
0.747417000	0.496413000	0.003310000
0.002380000	0.251111000	0.624483000
0.250523000	0.500204000	0.870143000
0.751283000	0.001480000	0.373382000
0.751385000	0.001512000	0.873513000
0.751173000	0.501299000	0.373228000
0.751021000	0.501415000	0.873665000

0.251479000	0.126535000	0.251963000
0.251364000	0.624040000	0.252468000
0.751226000	0.125564000	0.248517000
0.001214000	0.751472000	0.624803000
0.751271000	0.625286000	0.248475000
0.751383000	0.625338000	0.748584000
0.001920000	0.251435000	0.124699000
0.251338000	0.002306000	0.873622000
0.001508000	0.751309000	0.124553000
0.249685000	0.495605000	0.381625000
0.500895000	0.251652000	0.124718000
0.251345000	0.624611000	0.745257000
0.251422000	0.000797000	0.373954000
0.751378000	0.125539000	0.748588000
0.251552000	0.125932000	0.746349000
0.001647000	0.625325000	0.998591000
0.501167000	0.751236000	0.124613000
0.501104000	0.375309000	0.748421000
0.501111000	0.875267000	0.248559000
0.751334000	0.501274000	0.625252000
0.003455000	0.125914000	0.498308000
0.001620000	0.125510000	0.998652000
0.750758000	0.374371000	0.498409000
0.001605000	0.625152000	0.498661000
0.497494000	0.126307000	0.498413000
0.001542000	0.875434000	0.748705000
0.501210000	0.125548000	0.998670000
0.501148000	0.625248000	0.998580000
0.001263000	0.001453000	0.374314000
0.001542000	0.001478000	0.874685000
0.501353000	0.001487000	0.874721000
0.001829000	0.501332000	0.874512000
0.501135000	0.001553000	0.374421000
0.500352000	0.501290000	0.374886000
0.501449000	0.751344000	0.624721000
0.500142000	0.251427000	0.624400000
0.498842000	0.624836000	0.498593000
0.001620000	0.875315000	0.248458000
0.001310000	0.501253000	0.374397000
0.501286000	0.751282000	0.873551000
0.251339000	0.876389000	0.248627000
0.250355000	0.628184000	0.499394000
0.751382000	0.126594000	0.998563000
0.750768000	0.626390000	0.498554000
0.251009000	0.251539000	0.124643000
0.251238000	0.255722000	0.620157000
0.251407000	0.749884000	0.123777000
0.750330000	0.126626000	0.498320000
0.251507000	0.126926000	0.000000000
0.751287000	0.251431000	0.123535000
0.751357000	0.626424000	0.998550000
0.251477000	0.625835000	0.998459000
0.001477000	0.126552000	0.248399000
0.001582000	0.626319000	0.248559000
0.501211000	0.126601000	0.248535000
0.251241000	0.752678000	0.624096000
0.501425000	0.126575000	0.748670000

0.001779000	0.375170000	0.748541000
0.751177000	0.251370000	0.623453000
0.751309000	0.751394000	0.123415000
0.501268000	0.626311000	0.748450000
0.500913000	0.375561000	0.248759000
0.001459000	0.126511000	0.748729000
0.751316000	0.876537000	0.248506000
0.501130000	0.626245000	0.248675000
0.251446000	0.000000000	0.624398000
0.500576000	0.251395000	0.373527000
0.500863000	0.251562000	0.873547000
0.501275000	0.751270000	0.373486000
0.251568000	0.376638000	0.250400000
0.251616000	0.376009000	0.746874000
0.001870000	0.251385000	0.873498000
0.251442000	0.876569000	0.749702000
0.751371000	0.376382000	0.748655000
0.001490000	0.751359000	0.873550000
0.500078000	0.501312000	0.874348000
0.251330000	0.002382000	0.125077000
0.250137000	0.122163000	0.499577000
0.250488000	0.500309000	0.126178000
0.249826000	0.494813000	0.613934000
0.751333000	0.001512000	0.124536000
0.751281000	0.001638000	0.624656000
0.751266000	0.376462000	0.248456000
0.001940000	0.251072000	0.373419000
0.501359000	0.875381000	0.748587000
0.751387000	0.876593000	0.748582000
0.001260000	0.751303000	0.373357000
0.001617000	0.375332000	0.248597000
0.750928000	0.501412000	0.124427000
0.001591000	0.376407000	0.998603000
0.000630000	0.375228000	0.498253000
0.500895000	0.876839000	0.498609000
0.501278000	0.876489000	0.998650000
0.126737000	0.001739000	0.248568000
0.751227000	0.751474000	0.623649000
0.127011000	0.501383000	0.250773000
0.126639000	0.501433000	0.746312000
0.001661000	0.626389000	0.748521000
0.626359000	0.001571000	0.248410000
0.626050000	0.501411000	0.248615000
0.501131000	0.376517000	0.998611000
0.626251000	0.501424000	0.748838000
0.126794000	0.251745000	0.998660000
0.625348000	0.751479000	0.748552000
0.126708000	0.751397000	0.998509000
0.371568000	0.256432000	0.498117000
0.626314000	0.251608000	0.998480000
0.626284000	0.750728000	0.498351000
0.626446000	0.001599000	0.748421000
0.626402000	0.751452000	0.998500000
0.375138000	0.251736000	0.998624000
0.129747000	0.256177000	0.498083000
0.499928000	0.375685000	0.498255000
0.125567000	0.751921000	0.498486000

0.625320000	0.250402000	0.498899000
0.876435000	0.251641000	0.248707000
0.376222000	0.251696000	0.249218000
0.875486000	0.251595000	0.998511000
0.874967000	0.751050000	0.498316000
0.875634000	0.249874000	0.498914000
0.375284000	0.751431000	0.998491000
0.376477000	0.751291000	0.748532000
0.875467000	0.751471000	0.998470000
0.125521000	0.751360000	0.748560000
0.625114000	0.251817000	0.248684000
0.876531000	0.751483000	0.748565000
0.125799000	0.251673000	0.249427000
0.125667000	0.251404000	0.747721000
0.125652000	0.751144000	0.248387000
0.375465000	0.751710000	0.498534000
0.376378000	0.251384000	0.747813000
0.625200000	0.251686000	0.747985000
0.625205000	0.751412000	0.248317000
0.876624000	0.251613000	0.748059000
0.876475000	0.751412000	0.248330000
0.376310000	0.751179000	0.248510000
0.251148000	0.257224000	0.377917000
0.374926000	0.501553000	0.746048000
0.000000000	0.876938000	0.498492000
0.624962000	0.500337000	0.499317000
0.751348000	0.751426000	0.874594000
0.375468000	0.001614000	0.748614000
0.374475000	0.501549000	0.250986000
0.875352000	0.001589000	0.248301000
0.875424000	0.001606000	0.748442000
0.875352000	0.501383000	0.248295000
0.751120000	0.251416000	0.374662000
0.875423000	0.501405000	0.749190000
0.376430000	0.001571000	0.998534000
0.751114000	0.751439000	0.374419000
0.375311000	0.001703000	0.248422000
0.876374000	0.002316000	0.498527000
0.251160000	0.752140000	0.374650000
0.877615000	0.500345000	0.499204000
0.876590000	0.501410000	0.998601000
0.125661000	0.000000000	0.498236000
0.250995000	0.251104000	0.873998000
0.375654000	0.000335000	0.498224000
0.251413000	0.750026000	0.874424000
0.126656000	0.001635000	0.748488000
0.375904000	0.501551000	0.998277000
0.751314000	0.251432000	0.874552000
0.876553000	0.001596000	0.998471000
0.128096000	0.498124000	0.499557000
0.372104000	0.498256000	0.499722000
0.624902000	0.501425000	0.998563000
0.001473000	0.001555000	0.623564000
0.001855000	0.501369000	0.123466000
0.501321000	0.001457000	0.123515000
0.500986000	0.001687000	0.623522000
0.624634000	0.002309000	0.498505000

0.001547000	0.001435000	0.123470000
0.625327000	0.001636000	0.998436000
0.251595000	0.375291000	0.998664000
0.001116000	0.501030000	0.623975000
0.751348000	0.375336000	0.998534000
0.750367000	0.875776000	0.498541000
0.751388000	0.875452000	0.998561000
0.500584000	0.501117000	0.623567000
0.499953000	0.501359000	0.123642000
0.125711000	0.501465000	0.998373000
0.001592000	0.876536000	0.998608000
0.250614000	0.873299000	0.498577000
0.251395000	0.875309000	0.000000000
0.125622000	0.001634000	0.998547000
0.497389000	0.996422000	0.003450000
0.497421000	0.246680000	0.753603000
0.247415000	0.247087000	0.003182000
0.249329000	0.235246000	0.501677000
0.248570000	0.744198000	0.503701000
0.748618000	0.247633000	0.503657000
0.997274000	0.746277000	0.253626000
0.748848000	0.746878000	0.503650000
0.997533000	0.246711000	0.753398000
0.997382000	0.746102000	0.753459000
0.497546000	0.246621000	0.253566000
0.747376000	0.746409000	0.003440000
0.497406000	0.746091000	0.753503000
0.247431000	0.746201000	0.003321000
0.997107000	0.246700000	0.253457000
0.004285000	0.497286000	0.503309000
0.997354000	0.996500000	0.003360000
0.497515000	0.746211000	0.253715000
0.747512000	0.996478000	0.753485000
0.247927000	0.496729000	0.751677000
0.247570000	0.996150000	0.253494000
0.747711000	0.496540000	0.253580000
0.747453000	0.996481000	0.253540000
0.247576000	0.996687000	0.753257000
0.997801000	0.496501000	0.003042000
0.998319000	0.997333000	0.503454000
0.497312000	0.496524000	0.003238000
0.247784000	0.496579000	0.261054000
0.492855000	0.497458000	0.503306000
0.747273000	0.246625000	0.003586000
0.747860000	0.496426000	0.753094000
0.498180000	0.997118000	0.503506000

=====

Appendix S4. POSCAR file of the 4×4×4 supercell of Mn-doped KTN

Mn-KTN

1.0

```
15.9399995804      0.0000000000      0.0000000000
0.0000000000      15.9695816040      0.0000000000
0.0000000000      0.0000000000      15.9695816040
```

```
Mn   K   Nb   O   Ta
1   64  31  191  32
```

Direct

```
0.249990000      0.503583000      0.502513000
0.374479000      0.374595000      0.126391000
0.374668000      0.374768000      0.626239000
0.374862000      0.873536000      0.126243000
0.375506000      0.873135000      0.626918000
0.875055000      0.373932000      0.126272000
0.875256000      0.374600000      0.625808000
0.874998000      0.873593000      0.626163000
0.125511000      0.374596000      0.126389000
0.875027000      0.873562000      0.876229000
0.125137000      0.873535000      0.126244000
0.124485000      0.873136000      0.626917000
0.624938000      0.373933000      0.126272000
0.624738000      0.374601000      0.625810000
0.624997000      0.873513000      0.126200000
0.624996000      0.873594000      0.626162000
0.125325000      0.374769000      0.626233000
0.125712000      0.374556000      0.875814000
0.124204000      0.374705000      0.375843000
0.875094000      0.373948000      0.876197000
0.875198000      0.374512000      0.376914000
0.374832000      0.873639000      0.876217000
0.374277000      0.374554000      0.875815000
0.375791000      0.374704000      0.375838000
0.375665000      0.873076000      0.375713000
0.624968000      0.873504000      0.376428000
0.624896000      0.373949000      0.876197000
0.624799000      0.374511000      0.376914000
0.125158000      0.873639000      0.876216000
0.124335000      0.873075000      0.375713000
0.624963000      0.873563000      0.876228000
0.875032000      0.873505000      0.376432000
0.875010000      0.623486000      0.876270000
0.875001000      0.873514000      0.126197000
0.374863000      0.123920000      0.876049000
0.373748000      0.622081000      0.378028000
0.374199000      0.622832000      0.875655000
0.875223000      0.123758000      0.376335000
0.875013000      0.123817000      0.876232000
0.875467000      0.622753000      0.377248000
0.124013000      0.123088000      0.375147000
0.125129000      0.123918000      0.876049000
0.375981000      0.123088000      0.375147000
0.125790000      0.622833000      0.875653000
0.624771000      0.123757000      0.376336000
```

0.624979000	0.123816000	0.876231000
0.624537000	0.622755000	0.377246000
0.624980000	0.623485000	0.876270000
0.125042000	0.123864000	0.126412000
0.124144000	0.123047000	0.627043000
0.126251000	0.622080000	0.378028000
0.126587000	0.621776000	0.625077000
0.624972000	0.123771000	0.126197000
0.624782000	0.123841000	0.626215000
0.624959000	0.623513000	0.126261000
0.624533000	0.622781000	0.625676000
0.374950000	0.123865000	0.126412000
0.375851000	0.123047000	0.627043000
0.374218000	0.622814000	0.126741000
0.125780000	0.622814000	0.126741000
0.373401000	0.621777000	0.625077000
0.875024000	0.123770000	0.126198000
0.875211000	0.123842000	0.626214000
0.875041000	0.623512000	0.126260000
0.875459000	0.622781000	0.625675000
0.499504000	0.746016000	0.503898000
0.749980000	0.246265000	0.253891000
0.249982000	0.746061000	0.753089000
0.250015000	0.745775000	0.254010000
0.249982000	0.247301000	0.753077000
0.249979000	0.247352000	0.252763000
0.000129000	0.745946000	0.003808000
0.499842000	0.745945000	0.003808000
0.500292000	0.247443000	0.503697000
0.499922000	0.246357000	0.003765000
0.000463000	0.746016000	0.503900000
0.999692000	0.247442000	0.503698000
0.000041000	0.246359000	0.003763000
0.749982000	0.745957000	0.753684000
0.750022000	0.745876000	0.253883000
0.749982000	0.246357000	0.753787000
0.000075000	0.496223000	0.253559000
0.749988000	0.996628000	0.503709000
0.999815000	0.996008000	0.253488000
0.999861000	0.996120000	0.753793000
0.499733000	0.496107000	0.753217000
0.500151000	0.996006000	0.253485000
0.500077000	0.996124000	0.753791000
0.499906000	0.496221000	0.253561000
0.249987000	0.996576000	0.003542000
0.249989000	0.990707000	0.503326000
0.749979000	0.496193000	0.003500000
0.749990000	0.496378000	0.503695000
0.000226000	0.496105000	0.753211000
0.249977000	0.496380000	0.002875000
0.749987000	0.996230000	0.003705000
0.250011000	0.125956000	0.745997000
0.250001000	0.494414000	0.378877000
0.250007000	0.500682000	0.871964000
0.750008000	0.001469000	0.373654000
0.750009000	0.001572000	0.873681000
0.750010000	0.501335000	0.373728000

0.750009000	0.501421000	0.874042000
0.250007000	0.126562000	0.252405000
0.250007000	0.624350000	0.251939000
0.001069000	0.251248000	0.624364000
0.750006000	0.125484000	0.248609000
0.750010000	0.125479000	0.748440000
0.750008000	0.625309000	0.248711000
0.750010000	0.625441000	0.748547000
0.000545000	0.251550000	0.124814000
0.250009000	0.002530000	0.873738000
0.000140000	0.751429000	0.124746000
0.000027000	0.751210000	0.624570000
0.499475000	0.251552000	0.124814000
0.250010000	0.625154000	0.747175000
0.250009000	0.001301000	0.374308000
0.000575000	0.501408000	0.374904000
0.499869000	0.751428000	0.124747000
0.000251000	0.875315000	0.248676000
0.000041000	0.875418000	0.748468000
0.499798000	0.375323000	0.248762000
0.499991000	0.375150000	0.748490000
0.499756000	0.875315000	0.248676000
0.750011000	0.501253000	0.625120000
0.003194000	0.126160000	0.498405000
0.000137000	0.125550000	0.998625000
0.000216000	0.375323000	0.248760000
0.002224000	0.624637000	0.498743000
0.496817000	0.126163000	0.498406000
0.499881000	0.125551000	0.998625000
0.497789000	0.624633000	0.498742000
0.499879000	0.625418000	0.998725000
0.000075000	0.001593000	0.374528000
0.000066000	0.001538000	0.874644000
0.499949000	0.001538000	0.874643000
0.999669000	0.501377000	0.874890000
0.499940000	0.001594000	0.374527000
0.000140000	0.625419000	0.998724000
0.499989000	0.751208000	0.624569000
0.498947000	0.251250000	0.624363000
0.499441000	0.501408000	0.374909000
0.499932000	0.751424000	0.873771000
0.250006000	0.123560000	0.499748000
0.250005000	0.627580000	0.499472000
0.250009000	0.626298000	0.998709000
0.750009000	0.126408000	0.998558000
0.750008000	0.626189000	0.498598000
0.750009000	0.626364000	0.998659000
0.250006000	0.250719000	0.124838000
0.250008000	0.257640000	0.620490000
0.250007000	0.750638000	0.124165000
0.750006000	0.126354000	0.498336000
0.250010000	0.126534000	0.999242000
0.750008000	0.251406000	0.123850000
0.750007000	0.751508000	0.123787000
0.000080000	0.126353000	0.248528000
0.999903000	0.126317000	0.748604000
0.000219000	0.626102000	0.248708000

0.000066000	0.626209000	0.748488000
0.499933000	0.126353000	0.248528000
0.250008000	0.752704000	0.624209000
0.500114000	0.126318000	0.748604000
0.000028000	0.375149000	0.748491000
0.750007000	0.251363000	0.623537000
0.499953000	0.626208000	0.748487000
0.499789000	0.626099000	0.248709000
0.750006000	0.876302000	0.248595000
0.000081000	0.751135000	0.373806000
0.000082000	0.751426000	0.873771000
0.499196000	0.251214000	0.373834000
0.499528000	0.251508000	0.873733000
0.499929000	0.751135000	0.373807000
0.250008000	0.376001000	0.248280000
0.250011000	0.375281000	0.750015000
0.250006000	0.876408000	0.248923000
0.000491000	0.251507000	0.873734000
0.250011000	0.876661000	0.749496000
0.750010000	0.376083000	0.748651000
0.500346000	0.501376000	0.874890000
0.750010000	0.876410000	0.748464000
0.250007000	0.002663000	0.125248000
0.250010000	0.000342000	0.624212000
0.250006000	0.500709000	0.125013000
0.250000000	0.494971000	0.621571000
0.750008000	0.001574000	0.124623000
0.750009000	0.001560000	0.624470000
0.750006000	0.376151000	0.248620000
0.000821000	0.251212000	0.373833000
0.499977000	0.875419000	0.748468000
0.750008000	0.501439000	0.124848000
0.750010000	0.374473000	0.498439000
0.999021000	0.375477000	0.498367000
0.000101000	0.876403000	0.998620000
0.500993000	0.375482000	0.498367000
0.499983000	0.376203000	0.998709000
0.500300000	0.876453000	0.498601000
0.499915000	0.876404000	0.998621000
0.125599000	0.001896000	0.248560000
0.750006000	0.751528000	0.623670000
0.125763000	0.501584000	0.249959000
0.125047000	0.501630000	0.748703000
0.999715000	0.876454000	0.498600000
0.625432000	0.001659000	0.248359000
0.625360000	0.501559000	0.248898000
0.625573000	0.501714000	0.749175000
0.128721000	0.258099000	0.498393000
0.125691000	0.251760000	0.998636000
0.624496000	0.751656000	0.748364000
0.125551000	0.751782000	0.998529000
0.624763000	0.250235000	0.498847000
0.625378000	0.251640000	0.998479000
0.625272000	0.750901000	0.498393000
0.625461000	0.001674000	0.748249000
0.625421000	0.751736000	0.998483000
0.125271000	0.752440000	0.498488000

0.374323000	0.251760000	0.998635000
0.375296000	0.751266000	0.248535000
0.375347000	0.251267000	0.747680000
0.375181000	0.251662000	0.249389000
0.874636000	0.251639000	0.998479000
0.874736000	0.750904000	0.498396000
0.875246000	0.250231000	0.498848000
0.374460000	0.751782000	0.998528000
0.375465000	0.751500000	0.748416000
0.874589000	0.751736000	0.998482000
0.875580000	0.251725000	0.248758000
0.124548000	0.751500000	0.748417000
0.875571000	0.751535000	0.248484000
0.875517000	0.751657000	0.748364000
0.124831000	0.251662000	0.249390000
0.124667000	0.251267000	0.747679000
0.124719000	0.751265000	0.248535000
0.374737000	0.752437000	0.498489000
0.624429000	0.251725000	0.248757000
0.624426000	0.251566000	0.747963000
0.624444000	0.751533000	0.248485000
0.875586000	0.251565000	0.747965000
0.371290000	0.258105000	0.498393000
0.250007000	0.258689000	0.377981000
0.125466000	0.001795000	0.748365000
0.250008000	0.752339000	0.374778000
0.750009000	0.251405000	0.874601000
0.750008000	0.751467000	0.374649000
0.750007000	0.751569000	0.874615000
0.374546000	0.001796000	0.748365000
0.374248000	0.501584000	0.249963000
0.374964000	0.501631000	0.748699000
0.874577000	0.001661000	0.248359000
0.874550000	0.001675000	0.748248000
0.874651000	0.501557000	0.248894000
0.750008000	0.251367000	0.374791000
0.874439000	0.501714000	0.749177000
0.375403000	0.001803000	0.998485000
0.374410000	0.001897000	0.248558000
0.375641000	0.501701000	0.998621000
0.875799000	0.002518000	0.498412000
0.875520000	0.001794000	0.998393000
0.876227000	0.500594000	0.499369000
0.875464000	0.501719000	0.999091000
0.125070000	0.000773000	0.498181000
0.250007000	0.250025000	0.874285000
0.374942000	0.000776000	0.498181000
0.250007000	0.751163000	0.874691000
0.371723000	0.494745000	0.500346000
0.128281000	0.494747000	0.500345000
0.124611000	0.001803000	0.998485000
0.624493000	0.001794000	0.998393000
0.623798000	0.500593000	0.499369000
0.624548000	0.501718000	0.999091000
0.000217000	0.001673000	0.623570000
0.000238000	0.501408000	0.123998000
0.000136000	0.501211000	0.624090000

0.499903000	0.001518000	0.123771000
0.499799000	0.001674000	0.623568000
0.624211000	0.002519000	0.498412000
0.000112000	0.001517000	0.123772000
0.250011000	0.374809000	0.998762000
0.250009000	0.873787000	0.498747000
0.250009000	0.875718000	0.999273000
0.750009000	0.375263000	0.998637000
0.750006000	0.875587000	0.498482000
0.750009000	0.875545000	0.998569000
0.000035000	0.376203000	0.998708000
0.499878000	0.501211000	0.624087000
0.499779000	0.501408000	0.124000000
0.124371000	0.501701000	0.998619000
0.500130000	0.246444000	0.253540000
0.000022000	0.745956000	0.753679000
0.000020000	0.745927000	0.253708000
0.000017000	0.246516000	0.753653000
0.999832000	0.246445000	0.253538000
0.749982000	0.746209000	0.503655000
0.249984000	0.746172000	0.003327000
0.749991000	0.246868000	0.503662000
0.249983000	0.743846000	0.503659000
0.249997000	0.237066000	0.501751000
0.249983000	0.246834000	0.003101000
0.499947000	0.246519000	0.753654000
0.499977000	0.996310000	0.003499000
0.749985000	0.746277000	0.003604000
0.749983000	0.246474000	0.003699000
0.749981000	0.496244000	0.753289000
0.749972000	0.996339000	0.753579000
0.495167000	0.496235000	0.503423000
0.249991000	0.496110000	0.257900000
0.499780000	0.496291000	0.003181000
0.000117000	0.996499000	0.503467000
0.004810000	0.496234000	0.503422000
0.000174000	0.496293000	0.003179000
0.999996000	0.996311000	0.003501000
0.749984000	0.996206000	0.253545000
0.749991000	0.496235000	0.253556000
0.249971000	0.996651000	0.753241000
0.249985000	0.996119000	0.253424000
0.249978000	0.496187000	0.747825000
0.500014000	0.745926000	0.253707000
0.499943000	0.745957000	0.753679000
0.499859000	0.996497000	0.503465000

=====

Appendix S5. POSCAR file of the $\sqrt{2} \times \sqrt{2} \times 2$ supercell of orthorhombic KTN

O-phase-KTN

1.0

5.6461000443	0.0000000000	0.0000000000
0.0000000000	7.9699997902	0.0000000000
0.0000000000	0.0000000000	5.6461000443

K	Ta	Nb	O
4	2	2	12

Direct

0.000000000	0.749950000	0.507470000
0.000000000	0.250050000	0.507470000
0.500000000	0.249950000	0.007470000
0.500000000	0.750050000	0.007470000
0.000000000	0.000000000	0.015490000
0.500000000	0.500000000	0.515490000
0.000000000	0.500000000	0.015700000
0.500000000	0.000000000	0.515700000
0.000000000	0.248260000	0.994740000
0.000000000	0.751740000	0.994740000
0.500000000	0.748260000	0.494740000
0.500000000	0.251740000	0.494740000
0.247100000	0.000000000	0.242280000
0.752900000	0.000000000	0.242280000
0.747100000	0.500000000	0.742280000
0.252900000	0.500000000	0.742280000
0.750230000	0.000000000	0.745420000
0.249770000	0.000000000	0.745420000
0.250230000	0.500000000	0.245420000
0.749770000	0.500000000	0.245420000

Appendix S6. POSCAR file of the $\sqrt{2} \times \sqrt{2} \times 2$ supercell of tetragonal KTN

T-phase-KTN

1.0

```
5.6446876526 0.0000000000 0.0000000000
0.0000000000 5.6446876526 0.0000000000
0.0000000000 0.0000000000 8.0139999390
```

```
K Ta Nb O
4 2 2 12
```

Direct

```
0.000000000 0.500000000 0.255590000
0.500000000 0.000000000 0.255590000
0.500000000 0.000000000 0.755590000
0.000000000 0.500000000 0.755590000
0.000000000 0.000000000 0.010360000
0.500000000 0.500000000 0.510360000
0.000000000 0.000000000 0.510420000
0.500000000 0.500000000 0.010420000
0.000000000 0.000000000 0.244620000
0.500000000 0.500000000 0.744620000
0.000000000 0.000000000 0.747600000
0.500000000 0.500000000 0.247600000
0.748270000 0.748270000 0.497910000
0.251730000 0.251730000 0.497910000
0.251730000 0.748270000 0.497910000
0.748270000 0.251730000 0.497910000
0.248270000 0.248270000 0.997910000
0.751730000 0.751730000 0.997910000
0.751730000 0.248270000 0.997910000
0.248270000 0.751730000 0.997910000
```

Response to Reviewers' comments and the editorial requests in the Author Checklist

We sincerely thank the reviewers and the editors for the recommendations and suggestions to further improve our work. We have revised our manuscript accordingly and the point-by-point responses are enclosed below. We hope our responses, along with the revisions we have made, will now make this manuscript suitable for publication in *Nature Communications*.

[Reviewers' comments are in black; Author responses are in blue.]

Response to the editorial requests:

We greatly appreciate the editors for the time and efforts on our manuscript. We have revised our paper to address the editorial requests. A completed copy of Author Checklist has been uploaded.

Response to Reviewers' comments:

Reviewer #2

The authors have revised the manuscript according to reviewer suggestions and the manuscript can be accepted for publication.

RESPONSE: We sincerely appreciate the reviewer's positive comment and the suggestion for the acceptance of our manuscript.

(For the record, I still think that the ELF plots are not a valuable contribution to otherwise rigorous paper. The literature added by the authors does not include any rigorous explanation of ELF values smaller than 0.5. Instead, just similar plots as in the current paper. The authors state in the rebuttal that "For example, in BiFeO₃ and GaFeO₃ materials, the finite ELF values of about 0.3~0.4 between A-site atoms and oxygen atoms can indicate the hybridization interaction and some degree of covalent characteristics" (2,3). Here, "hybridization interaction" is an ill-defined term with no meaning and the claim about covalent characteristics has no foundation. I suggest that

the authors go back to the fundamental papers regarding ELF, written before people started misusing and misinterpreting ELF values smaller than 0.5. In his 1998 review, Burdett writes: "Thus it seems that ELF values below $ELF = 0.5$ can be interpreted as representing regions where either there is **very little electron density**, such as between atomic shells, or in regions where the contribution of a great number of nodes to ELF far outweighs the contribution of the density." (Electron Localization in Molecules and Solids: The Meaning of ELF, J. Phys. Chem. A 1998, 102, 31, 6366–6372). This statement is based on rigorous analysis, which is typically lacking from ELF papers using ELF scale [0..1].)

RESPONSE: We sincerely thank the reviewer for the knowledgeable comment. The recommended paper (J. Phys. Chem. A 1998, 102, 31, 6366–6372) is of great value to gain a deeper understanding of ELF. As described in the recommended paper, ELF is algebraically defined for a system with σ -spin electron density $\rho_\sigma(x, y, z)$ and a set of occupied molecular orbitals $\{\psi_{i\sigma}\}$ as

$$ELF = \frac{1}{1 + \left[D_\sigma(x, y, z) / D_{\sigma, \text{gas}}(x, y, z) \right]^2} \quad (1)$$

where

$$D_\sigma(x, y, z) / D_{\sigma, \text{gas}}(x, y, z) = 0.3483 \rho_\sigma^{-5/3} \left[\sum_i |\nabla \psi_{i\sigma}|^2 - \frac{1}{8} |\nabla \rho_\sigma|^2 / \rho_\sigma \right] \quad (2)$$

D_σ is the leading term in the Taylor expansion of the spherically averaged σ -spin pair probability, and the subscript “gas” refers to the corresponding value for the homogeneous electron gas. For the vast majority of systems of chemical interest, the first term is much larger than the second term in Eq. (2). The ratio $\sum_i |\nabla \psi_i|^2 / \rho^{5/3}$ is the crucial parameter for ELF. The kinetic energy density of the electrons is directly related to the slope of the wave function, or gradient, $\nabla \psi_i$. **Hence, in addition to the electron density, the $\nabla \psi_i$ also influences ELF. The synergistic interaction between ρ and $\sum_i |\nabla \psi_i|^2$ determines the value of ELF.**

The focus of our calculations aims to highlight the different effects induced by Fe and Mn doping. To this end, we do not limit ourselves to the exploration of a single

case, but focus on an in-depth analysis by comparing the ELF results for a variety of structures. Our results show that the doping of Fe and Mn obviously influences the ELF values. Moreover, different variations of ρ and $\nabla\psi_i$ result in differences in the effect of these two dopants on the ELF. **The ELF results are meaningful from a comparative perspective. Given the close connection between ELF and wave function, there probably lie more interesting effects behind this phenomenon that are worth exploring, inspiring future research. Hence, the ELF results are maintained in the revised manuscript.** We will also proceed with additional related studies in the future to elucidate the deeper reasons responsible for the diverse ELF changes from different transition metal doping.